# Multigroup analysis of compositions of microbiomes with covariate adjustments and repeated measures

**Huang Lin** [1,2] **& Shyamal Das Peddada** [1] ✉

Microbiome differential abundance analysis methods for two groups are well-established in the literature. However, many microbiome studies involve more than two groups, sometimes even ordered groups such as stages of a disease, and require different types of comparison. Standard pairwise comparisons are inefficient in terms of power and false discovery rates. In this Article, we propose a general framework, ANCOM-BC2, for performing a wide range of multigroup analyses with covariate adjustments and repeated measures. We illustrate our methodology through two real datasets. The first example explores the effects of aridity on the soil microbiome, and the second example investigates the effects of surgical interventions on the microbiome of patients with inflammatory bowel disease.

The differential abundance (DA) analysis of microbial taxa between two study groups is well-studied in the literature. Often two types of parameter are considered, namely the relative abundance and the absolute abundance of a taxon in a unit volume of an ecosystem. There exist several methods in the literature that can be used for performing differential relative abundance analysis between two groups such as count regression for correlated observations with the beta-binomial (CORNCOB)[1]. While relative abundance (same as relative proportions) is a natural measure to consider, the DA of relative abundances has an important limitation. Specifically, differences in the absolute abundance of a single taxon between two groups may result in differential relative abundances of all taxa between the two groups[2,3]. While this is mathematically correct, it does not help the researcher to discover the specific taxon that was DA between the two groups.

As an alternative to differential relative abundance analysis, several methods proposed in the literature can be used for differential absolute abundance analysis (hereafter referred to as DA analysis), which is the focus of this Article. Some examples include analysis of composition of microbiomes (ANCOM)[2], analysis of compositions of microbiomes with bias correction (ANCOM-BC)[3], linear models for DA analysis (LinDA)[4] and logistic compositional analysis (LOCOM)[5]. However, the methodology for multigroup DA analysis is not well-developed in the literature. Some researchers perform a series of pairwise tests with a false discovery rate (FDR) control within each pairwise comparison and

pool the results from all such pairwise comparisons to interpret the data. Such a strategy does not account for the fact that multiple tests and multiple pairwise comparisons are being performed and hence the overall FDR is not controlled.

Standard procedures, such as the Benjamini–Hochberg procedure[6], are designed for testing multiple hypotheses between two groups. When there are more than two groups, the standard concept of FDR, and methods controlling the corresponding error rates, need to be modified according to the study design and type of analyses to be performed[7–9]. Some examples of interest include the following. (1) Multiple pairwise comparisons, in which a dietitian may be interested in making all pairwise comparisons of the gut microbial compositions among participants receiving diets $D_1$, $D_2$ or $D_3$. Furthermore, for each pairwise comparison, the goal is often to identify taxa whose abundance increased (or decreased). (2) Multiple pairwise comparisons against a specific reference group, the same as in scenario (1), but the investigator is only interested in comparing groups $D_2$ and $D_3$ against $D_1$, the reference group. (3) Pattern analysis over ordered study groups, where, in some instances, an investigator may be interested in discovering trends or patterns in abundances of taxa over ordered groups, such as the health of participants, changes in climate, doses of a drug and so on. For instance, during normal pregnancy, women experience major changes in their gut and vaginal microbiome[10]. These changes are necessary for maternal metabolism, immune response and

[1]Biostatistics and Computational Biology Branch, NIEHS, NIH, Research Triangle Park, NC, USA. [2]Present address: Department of Epidemiology and Biostatistics, University of Maryland, College Park, MD, USA. ✉e-mail: shyamal.peddada@nih.gov

hormonal changes to support pregnancy and to provide healthy flora for babies at birth[11,12]. Thus, as the pregnancy progresses from the first to the third trimester, a researcher may be interested in discovering temporal changes in microbiota. Thus, in many scientific investigations, researchers are interested in studying changes in the microbiome over ordered conditions. The patterns of microbial abundance may not always be monotonic. They may display other shapes, such as an umbrella or an inverted umbrella with the location of the peak or trough unknown a priori. Additionally, depending on the scientific question of interest, repeated measures are taken on the same participant. Although the pattern analyses mentioned here could be accomplished by conducting a sequence of pairwise tests over adjacent ordered groups, such a strategy may have lower power than a test designed for pattern analysis, as will be demonstrated in the analysis of soil aridity data described later in this Article.

The objective of this Article is to develop methodologies for performing multigroup DA analyses. A formal methodology for performing such analyses does not appear to be available in the literature, with a few exceptions, such as ANCOM-II (ref. 13). While ANCOM-II considered the above testing problems, it does not develop a formal framework for bias correction. The more recent methodology LinDA[4], which uses a model similar to the one developed in ANCOM-II, does not address the above multigroup testing problems. Thus, there is a major gap in the literature for analyzing multigroup microbiome studies, which will be filled by the methodology developed in this Article called analysis of compositions of microbiomes with bias correction 2 (ANCOM-BC2).

Although the ANCOM-BC methodology accounted for sample-specific bias, for better control of FDR, ANCOM-BC2 also accounts for taxon-specific bias. This is important because sequencing efficiencies can vary across taxa, leading to a taxon-specific bias when some taxa are preferentially measured over others during sequencing. For example, gram-positive bacteria have stronger cell walls than gram-negative bacteria, making them harder to extract during the data preparation step. Consequently, gram-positive bacteria may be underrepresented in the observed counts, leading to biased results if taxon-specific biases are not properly accounted for in the analysis[14]. Also, it is well-known that small effect sizes are associated with small variances in high throughput data[15]. Consequently, in such cases, the value of the test statistics is inflated, resulting in a highly significant $P$ value. Inspired by the significance analysis of microarrays (SAM)[15] methodology, we regularize the variance to avoid inflated values for the test statistics and hence moderate the $P$ values for a better control of FDR. Lastly, zeros are a common problem for log-abundance based DA methods, including ANCOM-BC. Often such methods use pseudo-counts to deal with zero before taking logarithms. However, the choice of pseudo-count can affect the results for rare taxa containing excess zeros, which potentially leads to an inflated FDR[13,16,17]. To mitigate this issue, we conduct a sensitivity analysis to filter a DA taxon that potentially is a false positive. Details of the procedure are provided in the Methods section.

Using constrained statistical inference-based methods[7] and mixed directional FDR (mdFDR) methods for multiple pairwise comparisons[8,9], along with the above-noted modifications to ANCOM-BC, in this Article we develop ANCOM-BC2 for multigroup microbiome studies. ANCOM-BC2 allows modeling covariates as well as repeated measures. The performance of ANCOM-BC2 is evaluated using extensive simulation studies under a variety of settings. ANCOM-BC2 is also illustrated using two publicly available data, namely soil microbiome data and irritable bowel disease data.

## Results

### Simulations: settings

Inspired by applications, we conducted simulation studies under various scenarios incorporating different exposure types and covariate adjustments. We compared the performance of ANCOM-BC2, with

ANCOM-BC (ref. 3), as well as state-of-the-art DA methods for absolute abundances: (1) LinDA[4] and (2) LOCOM[5]. Although designed for relative abundances, CORNCOB, a DA method based on beta-binomial regression model, was also included in the simulation studies.

The absolute abundances were simulated using the Poisson log-normal (PLN) model as done in linear decomposition model framework[18]. The PLN model postulates that absolute abundance follows a Poisson distribution with a multivariate log-normal distribution for the mean. The population mean and covariance matrix for absolute abundance in the PLN model were derived from the upper respiratory tract (URT) microbiome data, featuring 60 samples and 382 operational taxonomic units (OTUs), extracted from the original 856-OTU dataset[19]. OTUs present in less than 5% of samples were omitted. It is important to note that ANCOM-BC2 is not based on PLN model and thus, this simulation set-up does not inherently favor ANCOM-BC2 over the competing methods described in this Article.

Motivated by the limitations of ANCOM-BC identified through our experience and in the literature, we conducted an exhaustive simulation study that includes edge cases where ANCOM-BC performs poorly. Additional details regarding the simulation design are provided in Extended Data Fig. 1. Many DA methods implicitly assume that many taxa (for example, more than 50%) are not DA. To understand the breakdown point of various methods, we varied the proportion of DA taxa from 5 to 90%. Our evaluation of pseudo-count effects on zeros led to two ANCOM-BC2 versions: ANCOM-BC2 (no filter) and ANCOM-BC2 (SS filter, where SS denotes sensitivity score), detailed in the Methods section. Notably, ANCOM-BC2 (SS filter) is intrinsically more conservative. For the control of FDR due to multiple testing, we favored the Holm–Bonferroni method[20] over the Benjamini–Hochberg procedure[6] for all DA methods. The Holm–Bonferroni method, which allows arbitrary dependence structure among the underlying $P$ values, is recognized to be robust to some extent for inaccurate $P$ values[21], a common problem with all DA methods. Further information regarding the simulation study set-up is provided in the Supplementary Methods.

### Simulations: continuous and binary exposures

Figure 1a presents the simulation results when the exposure variable is continuous. Both versions of ANCOM-BC2 had smaller FDR compared to other methods. ANCOM-BC2 (SS filter) consistently controlled FDR below the nominal level of 0.05. By contrast, the FDR of ANCOM-BC2 (no filter) increased with sample size, a consequence of excess zeros across the distribution of the exposure variable, which is more likely to generate false positives with a larger sample size. Both versions of ANCOM-BC2 generally outperformed all other methods, with ANCOM-BC2 (no filter) achieving the highest power. Conversely, all competing methods had considerably higher FDR than both versions of ANCOM-BC2. For instance, the FDR of LOCOM ranged from 5 to 40%. Similarly, LinDA and ANCOM-BC had FDRs ranging from 5 to 70%. LOCOM experienced a substantial decrease in power for small sample sizes. For example, the power was as low as 20% for $n = 10$. Although ANCOM-BC and LinDA had larger powers, they suffered from high FDR, exceeding the nominal level in most scenarios. We further note that as the sample size increased, the FDR of ANCOM-BC, LinDA and LOCOM increased. This suggests a systematic bias within these test statistics. The FDR of CORNCOB, a method designed for DA of relative abundances, consistently exceeded the nominal level and reached its maximum when a large number of taxa were differentially abundant (between 20 and 50%). This is attributed to the fact that differential absolute abundance in a single taxon could induce differential relative abundance of many null taxa[2,22].

Figure 1b presents the simulation results for DA analysis for a binary exposure. These results are generally consistent with those presented in Fig. 1a. The FDRs of competing methods were substantially inflated compared to the two versions of ANCOM-BC2, and those FDRs monotonically increased with sample size. The two versions

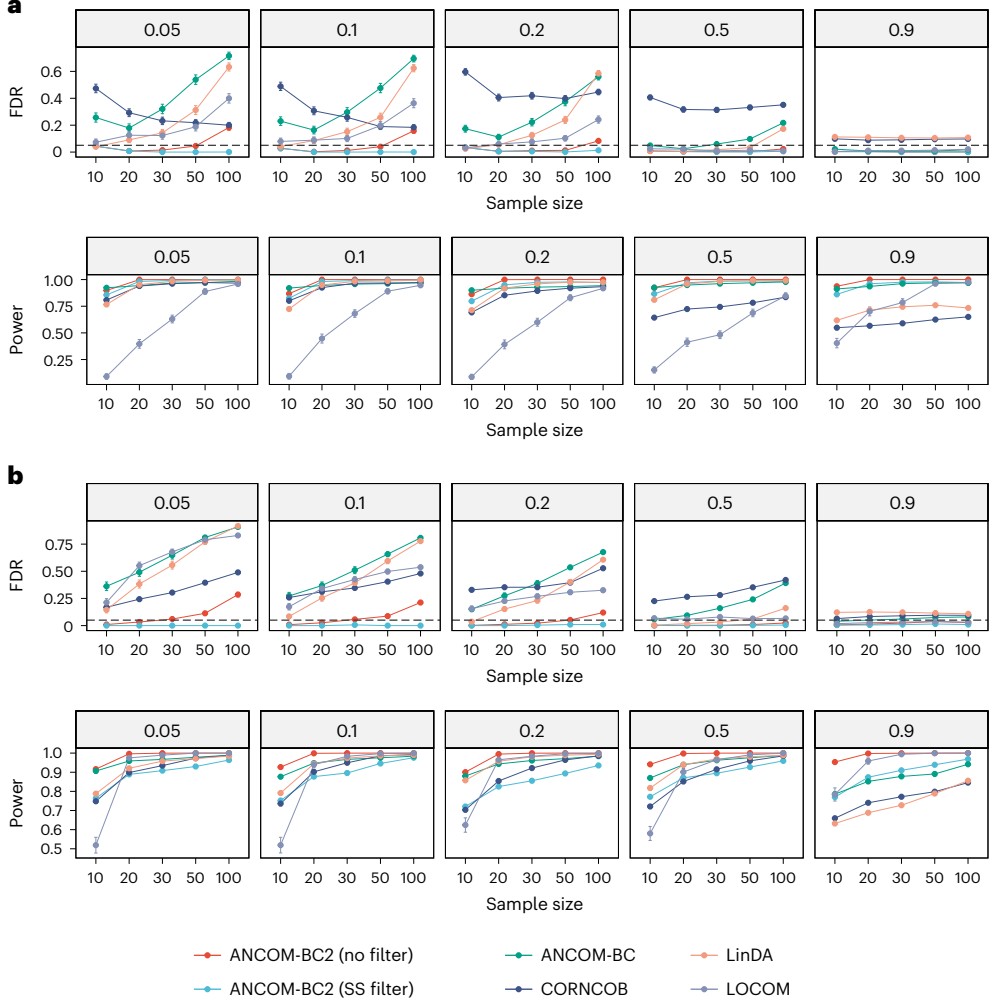

**Fig. 1 | FDR and power comparisons for continuous and binary exposures. a,b**, The FDR and power of various DA methods for continuous (**a**) and binary exposures (**b**) are summarized. Synthetic datasets were generated using the PLN model[18] based on the mean vector and covariance matrix estimated from the URT dataset[19]. The *x* axis represents the sample size (or sample size per group for the binary exposure), and the *y* axis shows the FDR or power. The dashed lines denote the nominal level of FDR (FDR = 0.05). The proportion of true DA taxa are provided in the top of each panel. The mean estimated FDR (or power) ± standard errors (indicated by error bars) derived from 100 simulation runs are provided in each panel.

of ANCOM-BC2 consistently maintained lower FDR than all competing methods. Similar to the continuous exposure variable case, ANCOM-BC2 (SS filter) always controlled the FDR at the nominal level, whereas ANCOM-BC2 (no filter) controlled FDR at the nominal for small to moderate sample sizes. For large sample sizes (for example, more than 50), it failed to control FDR within the nominal level but still had substantially lower FDR than LOCOM, LinDA, ANCOM-BC and CORNCOB. However, ANCOM-BC2 (no filter) had the highest power among all the methods. On the other hand, ANCOM-BC2 (SS filter) sacrificed about 10% of power, a concession that enables the control of FDR across all simulation settings.

To evaluate the power and FDR trade-off across the diverse DA methods, we computed the FDR adjusted power (FAP), as detailed in the Supplementary Methods. This measure (not a probability) is represented in relation to power in Extended Data Fig. 2. An elevated FAP indicates a superior power and FDR trade-off for a given power. Extended Data Fig. 2a corresponds to the continuous exposure case and Extended Data Fig. 2b pertains to the binary exposure case. From the cumulative distribution plots, we see that for any given power, both versions of ANCOM-BC2 have stochastically larger FAP values than all other methods (that is, their cumulative distribution functions are more to the right), with ANCOM-BC2 (SS filter) being stochastically the

largest. Since, in practice not all methods have the same FDR, hence to account for the power and FDR trade-off, we advocate the use of FAP as a measure for comparing DA methods.

## Simulations: multiple groups
The simulation settings for multigroup comparisons mimic those outlined in the previous section.

**Multiple pairwise comparisons against a reference group.** We assessed the performance of ANCOM-BC2 (SS filter) and ANCOM-BC2 (no filter), ANCOM-BC and LinDA across three experimental groups with covariate adjustments. LOCOM and CORNCOB were not included because they are not designed for multiple groups. As illustrated in Fig. 2a, both versions of ANCOM-BC2 yielded smaller mixed directional FDR (mdFDR)[8,9], compared to other methods. Note that mdFDR accounts for errors due to multiple testing, multiple comparisons and directional errors. Specifically, ANCOM-BC2 (SS filter) effectively controlled mdFDR below the nominal level of 0.05. Although in some cases it results in a loss of about 10–20% power, it ensures more stringent mdFDR control. Even with this power reduction, ANCOM-BC2 (SS filter) maintains a robust power (more than 0.8) in most scenarios. Without the filter, ANCOM-BC2 (no filter) remains to be the most

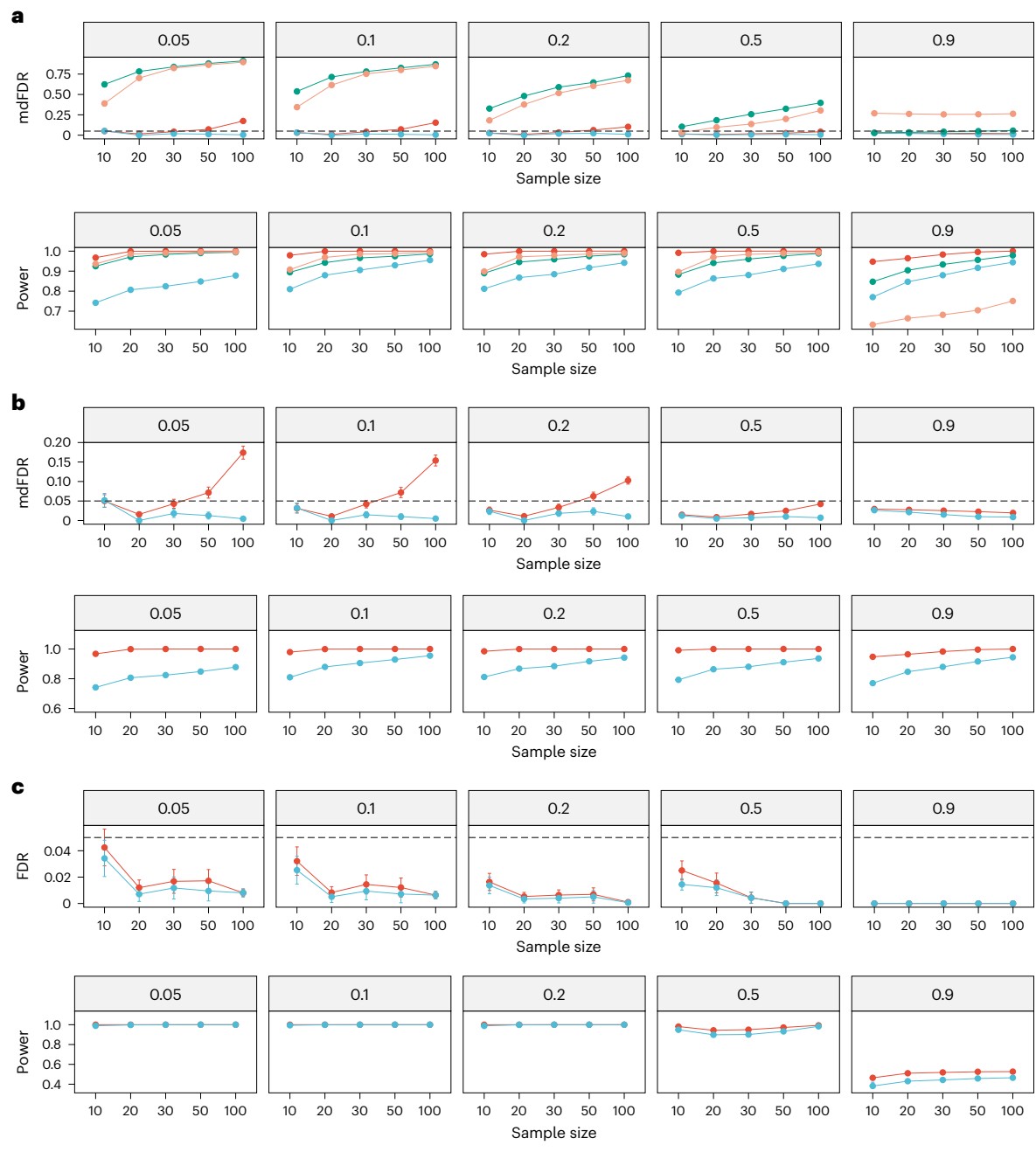

ANCOM-BC2 (no filter)   ANCOM-BC2 (SS filter)   ANCOM-BC   LinDA

**Fig. 2 | FDR (mdFDR) and power comparisons for multiple exposure groups.**
**a–c**, The FDR (mdFDR) and power of various DA methods for multiple pairwise comparisons against a reference group (**a**), multiple pairwise comparisons (**b**) and pattern analysis (**c**) are summarized. Synthetic datasets were generated using the PLN model[18] based on the mean vector and covariance matrix estimated from the URT dataset[19]. The *x* axis represents the sample size per group, and the *y* axis shows the FDR (mdFDR) or power. The dashed lines denote the nominal level of

FDR (FDR = 0.05) or mdFDR (mdFDR = 0.05). The proportion of true DA taxa are provided in the top of each panel. The mean estimated FDR (or power) ± standard errors (indicated by error bars) derived from 100 simulation runs are provided in each panel. Within the context of multiple pairwise comparisons, ANCOM-BC2 (SS filter) effectively controlled FDR (mdFDR) while maintaining power similar to ANCOM-BC2 (no filter).

powerful DA method of all. Despite its mdFDR occasionally surpassing 0.05 for larger sample sizes (more than 50), it was still markedly better than both LinDA and ANCOM-BC, which struggled to control mdFDR efficiently.

**Multiple pairwise comparisons.** We assessed ANCOM-BC2's performance when making all possible pairwise comparisons instead of comparing against a specific reference group as done above. Since the competing methods considered in this Article are not currently

designed for multiple pairwise comparisons, they are excluded. As depicted in Fig. 2b, ANCOM-BC2 (SS filter) effectively controlled the mdFDR below the nominal level of 0.05 while maintaining substantial power (more than 0.8) in most scenarios. However, as seen above, ANCOM-BC2 (no filter) controlled mdFDR within the nominal level for small sample sizes or when a large proportion of taxa are differentially abundant. However, when the sample sizes are large (for example, more than 50), it had an inflated mdFDR exceeding the nominal level.

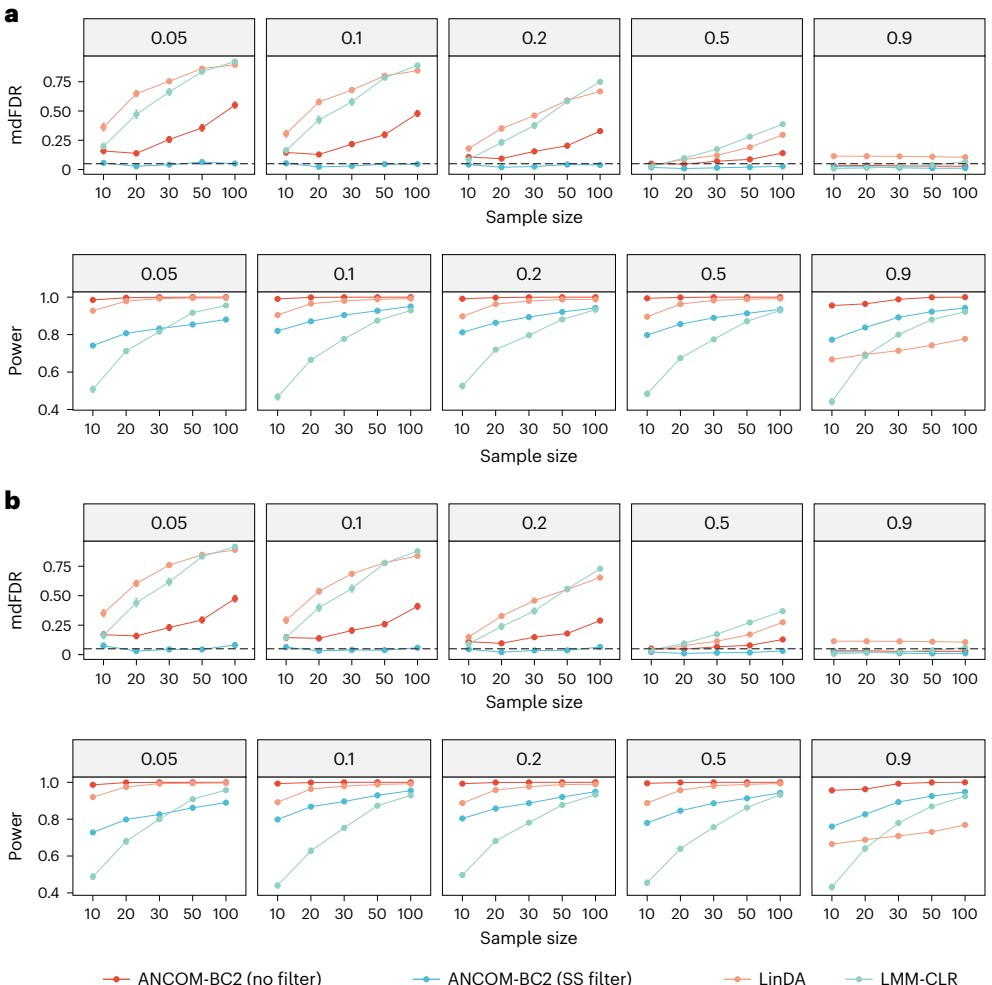

**Fig. 3 | mdFDR and power comparisons for correlated samples. a,b**, The mdFDR and power of various DA methods in a random intercept model (**a**) and a random coefficients model (**b**) are summarized. Synthetic datasets were generated using the PLN model[18] based on the mean and covariance estimated from the URT dataset[19]. The *x* axis represents the sample size per group, and the *y* axis shows the mdFDR or power. The dashed lines denote the nominal level of FDR (FDR = 0.05) or mdFDR (mdFDR = 0.05). The proportion of true DA taxa are provided in the top of each panel. The mean estimated mdFDR (or power) ± standard errors (indicated by error bars) derived from 100 simulation runs are provided in each panel.

**Pattern analysis.** Pattern analysis is another unique feature of ANCOM-BC2. In this simulation study, we modeled a scenario demonstrating a monotonically increasing pattern. Here, the log fold-change (denoted by $\delta$) among the DA (or nonnull) taxa between the second group and the reference group ranged from 0.5 to 2.0, and the log fold-change of the third group relative to the first group was taken to be $\delta + 1$. In this setting, a 'discovery' in pattern analysis refers to the identification of a taxon that displays a monotonically increasing pattern across all three groups. As described in Fig. 2c, both versions of ANCOM-BC2 controlled the FDR while maintaining high power exceeding 0.8 in most scenarios. Nonetheless, under the most extreme scenario where 90% of taxa were truly differentially abundant, ANCOM-BC2 encountered a power loss. The observed power loss is largely due to ANCOM-BC2's built-in bias correction, which assumes that there is a sufficient number of null taxa.

**Simulations: correlated samples**
In this section, we evaluated the performance of ANCOM-BC2 in comparison to LinDA when the samples across experimental groups were correlated, such as in a repeated measurement design. We also considered linear mixed model (LMM) on CLR-transformed data (LMM-CLR), a method commonly used for repeated measurements. The interpretation of LMM-CLR results differs from the previously

mentioned DA methods. According to LMM-CLR, a taxon is nonnull if it is differentially abundant relative to the geometric mean of all taxa, not its absolute. We included this method in our simulation study due to its frequent application in repeated measures analyses of microbiome data. ANCOM-BC, LOCOM and CORNCOB were excluded in this simulation as none of them are equipped to handle correlated experimental groups. We considered mixed-effects models with: (1) a random intercept and (2) a random intercept and a random slope. The random intercept had a standard deviation of 1 and the random slope had a standard deviation of 1.5, and both had mean zero. If both random effects were present, the correlation coefficient between them was set to 0.5. In each of these scenarios, the exposure variable consisted of three levels (that is, three experimental groups). The simulation study also included a continuous covariate. The remaining simulation settings adhered to those described in the previous sections (details in Supplementary Methods section). The simulation results for both scenarios are provided in Fig. 3. In each case, as in all previous settings, ANCOM-BC2 (SS filter) effectively controlled the mdFDR at or below the nominal level of 0.05, while maintaining substantial power (more than 0.8) in most of the simulation settings. On the other hand, ANCOM-BC2 (no filter) consistently exceeded the nominal mdFDR level of 0.05. Despite this, it had a larger power and smaller mdFDR than LinDA and LMM-CLR across all settings. LMM-CLR, generally exhibited the lowest

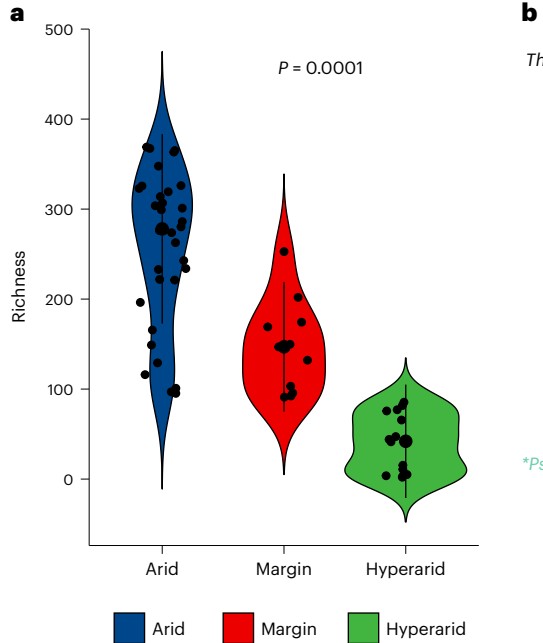

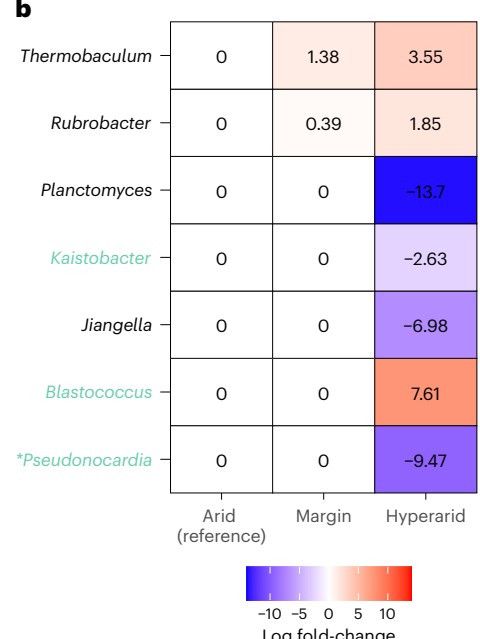

**Fig. 4 | DA analysis of desert soil microbial genera with increasing aridity.**
**a**, Violin plot illustrating the relationship between aridity and microbial richness. Samples encompass 63 biologically independent pits obtained from 18 distinct Atacama Desert sites in Chile[24]. Each violin's median value is signified by a central black dot, while the interquartile range is represented by a black bar. The violin's width mirrors the density of data points at each richness value. Individual data points are also displayed as jittered dots. A trend test using the constrained inference-based approach[7] suggests a significant decline in richness with increase in soil aridity ($P = 0.0001$). **b**, ANCOM-BC2 (no filter) pattern analysis heatmap in relation to aridity. Monotonically increasing and decreasing trends were evaluated across ordered soil categories, with arid soil as the reference. The columns denote soil categories and the significant genera identified by ANCOM-BC2 pattern analysis are provided in the rows. Each cell color represents abundance change: blue indicates reduction and red signifies increase. The log fold-changes relative to the reference group (arid group) are noted in each cell. The Holm–Bonferroni method was used for multiple testing correction. Genera represented in black are significant without a multiple testing correction, whereas those highlighted in green are significant after multiple testing correction. Additionally, genera marked with an asterisk are also significant after applying the ANCOM-BC2 (SS filter).

power among all methods while having inflated mdFDR across all simulation scenarios. Notably, LMM-CLR's rate of mdFDR rise was the most rapid with increasing sample size relative to the other methodologies.

## Additional simulation studies

In addition to the URT data, we also analyzed a subset from the Quantitative Microbiome Project[23], comprising 106 samples and 91 OTUs. The findings paralleled those from the URT dataset (Extended Data Figs. 3–5).

## Soil microbiome and aridity

Recently, Neilson et al.[24] investigated the differences in soil microbiomes according to soil aridity in the Atacama Desert in Chile. They classified soil samples into three ordered categories based on aridity, namely, arid, margin and hyper-arid, and sequenced data from 63 sample pits from 18 sites in the desert. Since they did not perform DA analyses of those data, we reanalyzed those data using the ANCOM-BC2 methodology. To begin with, we conducted a pattern analysis of richness with respect to the ordered aridity categories (arid to hyper-arid) (Fig. 4a). Using a constrained inference-based trend test[7], executed using ORIOGEN[25] with 10,000 bootstraps, we discovered a significant loss of richness with the increase in aridity ($P = 0.0001$). This finding is consistent with Neilson et al.[24].

Next, we conducted a pattern analysis using ANCOM-BC2 (no filter) to identify trends in microbial abundance across the ordered soil categories, with arid soil serving as the reference group. Significant genera are presented in Fig. 4b. Genera in green were determined to be significant after adjusting for multiple testing. Additionally, genera denoted by an asterisk were also identified as significant when the conservative ANCOM-BC2 (SS filter) was applied. *Blastococcus*, *Rubrobacter* and *Thermobaculum* increased in mean absolute abundance with soil aridity ($P < 0.05$). The trend in *Blastococcus* was significant even after adjusting for multiple testing (adjusted $P < 0.05$) (Fig. 4b). *Thermobaculum* is known for its thermophilic properties, with some species thriving in temperatures up to 90 °C (ref. 26). It has also been documented to possess antimicrobial-resistant genes[27,28]. Similarly, the two Actinobacteria genera, *Blastococcus* and *Rubrobacter*, are also known for their antibacterial resistance[29,30]. Thus, using ANCOM-BC2, we discovered genera that increased in abundance with aridity and may be antibacterial-resistant.

Elevated aridity in desert ecosystems has profound implications on soil health. For instance, increasing aridity in desert soils has been found to significantly diminish nitrogen-cycling microbes. Notable among the affected microbial taxa are *Nitrobacter*, a common contributor to nitrification, and potential widespread nitrogen fixers such as *Sinorhizobium*, *Rhizobium* and *Azospirillum*. These taxa were not detected in samples obtained from hyper-arid environments based on the results of the presence and absence test (Supplementary Table 1). In agreement with these findings, the ANCOM-BC2 (no filter) pattern analysis also revealed that increasing aridity correlates with significant reductions in beneficiary genera (Fig. 4b). The ANCOM-BC2 trend analysis revealed a significant decrease in the mean absolute abundance of *Jiangella*, *Kaistobacter*, *Planctomyces* and *Pseudonocardia* in relation to soil aridity ($P < 0.05$). Among them, *Kaistobacter* and *Pseudonocardia* remained significant after adjusting for multiple testing, and the result for *Pseudonocardia* did not change when the conservative ANCOM-BC2 (SS filter) was used. *Pseudonocardia* has been recognized for its nitrogen-fixing properties[31] and its significance to biotechnology stems from its ability to synthesize secondary metabolites with antibacterial, antifungal and antitumor properties[32]. Likewise, *Kaistobacter*

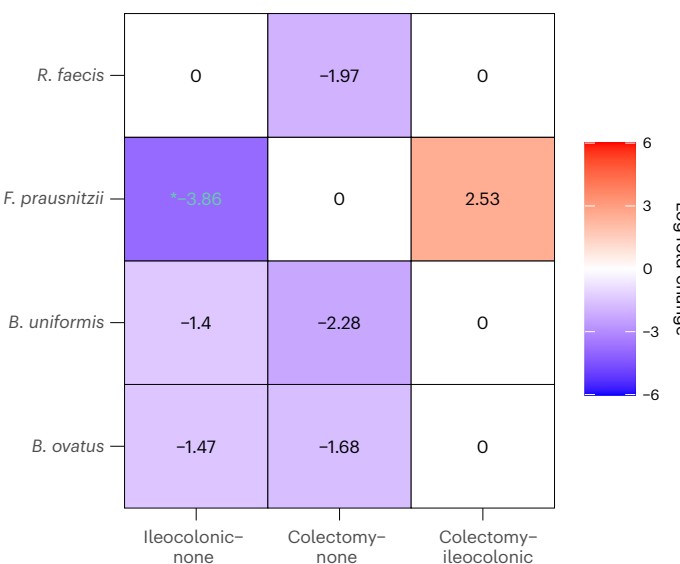

**Fig. 5 | Heatmap of ANCOM-BC2 (no filter) pairwise analysis evaluating the impact of surgical resection on microbial species.** In a cohort of patients with IBD[35], the analysis entailed multiple pairwise comparisons among three distinct groups: ileocolonic resection, colectomy and no intestinal surgery, while maintaining an overall mdFDR at 0.05. The columns denote the specific comparisons: ileocolonic resection versus no intestinal surgery, colectomy versus no intestinal surgery and ileocolonic resection versus colectomy. The rows list significant species as identified by ANCOM-BC2. Each cell is color-coded to represent significant changes in absolute abundance: blue represents reduced abundance and red indicates increased abundance. Multiple testing corrections were performed using the Holm–Bonferroni method. The text within each cell represents the log fold-change value. The log fold-change values displayed in black represent significant changes without adjustment for mdFDR, whereas those in green are significant after applying mdFDR control. Furthermore, values with an asterisk are significant following the application of the ANCOM-BC2 (SS filter).

is known to foster homeostasis within soil microbial communities and acts as a suppressor of soil-borne pathogens[33]. Moreover, *Jiangella*, a halotolerant actinobacterium, is distinguished by its association with nitrate solution, sulfonate transport systems, nitrite reductase and nitrogen fixation[34].

**Gut microbial composition of patients with IBD**

We illustrate ANCOM-BC2 using a longitudinal inflammatory bowel disease (IBD) dataset obtained from Fang et al.[35] to investigate the changes in the gut microbiome following gastrointestinal surgery in patients with IBD. The data in this study are based on 322 stool samples collected from 125 patients. Of these, 46 patients were diagnosed with ulcerative colitis and 79 with Crohn's disease. Stool samples were obtained from each participant at approximately 6-month intervals, beginning at the baseline time point. Specifically, 21 patients provided one sample, 38 patients provided two samples, 41 patients provided three samples, 23 patients provided four samples and two patients provided five samples. Of the total patient population, 87 (70.0%) had no history of intestinal surgery, while 22 patients with Crohn's disease had undergone ileocolonic resection and 13 patients with Crohn's disease and three patients with ulcerative colitis had undergone different types of colectomy. These surgeries occurred before the collection of the baseline stool sample. For the purposes of this study, we focused on comparing the microbial compositions between patients who had not undergone gastrointestinal surgery, those who had undergone ileocolonic resection and those who had undergone colectomies. We adjusted the ANCOM-BC2 model for IBD disease type (ulcerative colitis versus Crohn's disease) and two potential

confounders, namely disease state (inactive versus active) and antibiotic use (absent versus present).

We performed multiple pairwise comparisons among the three groups controlling the overall mdFDR at 0.05 using ANCOM-BC2 (no filter). The results are depicted in Fig. 5. The log fold-changes emphasized in green are significant after adjusting for mdFDR. Further, changes marked with an asterisk were also significant by ANCOM-BC2 (SS filter) method. Ileocolonic section is the surgical removal of the diseased section of the ileum, which is the junction area between the small and last intestines. By contrast, colectomy is the surgical removal of most or all of the large intestine. Our analysis revealed that almost no microbial species were differentially abundant between the two surgical groups of patients, except for *F. prausnitzii*, which is more abundant in the colectomy group.

We observed marked reductions in the absolute abundance of several commensal gut bacterial species in patients who had undergone either ileocolonic resection or colectomy, in comparison to patients without any history of intestinal surgery. The affected species included *Bacteroides* spp. (*ovatus* and *uniformis*), *Faecalibacterium prausnitzii* and *Roseburia faecis*. Of particular note is the significant decrease in *Faecalibacterium prausnitzii* in patients subjected to ileocolonic resection. This reduction remained noteworthy even after using the conservative ANCOM-BC2 (SS filter) together with multiple testing corrections. A crucial aspect to consider is that most of these bacterial species are intrinsically involved in the production of short-chain fatty acids such as acetate, propionate and butyrate[36–42]. These short-chain fatty acids are essential for maintaining gut health, bolstering gut barrier function, exhibiting anti-inflammatory properties and serving as energy sources for colonocytes. Thus, the surgical intervention on these patients, which was necessary, may have unintended effects on the host's immune response and overall health due to the reduction of some important gut microbiota.

## Discussion

In this article, we introduced a general framework called ANCOM-BC2 for performing DA analysis when the exposure variable is continuous, binary or (ordered) categorical. The proposed methodology allows for adjusting for covariates and repeated measures (longitudinal measures) while controlling for FDR, or mdFDR when the exposure variable has more than two groups and the researcher is interested in inferring whether the absolute abundance of a taxon increased or decreased within each pairwise comparison. Furthermore, using the theory of constrained statistical inference, ANCOM-BC2 allows researchers to infer patterns in microbial absolute abundance over ordered categories of exposure variables. For example, it allows a researcher to test whether a particular microbe increased (or decreased) in absolute abundance over ordered disease categories (very healthy to least healthy). This is a unique feature of ANCOM-BC2.

Driven by observed shortcomings of ANCOM-BC in specific edge cases, highlighted in our work and recent literature, we tailored our simulation study to evaluate ANCOM-BC2's performance in these scenarios as well. The results of our simulation study demonstrate that ANCOM-BC2 provides a better FDR control over competing methods tested here while maintaining high power. In particular, ANCOM-BC2 (SS filter) consistently controlled the FDR or mdFDR below the nominal level in all simulation settings considered in this Article while maintaining high power. By contrast, ANCOM-BC2 (no filter) emerged as the DA method with the highest power, displaying a smaller FDR or mdFDR when compared with competing methods other than ANCOM-BC2 (SS filter). According to the FAP score introduced in this Article, ANCOM-BC2 (SS filter) and ANCOM-BC2 (no filter) had stochastically larger FAP scores than competitors with ANCOM-BC2 (SS filter) having the highest score. In terms of practical application, we endorse the use of ANCOM-BC2 (no filter) for small to moderate sample sizes (for example, $n \leq 50$) when repeated measurements are absent. For larger

sample sizes (for example, $n > 50$) or in cases of repeated measures, ANCOM-BC2 (SS filter) is recommended due to its superior FDR control. In pattern analyses, both ANCOM-BC2 (no filter) and ANCOM-BC2 (SS filter) perform equally well in terms of FDR control within the nominal level, although ANCOM-BC2 (no filter) demonstrates a marginally superior power.

The power of ANCOM-BC2's pattern analysis was demonstrated in the soil microbiome data analyzed in this Article. When standard pairwise analyses were performed, only *Pseudonocardia* was differentially abundant across different groups (data not shown). However, using the pattern analysis, we discovered several taxa display increasing or decreasing trends over the ordered soil aridity groups. This is because, unlike pairwise comparisons, pattern analysis uses constrained inference methods, which 'borrow' information from ordered groups, thus increasing the effective sample size and the power[7,43,44].

The ileocolonic section and colectomy are procedures that surgically remove different regions of the intestines, and yet based on our analysis of the IBD data, there were no significant differences in the absolute abundance of most of the gut bacteria in these two groups. Furthermore, the two groups of patients have similarly reduced absolute abundances of certain bacteria relative to those who did not undergo either of the two surgeries. Based on these findings, it may be reasonable to hypothesize that most species of gut microbiota are spatially uniformly distributed in the ileum and large intestines.

## Online content

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

## Methods

### Notation

Notations used in the ANCOM-BC2 methodology are summarized in Table 1. The overall procedure of the ANCOM-BC2 methodology is summarized in Extended Data Fig. 6.

### ANCOM-BC2 for fixed-effects models

**Model assumptions. Assumption 1.** Multiplicative model for observed counts:

$$O_{ij} = S_i C_j A_{ij} E_{ij}.$$

Assumption 1 indicates that, in expectation, the observed counts of a taxon in a random sample is in constant proportion to the true absolute abundance in a unit volume of the ecosystem of the sample. This proportion can be decomposed into two parts: (1) sample-specific sampling fraction and (2) taxon-specific sequencing efficiency.

According to Assumption 1, for nonzero observed count, the above multiplicative model can be transformed into an additive model by log transformation

$$o_{ij} = s_i + c_j + a_{ij} + e_{ij}^{(o)}.$$

**Assumption 2.** Linear model for log true absolute abundances: for each taxon $j$, $a_{ij}$, $i = 1, \ldots, n$ are independently distributed, and

$$a_{ij} = \mathbf{b}_j^T \mathbf{x}_i + e_{ij}^{(a)},$$

where

(1)  $\mathbf{x}_i = (1, x_{i1}, x_{i2}, \ldots, x_{ip})^T$ are the covariates of interest (including the intercept) for the $i$th sample,
(2)  $\mathbf{b}_j = (b_{j0}, b_{j1}, b_{j2}, \ldots, b_{jp})^T$ are the corresponding coefficients for $x_i$.
(3)  $e_{ij}^{(a)}$, $i = 1, \ldots, n$ are independently distributed random errors for log true absolute abundances with $E(e_{ij}^{(a)}) = 0$, $\mathrm{Var}(e_{ij}^{(a)}) = \sigma_{jj}^{(a)}$.

**Assumption 3.** (Independent random error for log observed counts): assume there are random errors, $e_{ij}^{(o)}$, $i = 1, \ldots, n, j = 1, \ldots, d$ , for log observed counts $o_{ij}$, which are independently distributed with heteroskedasticity:

$$E(e_{ij}^{(o)}) = 0, \ \mathrm{Var}(e_{ij}^{(o)}) = \sigma_{ij}^{(o)}, \ e_{ij}^{(o)} \perp\!\!\!\perp e_{ij}^{(a)}.$$

**Regression framework.** Based on the Assumptions 2 and 3, $o_{ij}$ can be modeled as:

$$o_{ij} = s_i + c_j + \mathbf{b}_j^T \mathbf{x}_i + e_{ij}^{(a)} + e_{ij}^{(o)} := s_i + c_j + \mathbf{b}_j^T \mathbf{x}_i + e_{ij}, \quad (1)$$

with

$$E(o_{ij}) = s_i + c_j + \mathbf{b}_j^T \mathbf{x}_i, \ \mathrm{Var}(o_{ij}) = \mathrm{Var}(e_{ij}) = \sigma_{jj}^{(a)} + \sigma_{ij}^{(o)} := \sigma_{ij}^{(t)}.$$

where $\sigma_{ij}^{(t)}$ denotes the total variance.

Equation (1) can also be written in a vector notation as follows:

$$\mathbf{o}_j = \mathbf{s} + c_j \mathbf{1} + X\mathbf{b}_j + \mathbf{e}_j, \quad (2)$$

with

$$E(\mathbf{e}_j) = (0, \ldots, 0)^T,$$
$$E(\mathbf{o}_j) = \mathbf{s} + c_j \mathbf{1} + X\mathbf{b}_j,$$
$$\mathrm{Cov}(\mathbf{o}_j) = \begin{bmatrix} \sigma_{1j}^{(t)} & 0 & \ldots & 0 \\ 0 & \sigma_{2j}^{(t)} & \ldots & 0 \\ \vdots & \vdots & \ddots & \vdots \\ 0 & 0 & \ldots & \sigma_{nj}^{(t)} \end{bmatrix}.$$

### Table 1 | Summary of notation

| Notation | Description |
|---|---|
| $i$ | Sample index, $i=1, 2, \ldots, n$. |
| $j$ | Taxon index, $j=1, 2, \ldots, d$. |
| $k$ | Index of fixed effects, $k=1, 2, \ldots, p$. |
| $l$ | Index of random effects, $l=1, 2, \ldots, q$. |
| $x_{ik}$ | The $k$th fixed effect of interest for the $i$th sample. |
| $z_{il}$ | The $l$th random effect of interest for the $i$th sample. |
| $A_{ij}$ [b] | True absolute abundance of $j$th taxon in a unit volume of ecosystem of $i$th sample. |
| $O_{ij}$ [b] | Observed count of $j$th taxon in a random specimen taken from a unit volume of ecosystem of $i$th sample. |
| $E_{ij}$ [b] | Random error for taxon $j$ in sample $i$. |
| $S_i$ [a] | Sample-specific sampling fraction. |
| $C_j$ [a] | Taxon-specific sequencing efficiency. |
| $a_{ij}$ [b] | $\log A_{ij}$. |
| $o_{ij}$ [b] | $\log O_{ij}$. |
| $e_{ij}$ [b] | Random error for taxon $j$ in sample $i$ in log scale. |
| $s_i$ [a] | Sample-specific sampling fraction in log scale. |
| $c_j$ [a] | Taxon-specific sequencing efficiency in log scale. |

[a]Parameter. [b]Random variable.

where

(1)  $\mathbf{1} = (1, 1, \ldots, 1)^T$,
(2)  $\mathbf{o}_j = (o_{1j}, o_{2j}, \ldots, o_{nj})^T$,
(3)  $\mathbf{s} = (s_1, s_2, \ldots, s_n)^T$,
(4)  $\mathbf{b}_j = (b_{j0}, b_{j1}, b_{j2}, \ldots, b_{jp})^T$,
(5)  $\mathbf{e}_j = (e_{1j}, e_{2j}, \ldots, e_{nj})^T$,
(6)  $X = \begin{bmatrix} 1 & x_{11} & x_{12} & \ldots & x_{1p} \\ 1 & x_{21} & x_{22} & \ldots & x_{2p} \\ \vdots & \vdots & \vdots & \ddots & \vdots \\ 1 & x_{n1} & x_{n2} & \ldots & x_{np} \end{bmatrix}.$

It is important to note that within each sample $i$, for taxa $l \neq m$, $o_{il}$ and $o_{im}$ are not necessarily independent due to correlations between $a_{il}$ and $a_{im}$. Thus vectors $\mathbf{o}_l$ and $\mathbf{o}_m$ are not independent random vectors.

### Remove the effect of taxon-specific sequencing efficiency

To eliminate the effect of $c_j$, we first center the log observed counts across samples, that is

$$y_{ij} := o_{ij} - \bar{o}_{\cdot j} = (s_i - \bar{s}) + \mathbf{b}_j^T(\mathbf{x}_i - \bar{\mathbf{x}}) + (e_{ij} - \bar{e}_{\cdot j}),$$
$$:= \theta_i + \boldsymbol{\beta}_j^T \mathbf{x}_i + \epsilon_{ij}, \quad (3)$$

where

(1)  $\beta_{jk} = b_{jk}$ for $k = 1, \ldots, p$, and $\beta_{j0} = \mathbf{b}_j^T \bar{\mathbf{x}}$,
(2)  $\mathrm{Var}(\epsilon_{ij}) = \frac{(n-1)^2}{n^2} \sigma_{ij}^{(t)} + \frac{1}{n^2} \sum_{i' \neq i} \sigma_{i'j}^{(t)} := \sigma_{ij}$.

### Estimation of sample-specific bias

As can be seen from equation (3), $\boldsymbol{\beta}_j$ are not identifiable without determining the nuisance parameter $\theta_i$. We define bias-corrected log absolute abundance $y_{ij}^{(\mathrm{crt})} = y_{ij} - \theta_i$, then the ordinary least squares estimators of $\theta_i$ and $\boldsymbol{\beta}_j$ can be obtained by iteratively solving the following equations. For ease of exposition, the algorithm is described in the vector form, that is $\mathbf{y}_j = (y_{1j}, y_{2j}, \ldots, y_{nj})^T$, $\boldsymbol{\theta} = (\theta_1, \theta_2, \ldots, \theta_n)^T$ and so on.

**Algorithm 1.** Iterative maximum likelihood estimation

> **Initialize:**
>> For $j = 1, ..., d$
>>
>> $\boldsymbol{\theta} \leftarrow \mathbf{0}$
>>
>> $\mathbf{y}_j^{(\text{crt})} \leftarrow \mathbf{y}_j - \boldsymbol{\theta} = \mathbf{y}_j$
>>
>> $\boldsymbol{\beta}_j \leftarrow (X^T X)^{-1} X^T \mathbf{y}_j^{(\text{crt})} = (X^T X)^{-1} X^T \mathbf{y}_j$
>
> **While** not converge **do**
>> $\boldsymbol{\theta} \leftarrow \frac{1}{d} \sum_{j=1}^{d} (\mathbf{y}_j - X\boldsymbol{\beta}_j)$
>>
>> $\mathbf{y}_j^{(\text{crt})} \leftarrow \mathbf{y}_j - \boldsymbol{\theta}$
>>
>> $\boldsymbol{\beta}_j \leftarrow (X^T X)^{-1} X^T \mathbf{y}_j^{(\text{crt})}$
>
> **end while**
> On convergence,

$$\boldsymbol{\theta}^* = \frac{1}{d} \sum_{j=1}^{d} (\mathbf{y}_j - X\boldsymbol{\beta}_j^*), \; \mathbf{y}_j^{(\text{crt})*} = \mathbf{y}_j - \boldsymbol{\theta}^*, \; \boldsymbol{\beta}_j^* = (X^T X)^{-1} X^T \mathbf{y}_j^{(\text{crt})*}. \quad (4)$$

Therefore

$$
\begin{aligned}
\boldsymbol{\theta}^* &= \frac{1}{d} \sum_{j=1}^{d} (\mathbf{y}_j - X\boldsymbol{\beta}_j^*) = \frac{1}{d} \sum_{j=1}^{d} (\mathbf{y}_j - P\mathbf{y}_j^{(\text{crt})*}) \\
&= \frac{1}{d} \sum_{j=1}^{d} (\mathbf{y}_j - P\mathbf{y}_j + P\boldsymbol{\theta}^*) = \frac{1}{d} \sum_{j=1}^{d} \left[ \mathbf{y}_j^{(\text{crt})} + \boldsymbol{\theta} - P(\mathbf{y}_j^{(\text{crt})} + \boldsymbol{\theta}) + P\boldsymbol{\theta}^* \right] \\
&= (I - P)\boldsymbol{\theta} + P\boldsymbol{\theta}^* + \frac{1}{d} \sum_{j=1}^{d} (I - P)\mathbf{y}_j^{(\text{crt})} \\
&= (I - P)\boldsymbol{\theta} + P\boldsymbol{\theta}^* + \frac{1}{d} \sum_{j=1}^{d} \boldsymbol{\epsilon}_j,
\end{aligned} \quad (5)
$$

where

(1) $P = X(X^T X)^{-1} X^T$ is the projection matrix onto $\mathscr{C}(X)$, the column space of $X$,

(2) $\boldsymbol{\epsilon}_j = (I - P)\mathbf{y}_j^{(\text{crt})}$ with $E(\boldsymbol{\epsilon}_j) = \mathbf{0}$.

Rearranging equation (5), we see that

$$(I - P)\boldsymbol{\theta}^* = (I - P)\boldsymbol{\theta} + \frac{1}{d} \sum_{j=1}^{d} \boldsymbol{\epsilon}_j.$$

Taking expectations on both sides leads to

$$(I - P)[E(\boldsymbol{\theta}^*) - \boldsymbol{\theta}] = \mathbf{0}.$$

As $I - P$ is an orthogonal projector onto $\mathscr{C}(X)$, the above equation holds as long as either of the following is valid:

(1) $E(\boldsymbol{\theta}^*) - \boldsymbol{\theta} = \mathbf{0}$,

(2) $E(\boldsymbol{\theta}^*) - \boldsymbol{\theta} \in \mathscr{C}(X)$.

It is sufficient to consider (2) because (1) is the trivial case. If (1) were true then from (4) we deduce that there is no sample-specific effect and that $E(\boldsymbol{\beta}_j^*) = \boldsymbol{\beta}_j$. Suppose (2) is true, then there exists a vector $\boldsymbol{\delta} \neq 0 \in \mathbb{R}^p$, such that

$$E(\boldsymbol{\theta}^*) = \boldsymbol{\theta} - X\boldsymbol{\delta}. \quad (6)$$

Then by combining with equation (4), we have

$$E(\boldsymbol{\beta}_j^*) = \boldsymbol{\delta} + \boldsymbol{\beta}_j. \quad (7)$$

We shall denote $\boldsymbol{\theta}^*$ and $\boldsymbol{\beta}_j^*$ obtained from the above iterative algorithm as preliminary estimators of $\boldsymbol{\theta}$ and $\boldsymbol{\beta}_j$, respectively. Without loss of generality, throughout this Article we assume $X^T X$ is a full rank matrix. If it is not a full rank matrix, then we may use any generalized inverse

of $X^T X$ because $X\boldsymbol{\beta}_j^*$ in equation (5) is invariant of the choice of generalized inverse $(X^T X)^g$ used in $\boldsymbol{\beta}_j^* = (X^T X)^g X^T \mathbf{y}_j^{(\text{crt})}$. Thus the preliminary estimator $\boldsymbol{\theta}^*$ provided above is invariant of the choice of generalized inverse used in deriving $\boldsymbol{\beta}_j^*$. Furthermore, throughout this Article, we are interested in testing a hypothesis regarding linearly estimable parameters $A\boldsymbol{\beta}_j$, that is $\mathscr{C}(A^T) \subset \mathscr{C}(X^T)$ (ref. 45). Consequently, the estimator $A\boldsymbol{\beta}_j^*$ is invariant of the generalized inverse used in the estimation of $\boldsymbol{\beta}_j^*$. Hence, throughout this text, for simplicity of exposition, we shall assume $X^T X$ is of full rank.

For each taxon $j = 1, ..., d$, by equation (7), $\boldsymbol{\beta}_j^*$ is a biased estimator if $\boldsymbol{\delta} \neq 0$. Suppose we wish to test the following hypothesis

$$H_0 : A\boldsymbol{\beta}_j = A\boldsymbol{\beta}_j^0,$$

$$H_1 : A\boldsymbol{\beta}_j \neq A\boldsymbol{\beta}_j^0.$$

Under the null hypothesis, $E(A\boldsymbol{\beta}_j^*) - A\boldsymbol{\beta}_j^0 = A\boldsymbol{\delta} \neq \mathbf{0}$ and hence biased. The next step is to estimate this bias $\boldsymbol{\delta}$ and accordingly modify the estimator $A\boldsymbol{\beta}_j^*$ so that the resulting estimator is asymptotically centered at $A\boldsymbol{\beta}_j^0$ under the null hypothesis and hence the test statistic is asymptotically centered at zero.

First we make the following observations. As $E(\boldsymbol{\beta}_j^*) = \boldsymbol{\delta} + \boldsymbol{\beta}_j$, we note that as $n \to \infty$, for finite dimension $d$,

$$\Sigma_j^{-\frac{1}{2}} (\boldsymbol{\beta}_j^* - (\boldsymbol{\delta} + \boldsymbol{\beta}_j)) \to_d N_P(\mathbf{0}, I), \quad (8)$$

where

$$\Sigma_j = \lim_{n \to \infty} (X^T X)^{-1} (\sum_{i=1}^{n} \sigma_{ij}^2 x_i x_i^T)(X^T X)^{-1}. \quad (9)$$

As

$$E(\boldsymbol{\theta}^* + X\boldsymbol{\beta}_j^*) = \boldsymbol{\theta} - X\boldsymbol{\delta} + X(\boldsymbol{\delta} + \boldsymbol{\beta}_j) = \boldsymbol{\theta} + X\boldsymbol{\beta}_j,$$

that is $\boldsymbol{\theta}^* + X\boldsymbol{\beta}_j^*$ is an unbiased estimator of $\boldsymbol{\theta} + X\boldsymbol{\beta}_j$, hence a possible estimator of $\Sigma_j$ is given by

$$\hat{\Sigma}_j = (X^T X)^{-1} \left( \sum_{i=1}^{n} (y_{ij} - \theta_i^* - \boldsymbol{\beta}_j^{*T} x_i)^2 x_i x_i^T \right)(X^T X)^{-1}. \quad (10)$$

Under some mild regularity conditions[46], with finite $d$, we have the following consistency result

$$n(\hat{\Sigma}_j - \Sigma_j) \to_P 0, \text{ as } n \to \infty. \quad (11)$$

Therefore, replacing $\Sigma_j$ with $\hat{\Sigma}_j$ in equation (8) and appealing to Slutsky's theorem, we have

$$\hat{\Sigma}_j^{-\frac{1}{2}} (\boldsymbol{\beta}_j^* - (\boldsymbol{\delta} + \boldsymbol{\beta}_j)) \to_d N_P(\mathbf{0}, I), \text{ as } n \to \infty.$$

By equations (9) and (11), under some mild regularity conditions, for finite $d$, we obtain

$$\hat{\Sigma}_j \to_p 0, \text{ as } n \to \infty.$$

Consequently,

$$\boldsymbol{\beta}_j^* \to_P \boldsymbol{\delta} + \boldsymbol{\beta}_j, \text{ as } n \to \infty. \quad (12)$$

The above observation regarding the convergence of $\boldsymbol{\beta}_j^*$ plays a critical role in the following. Since the sampling fraction is constant for all taxa within a sample, we pool information across taxa within each sample when estimating $\boldsymbol{\delta}$. We model each taxon abundance using

the following Gaussian mixture model. For the $j$th taxon and the $k$th covariate, let $C_0$ denote the set of taxa that are not differentially abundant with respect to $x_{ik}$, that is, $C_0 = \{j \in (1, 2, \ldots, d): \beta_{jk} = 0\}$; let $C_1$ denote the set of taxa whose abundance decreases with $x_{ik}$, that is, $C_1 = \{j \in (1, 2, \ldots, d): \beta_{jk} < 0\}$, and let $C_2$ denote the set of taxa whose abundance increases with $x_{ik}$, that is, $C_2 = \{j \in (1, 2, \ldots, d): \beta_{jk} > 0\}$. Let $\pi_r$ denote the probability that a taxon belongs to set $C_r$, $r = 0, 1, 2$. For simplicity of estimation of parameters, similar to generalized estimating equations, we shall assume that $\beta_{jk}^*, j = 1, 2, \ldots, d$, are independently distributed. As commonly done in the analyses of various omics data, we ignore the underlying correlation structure when estimating $\boldsymbol{\delta}$. Thus, we model the distribution of $\beta_{jk}^*$ by Gaussian mixture model as follows:

$$f(\beta_{jk}^*) = \pi_0\phi\left(\frac{\beta_{jk}^* - \delta_k}{v_{j0}}\right) + \pi_1\phi\left(\frac{\beta_{jk}^* - (\delta_k + l_1)}{v_{j1}}\right) + \pi_2\phi\left(\frac{\beta_{jk}^* - (\delta_k + l_2)}{v_{j2}}\right),$$

(13)

where

(1)  $\phi$ is the standard normal density function,
(2)  $\delta_k, \delta_k + l_1$ and $\delta_k + l_2$ are means for $\beta_{jk}^*|C_0, \beta_{jk}^*|C_1$ and $\beta_{jk}^*|C_2$, respectively. $l_1 < 0, l_2 > 0$,
(3)  $v_{j0}, v_{j1}$ and $v_{j2}$ are variances of $\beta_{jk}^*|C_0, \beta_{jk}^*|C_1$ and $\beta_{jk}^*|C_2$, respectively.

Note that instead of fitting a multivariate Gaussian mixture model for all covariates together, we choose to fit a univariate Gaussian mixture model repeatedly for every single covariate. This repetition is simply because the sets of taxa $\{C_0, C_1, C_2\}$ are not necessarily the same for different covariates. Also, note that for a categorical covariate of $s + 1$ levels, this contains $s$ coefficients, for example $\beta_{j1}, \ldots, \beta_{js}$, and we shall fit the Gaussian mixture model for these $s$ coefficients separately.

For computational simplicity, we assume that $v_{j1} > v_{j0}, v_{j2} > v_{j0}$. Thus, without loss of generality for $\kappa_1, \kappa_2 > 0$, let $v_{j1} = v_{j0} + \kappa_1$ and $v_{j2} = v_{j0} + \kappa_2$. While this assumption is not a requirement for our method, it is reasonable to assume that variability among differentially abundant taxa is larger than that among the null taxa. By making this assumption, we simplify the computation.

Assuming samples are independent, we begin by first estimating $v_{j0}^2 = \text{Var}(\beta_{jk}^*)$. Note that $v_{j0}^2$ is the function of heteroscedastic variances, a consistent estimator of $v_{j0}^2$, which we refer to as $\hat{v}_{j0}^2$, is the $k$th diagonal element of $\hat{\Sigma}_j$ stated in equation (10). In all future calculations, we plug in $\hat{v}_{j0}^2$ for $v_{j0}^2$. This is similar in spirit to many statistical procedures involving nuisance parameters. The following lemma[47] is useful in the sequel.

**Lemma 1.** *Introducing the latent variable in calculating log-likelihood*:

$$\log f(x|\theta) = E_{f(z|x,\theta)}[\log f(z|\theta) + \log f(x|z, \theta)].$$

Let $\Theta = (\delta_k, \pi_1, \pi_2, \pi_3, l_1, l_2, \kappa_1, \kappa_2)^T$ denote the set of unknown parameters, then for each taxon the log-likelihood can be reformulated using Lemma 1, as follows:

$$\Theta \leftarrow \arg\max_{\Theta} \sum_{j=1}^{d} \sum_{r=0}^{2} P_{r,j}[\log \Pr(j \in C_r) + \log f(\beta_{jk}|j \in C_r)]. \quad (14)$$

Then the EM algorithm is described as follows:

- E step: compute conditional probabilities of latent variables.

  Define $P_{r,j} = \Pr(j \in C_r|\beta_{jk}, \Theta) = \dfrac{\pi_r\phi\left(\frac{\beta_{jk} - (\delta_k + l_r)}{v_{jr}}\right)}{\sum_r \pi_r\phi\left(\frac{\beta_{jk} - (\delta_k + l_r)}{v_{jr}}\right)}, r = 0, 1, 2; j = 1, \ldots, d,$

  which are conditional probabilities representing the probability that an observed value follows each distribution. Note that $l_0 = 0$.
- M step: maximize the likelihood function with respect to the parameters, given the conditional probabilities.

We shall denote the resulting estimator of $\delta_k$ on convergence of the algorithm by $\hat{\delta}_k^{EM}$.

As stated in Lin and Peddada[3], compared to $\hat{v}_{j0}^2$, the variance and covariance contributed by $\hat{\delta}_k^{EM}$ is negligible when the number of nondifferentially abundant taxa is large, such as when analyzing the microbiome data at the OTU, amplicon sequence variant (ASV) or species level of the phylogenetic tree.

The above procedure is applied to every $\beta_{jk}, k = 1, \ldots, p$, eventually, we obtain the estimator of $\boldsymbol{\delta}$ as

$$\hat{\boldsymbol{\delta}}^{EM} = (\hat{\delta}_1^{EM}, \hat{\delta}_2^{EM}, \ldots, \hat{\delta}_P^{EM})^T. \quad (15)$$

Therefore, the final estimator of $\boldsymbol{\beta}_j$ is defined as

$$\hat{\boldsymbol{\beta}}_j = \boldsymbol{\beta}_j^* - \hat{\boldsymbol{\delta}}^{EM}, \quad (16)$$

with

$$\hat{\boldsymbol{\beta}}_j \rightarrow_P \boldsymbol{\beta}_j, \text{ as } n \rightarrow \infty, \quad (17)$$

given that $\hat{\boldsymbol{\delta}}^{EM}$ is a good approximation of $\boldsymbol{\delta}$.

The estimation procedure is summarized in Algorithm 2.

**Algorithm 2.** EM algorithm
(1)  **input:**
    $\boldsymbol{\beta}_j^*, \Sigma_j, j = 1, \ldots, d$
(2)  **procedure** EM ($\boldsymbol{\beta}_j^*, \Sigma_j$)
(3)   **return** $\hat{\delta}_k^{EM}, k = 1, \ldots, P$
(4)  **end procedure**
(5)  **for** $k = 1, \ldots, p$ **do**
(6)   $\hat{\beta}_{jk} \leftarrow \beta_{jk}^* - \hat{\delta}_k^{EM}$
(7)  **end for**

For taxon $j$, we now describe our methodology for testing the following hypotheses

$$H_0 : A\boldsymbol{\beta}_j = A\boldsymbol{\beta}_j^0,$$

$$H_1 : A\boldsymbol{\beta}_j \neq A\boldsymbol{\beta}_j^0.$$

From Slutsky's theorem, as $n \rightarrow \infty$, the following test statistic is approximately central chi-square distributed under the null hypothesis

$$W_j = (A\hat{\boldsymbol{\beta}}_j - A\boldsymbol{\beta}_j^0)^T (A\hat{\Sigma}_j A^T)^{-1} (A\hat{\boldsymbol{\beta}}_j - A\boldsymbol{\beta}_j^0)$$

$$= (A\boldsymbol{\beta}_j^* - A\hat{\boldsymbol{\delta}}^{EM} - A\boldsymbol{\beta}_j^0)^T (A\hat{\Sigma}_j A^T)^{-1} (A\boldsymbol{\beta}_j^* - A\hat{\boldsymbol{\delta}}^{EM} - A\boldsymbol{\beta}_j^0)$$

$$\rightarrow_d \chi_q^2,$$

where $q = \text{rank}(A)$.

To control the FDR due to multiple testing, we recommend applying Holm−Bonferroni method[20] instead of Benjamini−Hochberg procedure[6] because the Holm−Bonferroni method does not require any assumptions regarding the dependence structure in the underlying $P$ values, and is also known to be a better method to control FDR when $P$ values are not accurate[21].

**Sample-specific biases estimation.** After obtaining $\hat{\boldsymbol{\delta}}^{EM}$, the estimator of sample-specific biases $\boldsymbol{\theta}$ is defined as follows:

$$\hat{\boldsymbol{\theta}} = \frac{1}{d}\sum_{j=1}^{d}(\mathbf{y}_j - X\hat{\boldsymbol{\beta}}_j). \quad (18)$$

Let $\Sigma^{(i)} = [\sigma_{lm}^{(i)}]_{l,m=1,\ldots,d}$ denote the $d \times d$ covariance matrix of $\boldsymbol{\epsilon}^{(i)} = (\epsilon_1^{(i)}, \epsilon_2^{(i)}, \ldots, \epsilon_d^{(i)})^T$, where $\sigma_{lm}^{(i)}$ is the $(l, m)$th element of $\Sigma^{(i)}$ and $\sigma_{jj}^{(i)}$ is the $j$th diagonal element of $\Sigma^{(i)}$. Furthermore, suppose

From Assumption 4, we have

$$0 \leq \mathbf{1}^T \Sigma^{(i)} \mathbf{1} = \sum_{l=1}^{d} \sum_{m=1}^{d} \sigma_{lm}^{(i)} = \sum_{j=1}^{d} \sigma_{jj}^{(i)} + \sum_{l \neq m}^{d} \sigma_{lm}^{(i)} \leq dK + \sum_{l \neq m}^{d} \sigma_{lm}^{(i)}.$$

Hence

$$0 \leq \frac{\mathbf{1}^T \Sigma^{(i)} \mathbf{1}}{d^2} \leq \frac{K}{d} + \frac{\sum_{l \neq m}^{d} \sigma_{lm}^{(i)}}{d^2} = o(1).$$

Thus, for each taxon $j = 1, 2, \ldots, d$, we have

$$\frac{1}{d} \sum_{j=1}^{d} (\mathbf{y}_j - (\boldsymbol{\theta} + X\boldsymbol{\beta}_j)) \rightarrow_P \mathbf{0}, \text{ as } d \rightarrow \infty. \tag{19}$$

Therefore, according to equations (17) and (19), as both $n, d \rightarrow \infty$,

$$\hat{\boldsymbol{\theta}} \rightarrow \boldsymbol{\theta}. \tag{20}$$

**Assumption 4.** Sparse correlations among taxa:

$$\sigma_{jj}^{(i)} < K < \infty,$$

$$\frac{\sum_{l \neq m}^{d} \sigma_{lm}^{(i)}}{d^2} = o(1).$$

**Remark 1.** Regularization of variance: to avoid the spurious detection of significance due to extremely small standard errors, particularly for rare taxa, we incorporated a small positive constant in the denominator of the ANCOM-BC2 test statistic for each taxon. This approach was inspired by the significance analysis of microarray methodology[15]. Specifically, the regularization factor was set as the fifth percentile of the distribution of standard errors for each fixed effect, unless otherwise specified.

**Remark 2.** Sensitivity analysis for the pseudo-count addition: to mitigate the risk of inflated false-positive rates resulting from the choice of pseudo-count in ANCOM-BC2, we conducted a sensitivity analysis to assess the impact of varying pseudo-count values on DA results. This is particularly important, as several studies have shown that the choice of pseudo-count can significantly influence the results of DA analysis methods[16,17]. For details regarding the sensitivity analysis and the definitions of the two version of ANCOM-BC2, refer to the section 'Strategies implemented in ANCOM-BC2 to handle zeros' below.

**Multigroup comparison.** In some applications, for a given taxon, researchers are interested in drawing inferences regarding DA among different pairs of experimental groups. We refer to this kind of problem as a multigroup comparison problem, and extra caution needs to be exercised to correct $P$ values due to multiple comparisons. For simplicity, we drop the subscript $j$ (taxon index) in the following discussions.

## Global test
For a given taxon and a total of $g + 1$ experimental groups (including the reference group), researchers may want to test whether there exists at least one group that is significantly different from others. For ease of exposition, we split the covariates $X$ into two parts, where $X_1$ stands for the group assignment and $X_2$ denotes the remaining covariates. Note that the difference of group effects against the reference group is estimable, while the individual group effect is not. For simplicity, in the discussions of multigroup comparisons among group 0 to group $g$, we assume group 0 is the reference group. We use $\beta_k, k = 1, \ldots, g$ to denote the group effect, but notice that it actually estimates $\beta_k - \beta_0$. We rewrite the model stated in equation (3) as

$$\mathbf{y} = \boldsymbol{\theta} + X_1 \boldsymbol{\beta} + X_2 \boldsymbol{\gamma} + \boldsymbol{\epsilon}, \tag{21}$$

where

(1) $\boldsymbol{\theta}$ is the sample-specific bias,
(2) $\boldsymbol{\beta}$ is the vector of group effects (as compared to group 0) of the order $g \times 1$,
(3) $X_1$ is the design matrix of the order $n \times g$ consisting of 0s and 1s,
(4) $X_2$ is the known matrix of other covariates (including the intercept) of the order $n \times (p - g + 1)$ with the corresponding regression parameter vector $\boldsymbol{\gamma}$ of the order $(p - g + 1) \times 1$.

The global test intends to test

$$H_0 : \cap_{k \in \{1, \ldots, g\}} \beta_k = 0,$$
$$H_1 : \cup_{k \in \{1, \ldots, g\}} \beta_k \neq 0,$$

which can be reformulated as

$$H_0 : A\boldsymbol{\beta} = \mathbf{0},$$
$$H_1 : A\boldsymbol{\beta} \neq \mathbf{0},$$

where

$$A = I_g = \begin{bmatrix} 1 & 0 & 0 & \ldots & 0 \\ 0 & 1 & 0 & \ldots & 0 \\ \vdots & \vdots & \ddots & \vdots & \vdots \\ 0 & 0 & \ldots & 0 & 1 \end{bmatrix}$$

with the test statistic

$$W = (A\hat{\boldsymbol{\beta}})^T (A \hat{\Sigma}^{(g)} A^T)^{-1} (A\hat{\boldsymbol{\beta}}) \rightarrow_d \chi_g^2, \text{ as } n \rightarrow \infty,$$

where $\hat{\Sigma}^{(g)}$ is the corresponding submatrix of $\hat{\Sigma}$ defined in equation (10).

Similarly, to control the FDR due to multiple testing, we recommend applying Holm–Bonferroni method[20] instead of the Benjamini–Hochberg procedure[6] due to the underlying complex dependence structure between taxa.

**Example 1.** Suppose there are three groups, namely, groups 0 (reference), 1 and 2, and no other covariates. For each sample $i$, $i = 1, \ldots, n$, we have:

$$y_i = \theta_i + \mu + \beta_1 I\{\text{group} = 1\} + \beta_2 I\{\text{group} = 2\} + \epsilon_i.$$

To test whether there is at least one group among 0, 1 and 2, that is significantly different from others, we test:

$$H_0 : \beta_1 = \beta_2 = 0,$$
$$H_1 : \beta_1 \neq 0 \cup \beta_2 \neq 0,$$

which is the same as testing:

$$H_0 : A\boldsymbol{\beta} = \mathbf{0},$$
$$H_1 : A\boldsymbol{\beta} \neq \mathbf{0},$$

where $A = \begin{bmatrix} 1 & 0 \\ 0 & 1 \end{bmatrix}$, and $\boldsymbol{\beta} = (\beta_1, \beta_2)^T$.

## Multiple pairwise comparisons
If we are interested in knowing whether the abundance increased or decreased between various pairs of groups, then it amounts to testing the following hypotheses:

$$H_{0,k,k'} : \beta_k = \beta_{k'}$$
$$H_{1,k,k'} : \{\beta_k < \beta_{k'}\} \cup \{\beta_k > \beta_{k'}\},$$

where $k \neq k' \in \{1, \ldots, g\}$. Denote the test statistic for a given pairwise comparison as

$$W_{kk'} = \frac{\hat{\beta}_k - \hat{\beta}_{k'}}{\sqrt{\widehat{\mathrm{Var}}(\hat{\beta}_k) + \widehat{\mathrm{Var}}(\hat{\beta}_{k'})}} \to_d N(0, 1), \text{ as } n \to \infty,$$

where $\widehat{\mathrm{Var}}(\hat{\beta}_k)$, $\widehat{\mathrm{Var}}(\hat{\beta}_{k'})$ are the $k$th and $k'$th diagonal elements of $\hat{\Sigma}^{(g)}$, respectively. Thus, the raw $P$ value for comparing group $k$ and group $k'$ is defined as:

$$P_{kk'} = 2[1 - \phi(|W_{kk'}|)].$$

For comparing with the reference group (group 0), the hypotheses become:

$$H_{0,k} : \beta_k = 0$$
$$H_{1,k} : \{\beta_k < 0\} \cup \{\beta_k > 0\}.$$

We also replace $\hat{\beta}_{k'}$ and $\widehat{\mathrm{Var}}(\hat{\beta}_{k'})$ with 0s in the test statistic.

Note that the null and alternative hypotheses for the global test are denoted as $H_0$ and $H_1$, a Type I error might occur due to wrongly rejecting $H_0$ or correctly rejecting $H_0$ but wrongly rejecting $H_{0,k,k'}$. A directional error might occur due to correctly rejecting $H_0$ but wrong assignment of the direction between $\beta_k$ and $\beta_{k'}$ while correctly rejecting $H_{0,k,k'}$. In this case, we need to control the error rate combining both type I and the directional errors in the FDR framework, which is referred to as mixed directional FDR (mdFDR)[8,9].

**Definition 1.** mdFDR: let $V(j)$ denote the indicator function of at least one type I error or directional error committed, that is

$$V(j) = \begin{cases} 1 & \text{if Type I or directional error occurs,} \\ 0 & \text{otherwise.} \end{cases}$$

Then, mdFDR is defined as the expected proportion of Type I and directional errors among all discovered taxa.

$$\mathrm{mdFDR} = E\left(\frac{\sum_{j=1}^d V(j)}{\max(R, 1)}\right),$$

where $R$ denotes the number of taxa discovered.

To control the mdFDR for all pairwise tests, we adopt the general mdFDR controlling procedure[9], and do the following:

(1) Apply the global test method stated above to obtain the $P$ value for each taxon. We denote these $P$ values as screening $P$ values. Apply the Benjamini–Hochberg procedure to identify taxa that are differentially abundant in at least one pairwise comparison. Let $R$ denote the number of taxa discovered.
(2) For each taxon discovered in step (1), apply any mixed directional family wise error controlling procedure, such as Holm–Bonferroni (default), Hochberg and so on, to the pairwise $P$ values ($P_{kk'}$) at level $R\alpha/d$.
(3) For a given taxon discovered in step 1, if a pairwise hypothesis is rejected in step (2), then we declare $\beta_k < \beta_{k'}$ or $\beta_k > \beta_{k'}$ according to $W_{kk'} < 0$ or more than 0.

It has been proved that under the assumption of independence of $P$ values obtained from the global test, the mdFDR of the above procedure is strongly controlled at level $\alpha$ (ref. 9).

**Example 2.** Suppose there are three groups, namely, groups 0 (reference), 1 and 2, and no other covariates. For each sample $i$, $i = 1, \ldots, n$, we have:

$$y_i = \theta_i + \mu + \beta_1 I\{\mathrm{group} = 1\} + \beta_2 I\{\mathrm{group} = 2\} + \epsilon_i.$$

To test whether the taxon is differentially abundant between group 1 and 0 (reference), we test:

$$H_0 : \beta_1 = 0,$$
$$H_1 : \{\beta_1 < 0\} \cup \{\beta_1 > 0\},$$

with the test statistic:

$$W_{10} = \frac{\hat{\beta}_1}{\sqrt{\widehat{\mathrm{Var}}(\hat{\beta}_1)}}.$$

Additionally, if we want to test whether the taxon is differentially abundant between group 1 and 2:

$$H_0 : \beta_1 = \beta_2,$$
$$H_1 : \{\beta_1 < \beta_2\} \cup \{\beta_1 > \beta_2\}.$$

The test statistic is:

$$W_{12} = \frac{\hat{\beta}_1 - \hat{\beta}_2}{\sqrt{\widehat{\mathrm{Var}}(\hat{\beta}_1) + \widehat{\mathrm{Var}}(\hat{\beta}_2)}}.$$

**Test against a specific group.** Often, researchers are interested in knowing whether the abundance increased or decreased in an ecosystem relative a prespecified group, say the control group. Again, assume group 0 is the reference group and $\beta_0 = 0$, then one may be interested in testing the following hypotheses:

$$H_{0,k} : \beta_k = 0,$$
$$H_{1,k} : \{\beta_k < 0\} \cup \{\beta_k > 0\},$$

where $k \in \{1, \ldots, g\}$.

As before, the pairwise test statistic is defined as follows:

$$W_k = \frac{\hat{\beta}_k}{\sqrt{\widehat{\mathrm{Var}}(\hat{\beta}_k)}} \to_d N(0, 1), \text{ as } n \to \infty,$$

where $\widehat{\mathrm{Var}}(\hat{\beta}_k)$ is the $k$th diagonal elements of $\hat{\Sigma}^{(g)}$. Thus, the raw $P$ value for comparing group $k$ and group 1 is defined as

$$P_k = 2[1 - \phi(|W_k|)].$$

Likewise, we apply the mdFDR controlling procedure for all pairwise tests. To improve power, we modify the global test mentioned earlier to a Dunnet-based test[48–50] as described below:

(1) The test statistic $W = \max_{k \in \{1, \ldots, g\}} |W_k|$,
(2) Generate $W_k^{(b)} \approx N(0, 1), k = 1, \ldots, g$.
(3) Compute $W^{(b)} = \max_{k \in \{1, \ldots, g\}} |W_k^{(b)}|$.
(4) Repeat the above steps $B$ times, we get the null distribution of $W$.

The screening $P$ value is calculated as:

$$P = \frac{1}{B} \sum_{b=1}^B I(W^{(b)} > W).$$

**Pattern analysis.** When the experimental groups are ordered naturally, such as doses of exposure or duration of exposure or stages of a disease and so on, for a given taxon, researchers may be interested in testing

whether the abundance of the taxon is changing with the ordered experimental groups according to some specific pattern. Thus, the null and alternative hypotheses one wants to test become (assume group 0 is the reference group):

$$H_0 : \beta_1 = \beta_2 = \dots = \beta_g = 0,$$

$$H_1 : \boldsymbol{\beta} = (\beta_1, \dots, \beta_g)^T \in \mathbb{C},$$

where $\mathbb{C}$ is one or a collection of patterns. Examples of patterns are given below.

**Example 3.** Simple order

$$\mathbb{C}_1 = \{0 \leq \beta_1 \leq \beta_2 \leq \dots \leq \beta_g\} \text{ with at least one strict inequality.} \quad (22)$$

**Example 4.** Tree order

$$\mathbb{C}_2 = \{\beta_k \geq 0, k = 1, \dots, g\} \text{ with at least one strict inequality.} \quad (23)$$

**Example 5.** Umbrella order

$$\mathbb{C}_4 = \{0 \leq \beta_1 \leq \dots \leq \beta_{k-1} \leq \beta_k \geq \beta_{k+1} \dots \geq \beta_g\}$$
$$\text{with at least one strict inequality.} \quad (24)$$

Estimation of $\boldsymbol{\beta}$ under a certain pattern (constraint) can be obtained by solving the following convex optimization (opt) problem[51]:

$$\hat{\boldsymbol{\beta}}^{\text{opt}} = \arg\min_{\boldsymbol{\beta} \in \mathbb{C}} (\hat{\boldsymbol{\beta}} - \boldsymbol{\beta})^T \hat{\Sigma}^{(g)^{-1}} (\hat{\boldsymbol{\beta}} - \boldsymbol{\beta}), \quad (25)$$

where $\hat{\Sigma}^{(g)}$ is the corresponding submatrix of $\hat{\Sigma}$ defined in equation (10). The solution to equation (25) can be numerically obtained by using a suitable convex optimization algorithm, such as CVXR (ref. 52).

**Example 6.** Suppose there are three groups, namely, groups 0 (reference), 1 and 2, and no other covariates. For each sample $i, i = 1, \dots, n$, we have:

$$y_i = \theta_i + \mu + \beta_1 I\{\text{group} = 1\} + \beta_2 I\{\text{group} = 2\} + \epsilon_i.$$

To test whether the group effect is monotonically increasing, we test:

$$H_0 : \beta_1 = \beta_2 = 0,$$

$$H_1 : \boldsymbol{\beta} \in \mathbb{C} = \{0 \leq \beta_1 \leq \beta_2\}, \text{ with at least one strict inequality.}$$

The estimation of $\beta$ under $\mathbb{C}$ can be obtained by solving:

$$\hat{\boldsymbol{\beta}}^{\text{opt}} = \arg\min_{\boldsymbol{\beta} \in \mathbb{R}^2} (\hat{\boldsymbol{\beta}} - \boldsymbol{\beta})^T \hat{\Sigma}^{(g)^{-1}} (\hat{\boldsymbol{\beta}} - \boldsymbol{\beta}),$$

$$\text{s.t.} A\boldsymbol{\beta} \geq \mathbf{0},$$

where $A = \begin{bmatrix} 1 & 0 \\ -1 & 1 \end{bmatrix}$, and $\boldsymbol{\beta} = (\beta_1, \beta_2)^T$.

Once the constrained estimator is obtained, there exist a variety of options to test the above hypotheses. For example, one may consider William's type of statistic[53]. We adopt the following definitions from Peddada et al.[7] to facilitate the construction of the test statistic.

**Definition 2.** Linked parameters: two parameters in a given pattern are said to be linked if the inequality between them is specified a priori.

**Definition 3.** Nodal parameter: for a given pattern, a parameter is said to be nodal if it is linked with every other parameter in the profile.

For example, every parameter is a nodal parameter in $\mathbb{C}_1$; no nodal parameter in $\mathbb{C}_2$ and $\beta_k$ is the only nodal parameter in $\mathbb{C}_3$.

**Definition 4.** Norm of maximum difference: define the norm $l_\infty(\mathbb{C})$ of pattern $\mathbb{C}$ as the maximum difference between the estimates of two linked parameters.

For example, $l_\infty(\mathbb{C}_3) = \max\{\hat{\beta}_k, \hat{\beta}_k - \hat{\beta}_g\}$.

Given a collection of potential patterns, $\mathbb{C}_1, \mathbb{C}_2, \dots, \mathbb{C}_T$, the William's type of test statistic is defined as:

$$W = \max\{l_\infty(\mathbb{C}_t), t = 1, \dots, T\},$$

$$\text{with } t^{\text{opt}} = \arg\max\{l_\infty(\mathbb{C}_t), t = 1, \dots, T\},$$

where $t^{\text{opt}}$ is regarded as the optimal pattern for the microbial abundance of a specific taxon.

Under null hypothesis, the expectations for $\hat{\beta}_k, k = 1, \dots, g$ are 0s; thus, we can construct the null distribution of $W$ as follows:

(1) Generate $\hat{\beta}_k^{(b)} \approx \sqrt{\widehat{\text{Var}}(\hat{\beta}_k)} N(0, 1), k = 1, \dots, g$.
(2) Obtain constrained regression estimators for $\hat{\beta}_k^{\text{opt},(b)}$ using the convex optimization problem described above.
(3) Compute $W^{(b)} = \max\{l_\infty(\mathbb{C}_t), t = 1, \dots, T\}$ using the simulated data under prespecified patterns.
(4) Repeat the above steps $B$ times, and we get the null distribution of $W$.

The raw $P$ value is calculated as

$$P = \frac{1}{B} \sum_{b=1}^{B} I(W^{(b)} > W).$$

We then apply the Holm–Bonferroni correction or Benjamini–Hochberg procedure on raw $P$ values to control the FDR.

## ANCOM-BC2 for mixed-effects models

Similar to the fixed-effects model stated in equation (3), for each taxon $j, j = 1, \dots, d$, and each sample $i, i = 1, \dots, n$, suppose each sample has $n_i$ observations and $\sum_i n_i = n$. The offset-based mixed-effects log-linear model is set up as

$$\mathbf{y}_{ij} = \theta_i \mathbf{1}_{n_i} + X_i \boldsymbol{\beta}_j + Z_i \boldsymbol{\alpha}_i + \boldsymbol{\epsilon}_{ij}, \quad (26)$$

where

(1) $\mathbf{y}_{ij}$ is the $n_i$ vector-centered observed counts,
(2) $\mathbf{1}_{n_i} = (1, \dots, 1)^T \in \mathbb{R}^{n_i}$ is a vector of 1s,
(3) $X_i$ is the $n_i \times p$ design matrix for fixed effects,
(4) $\boldsymbol{\beta}_j$ is the $p$ vector of fixed-effects regression coefficients to be estimated,
(5) $Z_i$ is the $n_i \times q$ design matrix for the random effects,
(6) $\boldsymbol{\alpha}_i$ is the $q$ vector random effects,
(7) $\boldsymbol{\epsilon}_{ij}$ is the $n_i$ vector residuals.

The following distributional assumptions are made

$$\boldsymbol{\alpha}_i \sim N(\mathbf{0}, D_{q \times q}),$$

$$\boldsymbol{\epsilon}_{ij} \sim N(0, \sigma_j^2 \mathbf{1}_{n_i}),$$

$$\boldsymbol{\alpha}_i \perp\!\!\!\perp \boldsymbol{\epsilon}_{ij} \text{ for } i = 1, \dots, n.$$

Thus, for each taxon $j, j = 1, \dots, d$, and each sample $i, i = 1, \dots, n$, we have

$$\mathbf{y}_{ij} \sim N(\theta_i \mathbf{1}_{n_i} + X_i \boldsymbol{\beta}_j, H_{ij}(\tau)),$$

where $H_{ij}(\tau) = Z_i D Z_i^T + \sigma_j^2 I_{n_i}$ (or $H_{ij}$ for short) denotes a general covariance matrix parametrized by $\tau$.

Stack up observations across samples, we have:

$$\mathbf{y}_j = \boldsymbol{\theta} + X\boldsymbol{\beta}_j + Z\boldsymbol{\alpha} + \boldsymbol{\epsilon}_j, \tag{27}$$

where

$$\mathbf{y}_j = \begin{bmatrix} y_{1j} \\ y_{2j} \\ \vdots \\ y_{nj} \end{bmatrix}, \boldsymbol{\theta} = \begin{bmatrix} \theta_1 \mathbf{1}_{n_1} \\ \theta_2 \mathbf{1}_{n_2} \\ \vdots \\ \theta_n \mathbf{1}_{n_n} \end{bmatrix}, X = \begin{bmatrix} X_1 \\ X_2 \\ \vdots \\ X_n \end{bmatrix}, \boldsymbol{\beta}_j = \begin{bmatrix} \beta_{j1} \\ \beta_{j2} \\ \vdots \\ \beta_{jp} \end{bmatrix},$$

$$Z = \begin{bmatrix} Z_1 & 0 & \dots & 0 \\ 0 & Z_1 & 0 & 0 \\ \vdots & \vdots & \ddots & \vdots \\ 0 & 0 & \dots & Z_1 \end{bmatrix}, \boldsymbol{\alpha} = \begin{bmatrix} \alpha_1 \\ \alpha_2 \\ \vdots \\ \alpha_n \end{bmatrix}, \boldsymbol{\epsilon}_j = \begin{bmatrix} \epsilon_{1j} \\ \epsilon_{2j} \\ \vdots \\ \epsilon_{nj} \end{bmatrix}.$$

That is,

$$\mathbf{y}_j \sim N \left( \boldsymbol{\theta} + X\boldsymbol{\beta}_j, H_j(\tau) = \begin{bmatrix} H_{1j}(\tau) & 0 & \dots & 0 \\ 0 & H_{2j}(\tau) & 0 & 0 \\ \vdots & \vdots & \ddots & \vdots \\ 0 & 0 & \dots & H_{nj}(\tau) \end{bmatrix} \right),$$

where $H_j(\tau)$ (or $H_j$ for short) is a block diagonal matrix.

Similarly, we estimate $\boldsymbol{\theta}$ and $\boldsymbol{\beta}_j$ iteratively to obtain the corresponding preliminary estimators. Compared to Algorithm 1, the maximum likelihood is replaced with restricted maximum likelihood (ReML)[54,55].

**Algorithm 3.** Iterative ReML estimation

1: **Initialize:**
 For $j = 1, \dots, d$
 $\boldsymbol{\theta} \leftarrow 0$
 $\mathbf{y}_j^{(\mathrm{crt})} \leftarrow \mathbf{y}_j - \boldsymbol{\theta} = \mathbf{y}_j$
 $\boldsymbol{\beta}_j \leftarrow \mathrm{ReML}(\mathbf{y}_j^{(\mathrm{crt})}) = \mathrm{ReML}(\mathbf{y}_j)$

(2) **While** not converge **do**

(3) $\quad \boldsymbol{\theta} \leftarrow \frac{1}{d} \sum_{j=1}^{d}(\mathbf{y}_j - X\boldsymbol{\beta}_j)$

(4) $\quad \mathbf{y}_j^{(\mathrm{crt})} \leftarrow \mathbf{y}_j - \boldsymbol{\theta}$

(5) $\quad \boldsymbol{\beta}_j \leftarrow \mathrm{ReML}(\mathbf{y}_j^{(\mathrm{crt})})$

(6) **end while**

Note that the estimators for regression coefficients $\boldsymbol{\beta}_j$ and variance components $\tau$ are obtained iteratively by maximizing the following log-likelihood function:

$$L(\tau|\mathbf{y}_j) = -\sum_{i=1}^{n} \log |H_{ij}| - \sum_{i=1}^{n} \log |X_i^T H_{ij}^{-1} X_i| - \sum_{i=1}^{n} (\mathbf{y}_{ij} - X_i\boldsymbol{\beta}_j)^T H_{ij}^{-1}(\mathbf{y}_{ij} - X_i\boldsymbol{\beta}_j), \tag{28}$$

where $\boldsymbol{\beta}_j \leftarrow (X^T H_j^{-1} X)^{-1} X^T H_j^{-1} \mathbf{y}_j$. As close-form solutions of equation (28) do not exist, the Newton–Raphson method[56] is usually used.

Suppose on convergence, $\boldsymbol{\theta} \leftarrow \boldsymbol{\theta}^*, \mathbf{y}_j^{(\mathrm{crt})} \leftarrow \mathbf{y}_j^{(\mathrm{crt})*}, H \leftarrow H^*, \boldsymbol{\beta}_j \leftarrow \boldsymbol{\beta}_j^*$, we have

$$\boldsymbol{\theta}^* = \frac{1}{d} \sum_{j=1}^{d}(\mathbf{y}_j - X\boldsymbol{\beta}_j^*),$$

$$\mathbf{y}_j^{(\mathrm{crt})*} = \mathbf{y}_j - \boldsymbol{\theta}^*,$$

$$\boldsymbol{\beta}_j^* = (X^T H_j^{*-1} X)^{-1} X^T H_j^{*-1} \mathbf{y}_j^{(\mathrm{crt})*}.$$

It is easy to show that there exists a vector $\boldsymbol{\delta} \in \mathbb{R}^P$, such that

$$E(\boldsymbol{\theta}^*) = \boldsymbol{\theta} - X\boldsymbol{\delta},$$

$$E(\boldsymbol{\beta}_j^*) = \boldsymbol{\delta} + \boldsymbol{\beta}_j.$$

that is, $\boldsymbol{\beta}_j^*$ is a biased estimator for $\boldsymbol{\beta}_j$.

Similar to the case of fixed-effects model, we fit the Gaussian mixture model to each $\beta_{jk}, k = 1, \dots, p$ separately, to correct the bias $\boldsymbol{\delta}$, and final estimators for $\boldsymbol{\beta}_j$ and $\boldsymbol{\theta}$ are given by

$$\hat{\boldsymbol{\beta}}_j = \boldsymbol{\beta}_j^* - \hat{\boldsymbol{\delta}}^{\mathrm{EM}},$$

$$\hat{\boldsymbol{\theta}} = \frac{1}{d} \sum_{j=1}^{d}(\mathbf{y}_j - X\hat{\boldsymbol{\beta}}_j).$$

The statistical inference, including multi-group comparisons, for mixed-effects models, aligns with those outlined in previous sections for fixed-effects models, and therefore, it is not repeated here.

**Strategies implemented in ANCOM-BC2 to handle zeros**

ANCOM-BC2 deals with zero-related challenges in microbiome data as follows. (1) Structural zero identification: taxa that are exclusively present in one ecosystem but absent in another, result in structural zeros. For example, some taxa are exclusive to desert regions but entirely absent in rainforests. Hence, they are structural zeros in rainforests. Those zeros should not be imputed or ignored, and such taxa are DA between the two regions. As the first step, using ANCOM-II (ref. [13]), ANCOM-BC2 identifies all DA taxa that are due to structural zeros, and no further analysis is performed on such taxa and they are cataloged separately in the software output. (2) Prevalence-based filtration: after filtering structural zeros, ANCOM-BC2 applies a prevalence-based filtration, akin to other DA methods. By default, taxa that feature in less than 10% of all samples are removed from further analysis. (3) Sensitivity analysis for pseudo-count addition to zeros: for the remaining taxa with some zeros, we perform a sensitivity analysis to assess their robustness to pseudo-counts as follows. Much like many DA analysis methodologies, since ANCOM-BC2 log transforms the observed counts, the counts need to be positive. Often pseudo-counts are added to deal with zeros. However, it is well-known that the choice of the pseudo-count can considerably influence the false-positive as well as false-negative rates[13,16,17]. To mitigate this concern, we conduct a sensitivity analysis to evaluate the effect of varying pseudo-counts on zeros for each taxon. This procedure incorporates the addition of an array of pseudo-counts (ranging from 0.01 to 0.5 in increments of 0.01) to the zero counts for each taxon. Corresponding to each pseudo-count, ANCOM-BC2 is used for each taxon and $P$ values for DA analysis are derived. The sensitivity score for each taxon is the proportion of instances where the $P$ values exceed the specified significance level. If the proportion of significant (or non-significant) results is 1 and the significance (or non-significance) aligns with significance (or non-significance) using complete data (excluding zeros), then the taxon is regarded as insensitive to the pseudo-count addition. Otherwise, it is deemed sensitive. This step remains a recommendation and is at the discretion of the users. We offer two versions of ANCOM-BC2 for flexibility: (1) ANCOM-BC2 (no filter): this version only uses the first two steps for handling zeros and uses complete data (that is, excludes zeros by treating them as missing completely at random) for bias correction and inference. While it has larger power, it might display an inflated FDR, especially with larger sample sizes or repeated measures. (2) ANCOM-BC2 (SS filter): this version uses all three aforementioned steps for dealing with zeros and also uses complete data for both bias correction and inference. Specifically, if a taxon is found to be sensitive to pseudo-counts then it is declared as non-significant taxon. While more conservative, it provides rigorous control of FDR, albeit with a possible decrease in power.

**Reporting summary**

Further information on research design is available in the Nature Portfolio Reporting Summary linked to this article.

## Data availability

The URT data were sourced from the LOCOM R package https://github.com/yijuanhu/LOCOM-Archive. The Quantitative Microbiome Project data are accessible via the SPRING R package (https://github.com/GraceYoon/SPRING) or the ANCOMBC package (https://www.bioconductor.org/packages/release/bioc/html/ANCOMBC.html). Data pertaining to soil microbiome for aridity and gut microbiome in patients with IBD are hosted on Qiita, with respective links available at https://qiita.ucsd.edu/study/description/10360 and https://qiita.ucsd.edu/study/description/11546, respectively. Please note that accessing data on Qiita requires account registration and sign-in.

## Code availability

ANCOM-BC2 has been implemented in the R package ANCOMBC, which is available on Bioconductor at https://www.bioconductor.org/packages/release/bioc/html/ANCOMBC.html. The code used for all analyses, with the exception of the trend test related to soil microbiome richness, in this Article is available in the associated GitHub repository and the corresponding Code Ocean capsule https://doi.org/10.24433/CO.0628172.v1. The specific trend test was conducted using ORIOGEN 4.01, obtainable at https://www.niehs.nih.gov/research/resources/software/biostatistics/oriogen/index.cfm.

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

## Acknowledgements

This research by H.L. and S.D.P. was supported (in part) by funding from the National Institute of Environmental Health Sciences (NIEHS) intramural program no. ZIA ES103389-01.

## Author contributions

S.D.P. and H.L. contributed equally to the theory and methodology described in this Article. They also contributed equally to writing and editing the article. All numerical works and computations were conducted by H.L. who developed ANCOM-BC2 pipeline in R that is freely and publicly available. Please contact H.L. for software requests.

## Competing interests

The authors declare no competing interests.

## Additional information

**Extended data** is available for this paper at https://doi.org/10.1038/s41592-023-02092-7.

**Correspondence and requests for materials** should be addressed to Shyamal Das Peddada.

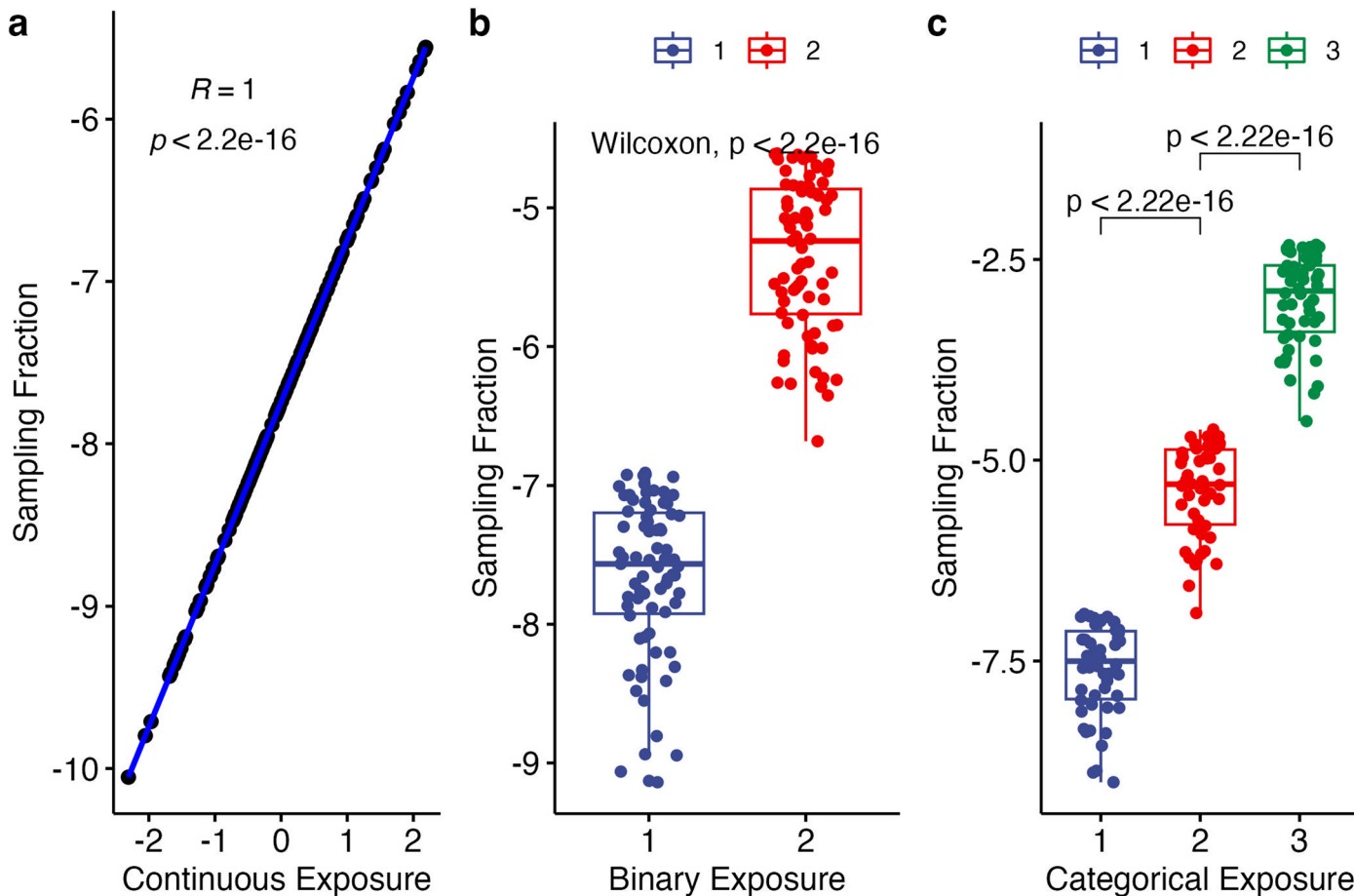

**Extended Data Fig. 1 | Illustration of batch effects in simulation studies where sampling fractions were programmed to correlate highly with the exposure of interest. (a)** Continuous exposure versus sampling fractions. Scatter plot for 150 simulated samples reveals the positive linear relationship between continuous exposure (X-axis) and sampling fractions (Y-axis). The regression fit is shown in blue. The strong correlation is emphasized by a Pearson's R of 1 and a two-sided p value < $2.2 \times 10^{-16}$ **(b)** Binary exposure versus sampling fractions. Box plots detail distributions of sampling fractions (Y-axis) across two groups (X-axis) based on 150 simulated samples (75 per group). Each box signifies the interquartile range (IQR) of the data, the median is indicated by the interior line, and whiskers extend to the maximum and minimum values within 1.5 times the IQR from the box. Potential outliers are represented as points outside the whiskers, and jittered points indicate individual data points. A two-sided p-value < $2.2 \times 10^{-16}$ from a Wilcoxon rank-sum test denotes significant group differences. **(c)** Categorical exposure versus sampling fractions. Box plots showcase distributions of sampling fractions (Y-axis) for three groups (X-axis) using 150 samples (50 per group). Each box, line, whisker, and point represents the same elements as in **(b)**. Pairwise significant differences are denoted by two-sided p-values < $2.2 \times 10^{-16}$ following a Wilcoxon rank-sum test.

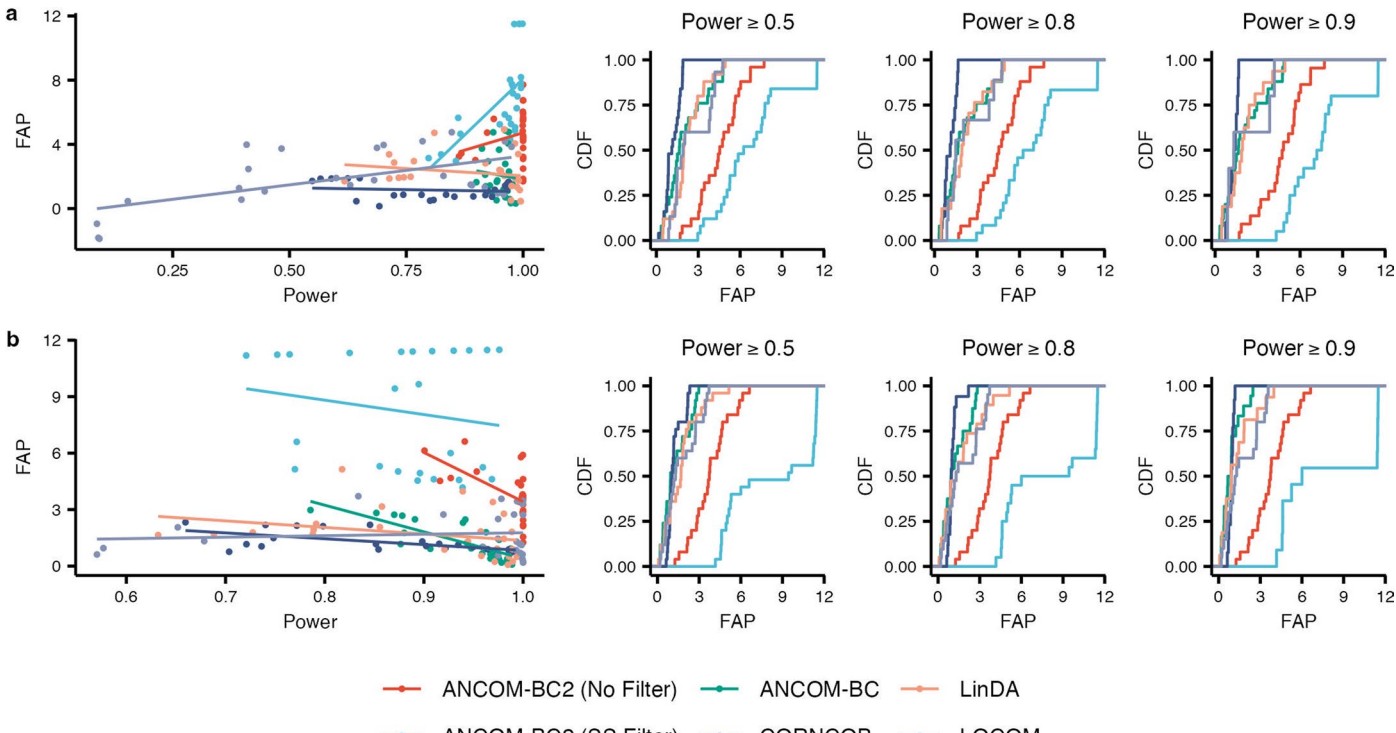

**Extended Data Fig. 2 | FDR Adjusted Power (FAP) among various DA methods.**
FAP, defined as the log ratio of power and FDR, was employed to illustrate the
power/FDR trade-off among all DA methods. FAP values were calculated using
power and FDR metrics obtained from the simulation studies carried out for both
**(a)** continuous and **(b)** binary exposure scenarios utilizing the URT dataset[19]. The
far left panels of this figure present scatter plots of FAP (Y-axis) corresponding
to the power (X-axis) for all DA methods considered in the simulation study
reported in Fig. 1 in the main text. FAPs are expressed as mean values deduced
from 100 simulation iterations per setting, with the linear regression line of FAP
against power superimposed over the points. On the right of the scatter plots in
each panel are the three cumulative density function (CDF) plots of FAP scores
of various DA methods corresponding to powers exceeding 0.5, 0.8, and 0.9,
respectively. These results underscore that both versions of ANCOM-BC2 have
stochastically larger FAP scores than the competitors, with ANCOM-BC2 (SS
Filter) being stochastically the largest.

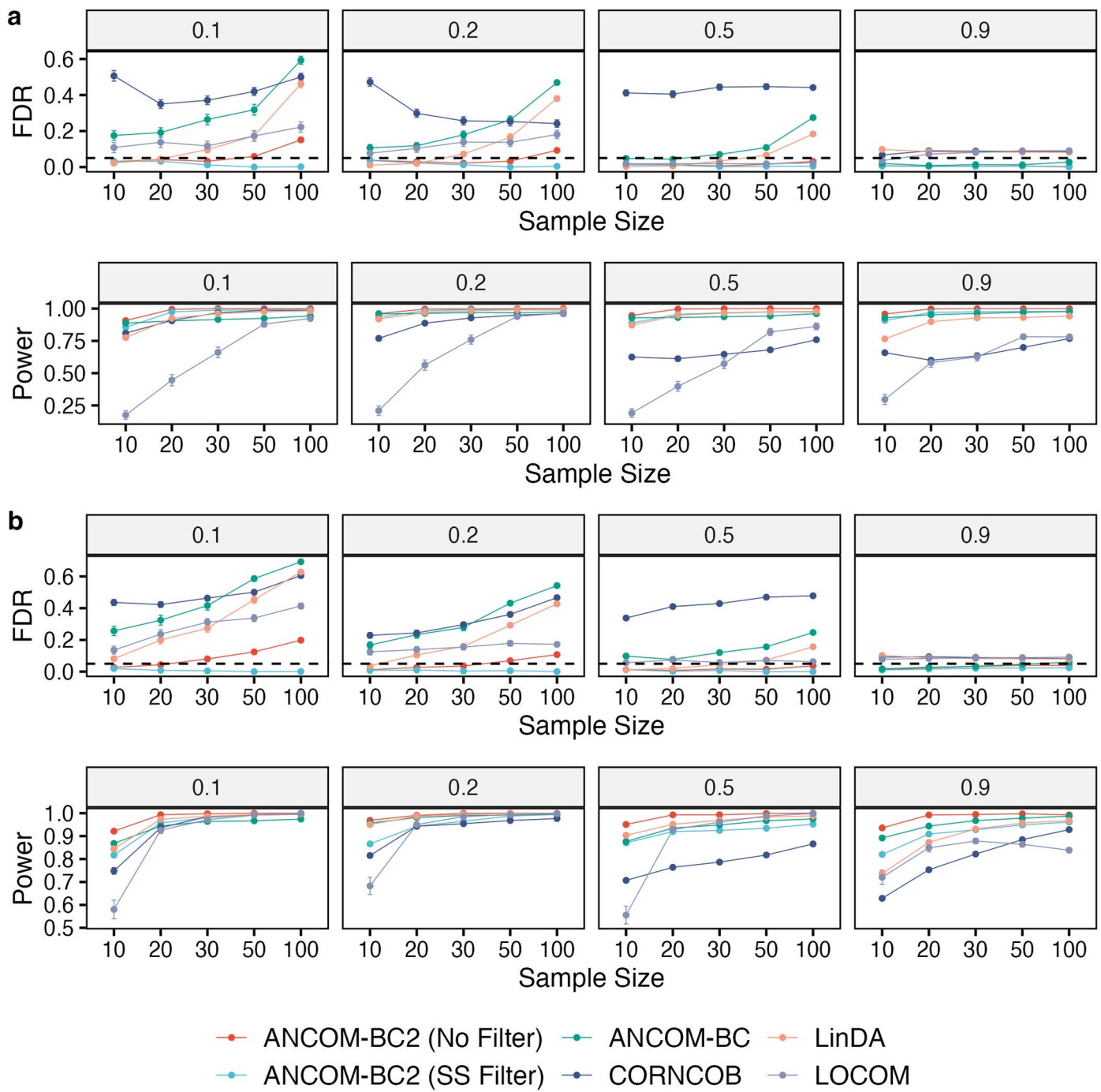

**Extended Data Fig. 3 | Evaluations of FDR (mdFDR) and power in identifying DA taxa in (a) continuous or (b) binary exposure.** Synthetic datasets were generated using the PLN model[18] based on the mean and covariance estimated from the QMP dataset[23]. The X-axis shows the sample size (or sample size per group for the categorical covariate), and the Y-axis shows the FDR (mdFDR) or power. Each panel title designates the proportion of true DA taxa. The depicted metrics represent mean values ± standard errors (indicated by error bars) derived from 100 simulation runs for each setting. This visualization underscores the superiority of ANCOM-BC2-both with and without the sensitivity score (SS) filter-in consistently preserving minimal FDR or mdFDR while attaining satisfactory power, outpacing all other assessed methods.

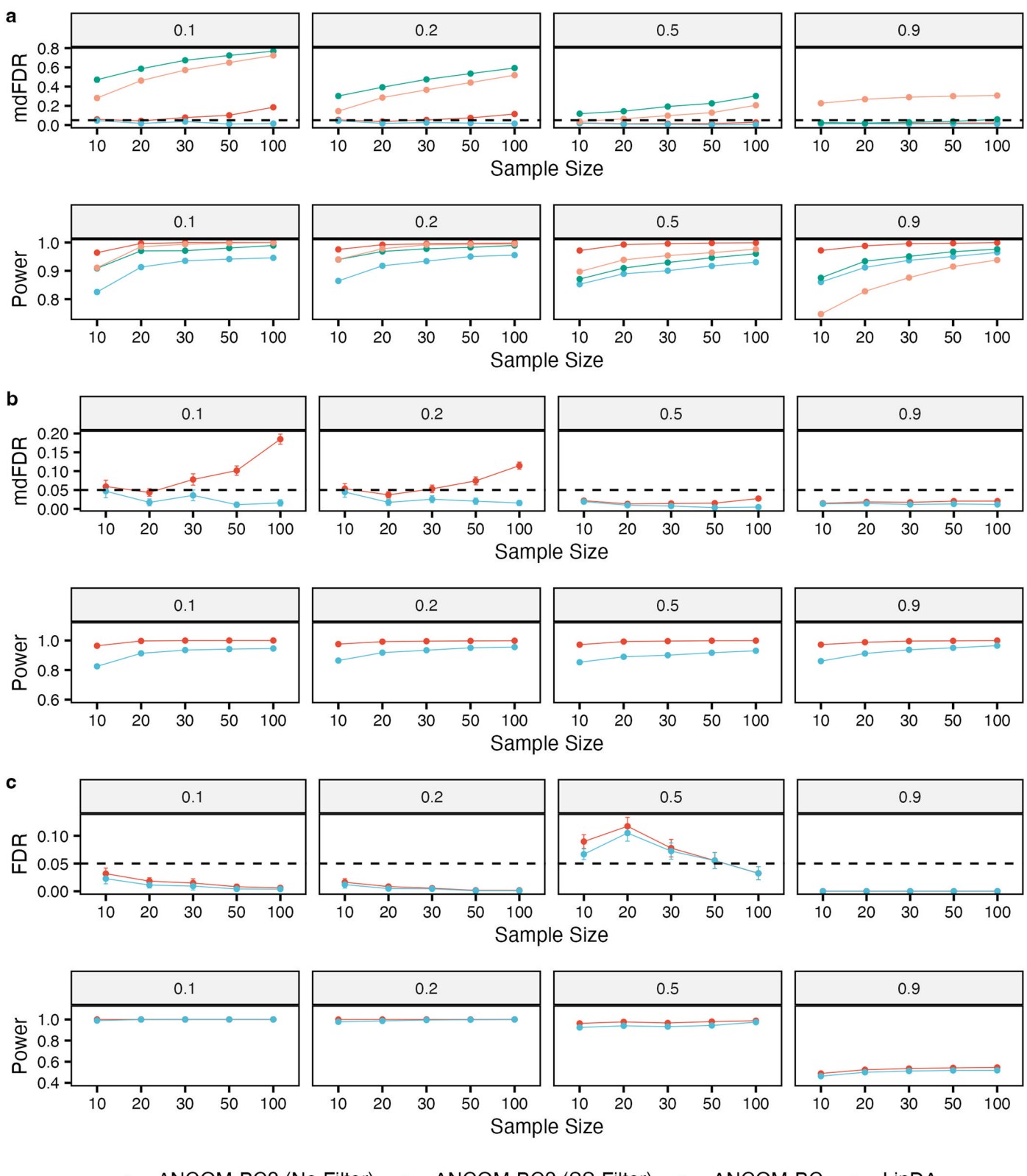

**Extended Data Fig. 4 | Evaluations of FDR (mdFDR) and power in identifying DA taxa in (a) multiple pairwise comparisons against a reference group, (b) multiple pairwise comparisons, and (c) pattern analysis.** Synthetic datasets were generated using the PLN model[18] based on the mean and covariance estimated from the QMP dataset[23]. The X-axis shows the sample size per group, and the Y-axis shows the FDR (mdFDR) or power. Each panel title designates the proportion of true DA taxa. The depicted metrics represent mean values ± standard errors (indicated by error bars) derived from 100 simulation runs for each setting. Within the context of multiple pairwise comparisons, ANCOM-BC2-when implemented with the SS filter-effectively controlled FDR (mdFDR) while maintaining power akin to its performance without the SS filter. In the pattern analysis, ANCOM-BC2-both with and without the SS filter-most often maintained the FDR under the nominal level while achieving adequate power, barring the scenario with 90% DA taxa. In this instance, ANCOM-BC2-both with and without the SS filter-experienced power loss due to inherent assumptions in bias correction.

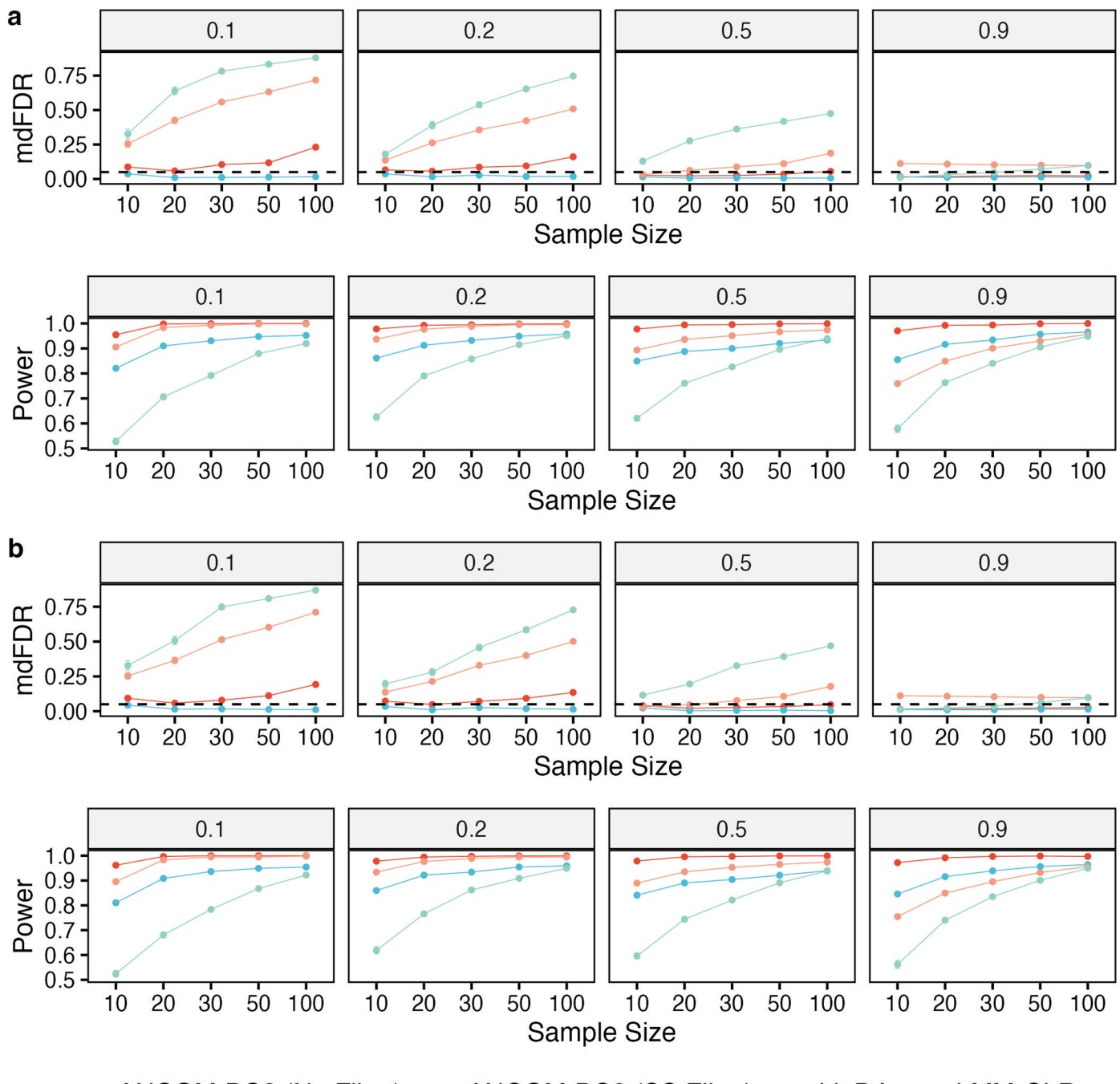

**Extended Data Fig. 5 | Evaluations of FDR (mdFDR) and power in identifying DA taxa in (a) a random intercept model, and (b) a random coefficients model.** Synthetic datasets were generated using the PLN model[18] based on the mean and covariance estimated from the QMP dataset[23]. The X-axis shows the sample size per group, and the Y-axis shows the FDR (mdFDR) or power. Each panel title designates the proportion of true DA taxa. The depicted metrics represent mean values ± standard errors (indicated by error bars) derived from 100 simulation runs for each setting. The outcomes accentuate that, when integrated with the SS filter, ANCOM-BC2 effectively moderates FDR (mdFDR) while retaining power parallel to its performance without the SS filter. In the absence of the SS filter, ANCOM-BC2 surpasses LinDA and LMM-CLR in maintaining consistently low FDR and equivalent power.

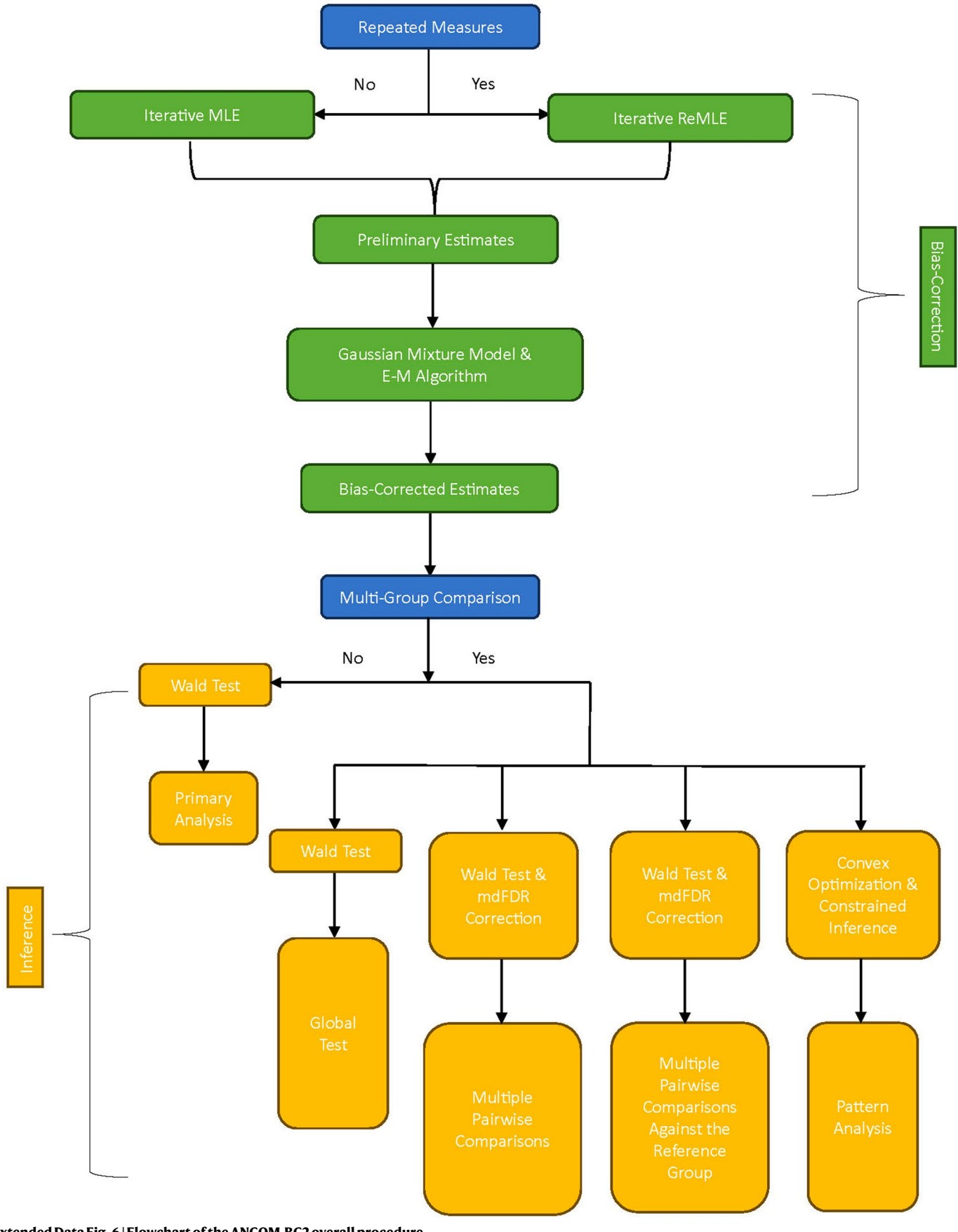

**Extended Data Fig. 6 | Flowchart of the ANCOM-BC2 overall procedure.**

# Reporting Summary

## Statistics

For all statistical analyses, confirm that the following items are present in the figure legend, table legend, main text, or Methods section.

| n/a | Confirmed | |
|---|---|---|
| ☐ | ☒ | The exact sample size (*n*) for each experimental group/condition, given as a discrete number and unit of measurement |
| ☐ | ☒ | A statement on whether measurements were taken from distinct samples or whether the same sample was measured repeatedly |
| ☐ | ☒ | The statistical test(s) used AND whether they are one- or two-sided<br>*Only common tests should be described solely by name; describe more complex techniques in the Methods section.* |
| ☐ | ☒ | A description of all covariates tested |
| ☐ | ☒ | A description of any assumptions or corrections, such as tests of normality and adjustment for multiple comparisons |
| ☐ | ☒ | A full description of the statistical parameters including central tendency (e.g. means) or other basic estimates (e.g. regression coefficient) AND variation (e.g. standard deviation) or associated estimates of uncertainty (e.g. confidence intervals) |
| ☐ | ☒ | For null hypothesis testing, the test statistic (e.g. *F*, *t*, *r*) with confidence intervals, effect sizes, degrees of freedom and *P* value noted<br>*Give P values as exact values whenever suitable.* |
| ☒ | ☐ | For Bayesian analysis, information on the choice of priors and Markov chain Monte Carlo settings |
| ☒ | ☐ | For hierarchical and complex designs, identification of the appropriate level for tests and full reporting of outcomes |
| ☐ | ☒ | Estimates of effect sizes (e.g. Cohen's *d*, Pearson's *r*), indicating how they were calculated |

*Our web collection on statistics for biologists contains articles on many of the points above.*

## Software and code

Policy information about availability of computer code

| Data collection | No software was used for data collection. |
|---|---|
| Data analysis | ANCOM-BC2 has been implemented in the R package ANCOMBC, which is available on Bioconductor at https://www.bioconductor.org/packages/release/bioc/html/ANCOMBC.html. The entirety of the analyses, conducted in RStudio (R version 4.2.2) on aarch64-apple-darwin20 (64-bit) running macOS Monterey 12.6.8, with the sole exception of the soil microbiome richness trend test, is provided in the related GitHub repository, which can be found at https://github.com/FrederickHuangLin/ANCOM-BC2-Code-Archive. For the aforementioned exception, ORIOGEN 4.01 was employed and is available at https://www.niehs.nih.gov/research/resources/software/biostatistics/oriogen/index.cfm. |

For manuscripts utilizing custom algorithms or software that are central to the research but not yet described in published literature, software must be made available to editors and reviewers. We strongly encourage code deposition in a community repository (e.g. GitHub). See the Nature Portfolio guidelines for submitting code & software for further information.

## Data

Policy information about availability of data

All manuscripts must include a data availability statement. This statement should provide the following information, where applicable:
- Accession codes, unique identifiers, or web links for publicly available datasets
- A description of any restrictions on data availability
- For clinical datasets or third party data, please ensure that the statement adheres to our policy

The upper respiratory tract (URT) data were sourced from the LOCOM R package (https://github.com/yijuanhu/LOCOM-Archive). The Quantitative Microbiome Project (QMP) data are accessible via the SPRING R package (https://github.com/GraceYoon/SPRING) or the ANCOMBC package (https://www.bioconductor.org/packages/release/bioc/html/ANCOMBC.html). Data pertaining to soil microbiome for aridity and gut microbiome in IBD patients are hosted on Qiita, with respective links available at https://qiita.ucsd.edu/study/description/10360 and https://qiita.ucsd.edu/study/description/11546, respectively.

## Human research participants

Policy information about studies involving human research participants and Sex and Gender in Research.

| | |
|---|---|
| Reporting on sex and gender | No experiment in this study |
| Population characteristics | No experiment in this study |
| Recruitment | No experiment in this study |
| Ethics oversight | No experiment in this study |

Note that full information on the approval of the study protocol must also be provided in the manuscript.

# Field-specific reporting

Please select the one below that is the best fit for your research. If you are not sure, read the appropriate sections before making your selection.

☒ Life sciences          ☐ Behavioural & social sciences          ☐ Ecological, evolutionary & environmental sciences

For a reference copy of the document with all sections, see nature.com/documents/nr-reporting-summary-flat.pdf

# Life sciences study design

All studies must disclose on these points even when the disclosure is negative.

| | |
|---|---|
| Sample size | We utilized two publicly available datasets: soil microbiome for aridity and gut microbiome in IBD patients. No sample size calculation was performed. |
| Data exclusions | We utilized all samples provided by the authors on the public website for the two datasets. No exclusions were made. |
| Replication | Not relevant. We used publicly available datasets and did not participate in sample generation. |
| Randomization | Not relevant. For this paper, we utilized two publicly available datasets: soil microbiome for aridity and gut microbiome in IBD patients. We did not conduct any randomization. |
| Blinding | Not relevant. Two real datasets used for illustrations in this paper, i.e., soil microbiome for aridity and gut microbiome in IBD patients, are publicly available. We were not involved in the blinding or unblinding process. |

# Reporting for specific materials, systems and methods

We require information from authors about some types of materials, experimental systems and methods used in many studies. Here, indicate whether each material, system or method listed is relevant to your study. If you are not sure if a list item applies to your research, read the appropriate section before selecting a response.

## Materials & experimental systems

| n/a | Involved in the study |
|---|---|
| ☒ | ☐ Antibodies |
| ☒ | ☐ Eukaryotic cell lines |
| ☒ | ☐ Palaeontology and archaeology |
| ☒ | ☐ Animals and other organisms |
| ☒ | ☐ Clinical data |
| ☒ | ☐ Dual use research of concern |

## Methods

| n/a | Involved in the study |
|---|---|
| ☒ | ☐ ChIP-seq |
| ☒ | ☐ Flow cytometry |
| ☒ | ☐ MRI-based neuroimaging |

