## [Peer Review File · Nature Methods]

Peer Review Information

Manuscript Title: Multi-group Analysis of Compositions of Microbiomes with Covariate Adjustments and Repeated Measures

Corresponding author name(s): Shyamal Das Peddada

Editorial Notes:

Reviewer Comments & Decisions:

Decision Letter, initial version:

7th Jun 2023

Dear Dr Peddada,

Your Article, "Multi-group Analysis of Compositions of Microbiomes with Covariate Adjustments and Repeated Measures", has now been seen by 3 reviewers. As you will see from their comments below, although the reviewers find your work of potential interest, they have raised a number of serious concerns. We are interested in the possibility of publishing your paper in Nature Methods, but would like to consider your response to these concerns before we reach a final decision on publication.

We therefore invite you to revise your manuscript to address all these concerns. In particular, additional data need to show substantial performance improvement over competing tools.

* include a point-by-point response to the reviewers and to any editorial suggestions

* please underline/highlight any additions to the text or areas with other significant changes to facilitate review of the revised manuscript

- * address the points listed described below to conform to our open science requirements
- * ensure it complies with our general format requirements as set out in our guide to authors at www.nature.com/naturemethods
- * resubmit all the necessary files electronically by using the link below to access your home page

[REDACTED]

We hope to receive your revised paper within three months. If you cannot send it within this time, please let us know. In this event, we will still be happy to reconsider your paper at a later date so long as nothing similar has been accepted for publication at Nature Methods or published elsewhere.

OPEN SCIENCE REQUIREMENTS

REPORTING SUMMARY AND EDITORIAL POLICY CHECKLISTS

DATA AVAILABILITY

We strongly encourage you to deposit all new data associated with the paper in a persistent repository where they can be freely and enduringly accessed. We recommend submitting the data to discipline-specific and community-recognized repositories; a list of repositories is provided here:

<http://www.nature.com/sdata/policies/repositories>

All novel DNA and RNA sequencing data, protein sequences, genetic polymorphisms, linked genotype and phenotype data, gene expression data, macromolecular structures, and proteomics data must be deposited in a publicly accessible database, and accession codes and associated hyperlinks must be provided in the "Data Availability" section.

CODE AVAILABILITY

Please include a "Code Availability" subsection in the Online Methods which details how your custom code is made available. Only in rare cases (where code is not central to the main conclusions of the paper) is the statement "available upon request" allowed (and reasons should be specified).

For more information on our code sharing policy and requirements, please see: <https://www.nature.com/nature-research/editorial-policies/reporting-standards#availability-of-computer-code>

SUPPLEMENTARY PROTOCOL

To help facilitate reproducibility and uptake of your method, we ask you to prepare a step-by-step Supplementary Protocol for the method described in this paper. We [encourage authors to share their step-by-step experimental protocols](https://www.nature.com/nature-research/editorial-policies/reporting-standards#protocols) on a protocol

sharing platform of their choice and report the protocol DOI in the reference list. Nature Portfolio 's Protocol Exchange is a free-to-use and open resource for protocols; protocols deposited in Protocol Exchange are citable and can be linked from the published article. More details can found at www.nature.com/protocolexchange/about.

ORCID

Nature Methods is committed to improving transparency in authorship. As part of our efforts in this direction, we are now requesting that all authors identified as 'corresponding author' on published papers create and link their Open Researcher and Contributor Identifier (ORCID) with their account on the Manuscript Tracking System (MTS), prior to acceptance. This applies to primary research papers only. ORCID helps the scientific community achieve unambiguous attribution of all scholarly contributions. You can create and link your ORCID from the home page of the MTS by clicking on 'Modify my Springer Nature account'. For more information please visit please visit www.springernature.com/orcid.

Best regards,
Lei

Lei Tang, Ph.D.
Senior Editor
Nature Methods

Reviewers' Comments:

Reviewer #1:

Remarks to the Author:

In the manuscript entitled "Multi-group Analysis of Compositions of Microbiomes with Covariate Adjustments and Repeated Measures", the authors developed a general framework for performing a wide range of multi-group analyses with covariate adjustments and repeated measures and showed that the proposed method outperforms existing state-of-art methods. We have many concerns on the current form of the manuscript:

1. The authors systematically compared the proposed method ANCOM-BC2 with the existing ones on simulated datasets. The simulated datasets were generated based on the Quantitative Microbiome Project (QMP) data, which consists only 91 operational taxonomic units (OTUs). Why didn't the authors use the upper respiratory tract (URT) microbiome data [PMID: 21188149] to generate simulated data? Note that both competitive methods of ANCOM-BC2 (i.e., LinDA and LOCOM) used this dataset to generate synthetic data.

2. The authors examined the FDR and power under the different true proportions of DA taxa, however, how were those taxa selected and the selection methods of DA taxa might impact the performance. The authors should explicitly address this issue.
3. What are the results for more sparse signals, e.g., the percentage of differential taxa is 5%?
4. Which kind of normalization and transformation were used?
5. The authors should provide more details regarding how they generated the simulated data, e.g., the distribution of the data and we cannot understand the reason for the significantly high FDR of other methods, including authors previously developed one without those details. For instance, LinDA assumes a log-normal distribution of the absolute abundance. Users might generate the simulated data from Poisson lognormal distribution where the mean and covariates were estimated from relative abundance.
6. The FDR of ANCOM-BC2 is close to 0 in various parameter setup. Is this still true if the authors generate the simulated dataset differently?

Reviewer #2:

Remarks to the Author:

Lin and Peddada present an extension of the original ANCOM-BC, called ANCOM-BC2, that is capable of testing for patterns and contrasts across multiple experimental groups and accounting for repeated measures. They also introduce some additional bias corrections, such as an estimation of taxon-level efficiencies and filters for false positive results due to the addition of pseudo-counts. They compare ANCOM-BC2 to a small set of competing methods on simulated data and apply it to some example analysis of sequencing data from soil and human gut microbiome samples.

The microbiome field as a whole is still looking for methods that can detect bacterial abundance changes in amplicon and metagenomic sequencing data with good sensitivity *and* precision, and I would agree that the added designs (grouping patterns and repeated measures) are common and not always handled very well by other methods. However, in its current form, I don't feel the manuscript is making a strong argument that this was achieved. There is a lack of transparency in the used methodology to compare ANCOM-BC2 to other methods, and the presented data does not support many of the claims made regarding performance. In particular, ANCOM-BC2 looks to be severely less powered than all other tested methods for small sample sizes, which limits its applicability in its current form. If those challenges can be addressed, I do think that the method would constitute a great addition to the field.

Major Comments

The analysis of method performance based on simulated data is a key outcome of the paper. So it was surprising to find basically no description of how the simulation was carried out. In particular, I would have liked to see the relevant key parameters of such a simulation such as the added effect size, the level of noise added to the data, the number of independent simulation runs, and how other model parameters were estimated from the QMP data.

The authors state in the caption for Figure 1 that “that ANCOM-BC2 outperformed all competing methods in terms of uniformly small FDR (mdFDR) and comparable power”. Though the first part of that statement rings true, the second is not supported by the presented data. In fact, ANCOM-BC2 is much less powered than other methods in all tested scenarios and in small sample sizes that power difference exceeds 50% meaning that in those cases ANCOM-BC2 will fail to detect the majority of true positives. It seems like ANCOM-BC2 just takes up a different spot on the sensitivity-precision tradeoff, where it achieves high precision but sacrifices sensitivity, whereas the other tested methods do the opposite. I appreciate the authors' effort to improve FDR control which is definitely important, but so is power, especially in microbiome data where effect sizes are small and large studies require substantial funding. Maybe ranking methods by F1-score would give a more balanced comparison here.

The FDR results in Figure 1 were a bit surprising to me. Based on the claims made in <https://doi.org/10.1038/s41467-020-17041-7> (in particular Fig. 4a) shouldn't ANCOM-BC control the FDR pretty well in the settings presented in panels (a) and (b)? Was some additional bias introduced in the simulation that can explain the inflated FDR?

In that vein, some of the introduced strategies are not specific to ANCOM-BC2. For instance, the presented method of filtering results sensitive to pseudo-count addition in Remark 2 is mostly a post-hoc filter that could be used with any of the competing methods. From Supplementary Figure 2, it looks like this could actually be responsible for the better FDR control. So I wonder how Figure 1 would look like if that post-hoc filter would be added to all tested methods.

The authors state that “By using real data as a template, we ensured that the data-generating process did not favor our methods, enabling a fair comparison across all methods.”, but it's impossible to assess that since the manuscript does not specify the actual statistical model used to generate the data. In the mentioned ANCOM-BC2 R package one of the vignettes seems to use a Poisson Log-Normal Model (PNLM) to model the simulated data, but I would argue that this does intrinsically favor methods with similar distribution assumption which includes ANCOM-BC2. Would the results look similar if the simulated data was generated from a Negative Binomial, Beta-Binomial, or a zero-inflated discrete count model?

The set of alternative methods tested is pretty small and limited to methods making similar assumptions as ANCOM-BC(2). How would this look when comparing to other commonly used methods such as ALDEX2 (<https://doi.org/10.1186/2049-2618-2-15>), DESEQ2 (<https://doi.org/10.1186/s13059-014-0550-8>), CORNCOB (<https://doi.org/10.1214/19-aos1283>), or any of the other methods tested in the original LinDA manuscript (<https://doi.org/10.1186/s13059-022-02655-5>)?

Minor Comments

Additional to the methods mentioned above, it might be worthwhile to add a baseline comparison where ANCOM-BC2 is compared to some regular mixed effects models on the CLR-transformed data, because that setup is often used for repeated measures analyses of microbiome data.

The code provided in the reproducible capsule is lacking some comments and general explanation of what is happening. It's really hard to follow it the way it is presented right now (pure wall of code with

no explanations in the Rmarkdown files).

Neither of the illustration examples explains how the raw data was processed and reads were mapped to taxa (which seems to be the input the authors used). This would be fine in the supplement.

Reviewer #3:

Remarks to the Author:

In this paper, Lin and Peddada extended their earlier work on ANCOM, ANCOM-BC to ANCOM-BC2. The new approach allows for two types of bias: the sample specific bias which is considered to be the same across taxa in the same sample and the taxon-specific bias that is considered to be the same across samples but different for different taxa. The paper also focused on multi-group analysis in which they investigate directional hypothesis, pairwise group comparison, trend test or pattern test. Have a method that is able to adjust for bias is a very important and difficult task for microbiome data analysis, given that there is mounting evidence indicating the existence and huge effect of taxon-specific bias. The proposed method is very complex with some heuristic decisions. Please see the following as my major comments:

1. The definition of "taxon-specific" bias is unclear to me. As I understand it, the taxon-specific bias is generated because some taxa are easier to measure than others, i.e., the sampling fraction is not uneven for different taxa even within a sample. However, in the method section, the author also stated that "Since the sampling fraction is constant for all taxa within a sample, we pool information across taxa within each sample when estimating δ ", which is contractive to my understanding of bias. Could the authors explain this point in more detail?
2. Was the underlying hypothesis a compositional hypothesis or a hypothesis on relative abundance? In specific, suppose that we have three taxa with real abundance of 100, 200, 300 in one sample, and abundance of 500, 200, 300 in another sample. If we are interested in the compositional hypothesis, only taxon one is differentially abundant. However, if we are interested in the differences in relative abundances, all three taxa are DA. Through the log-linear model setup, I feel that ANCOM-BC2 conducts hypothesis testing at the compositional level. However, this was not clear to me.
3. Given the algorithm is quite complex, it may be helpful if a flowchart is created to inform potential users the analysis steps using the ANCOM-BC2 algorithm, providing summary information about which test option is used (such as asymptotic results or the permutation test based on some specific statistics).
4. I am not exactly sure how β_j can be calculated at equation (4) and use the iterative approach. To me, both θ_i and β_j are non-identifiable. Does the proposed iterative approach achieve only one out of infinitely many parameters from the entire space? The rationale behind obtaining the bias-correction term δ was not clear to me.
5. I worry about the FDR control of the proposed method in more realistic setups. Many decisions in the algorithm are quite heuristic, such as the choice of sensitivity score cutoff (for the pseudo-count imputation) and the choices of multiple comparison adjustment. Heuristic decisions usually generate methods that control type I error and FDR in some simulation set up, but not in others. For example, does the method control FDR when the sample size is much larger? How about in situations when the taxon-specific bias factors are very big? In the simulation, the authors assessed situations when the bias

factor is sampled from $C \sim U[0.1, 1]$. However, experiment from mock community shows that the bias factor can be very uneven. How is the model performance under more extreme bias setups?

6. The cutoff of the sensitivity score for pseudo-count imputation is quite heuristic. It would be better if the authors can explain the rationale of choosing of sensitivity score of 3 as cutoff.

7. The comparison on FDR is unfair for the competing methods because they used the BH approach for FDR controls yet the proposed method used Bonferroni correction, which is designed to control the family wise error rate instead of FDR.

8. What is the computational time of the proposed algorithm, and compared to the competing methods?

9. The paper can also benefit from the discussion on the sparsity level of the microbiome taxa. Are rare taxa more likely to fail the sensitivity score cutoff compared to common taxa? If this is the case, does it indicate that we should filter the data more extensively? Some of the competing methods, such as LOCOM, generally requires filtering the taxa and only keep taxa that are present in a sufficient number of samples. I am wondering whether the FDR inflation can be due to the insufficient number of data? After all, in the simulation setup, the number of samples can go as low as 10.

10. In assumption 4 of page 12, a sparse correlation was assumed for the estimation of δ . However, in microbiome data, because all bacterial live in the same community, it is possible/or even likely that the correlations between taxa are dense instead of sparse. How critical is the sparse correlation assumption is for the algorithm?

Author Rebuttal to Initial comments

Response to comments by Reviewer 1:

We are very grateful to the reviewer for their valuable time and effort and constructive comments that have led to a substantial improvement in the content and presentation of our manuscript. In the following we provide item by item responses to the comments. Reviewer's comments are in italics and our response follow in regular font.

In the manuscript entitled "Multi-group Analysis of Compositions of Microbiomes with Covariate Adjustments and Repeated Measures", the authors developed a general framework for performing a wide range of multi-group analyses with covariate adjustments and repeated measures and showed that the proposed method outperforms existing state-of-art methods. We have many concerns on the current form of the manuscript:

1. The authors systematically compared the proposed method ANCOM-BC2 with the existing ones on simulated datasets. The simulated datasets were generated based on the Quantitative Microbiome Project

(QMP) data, which consists only 91 operational taxonomic units (OTUs). Why didn't the authors use the upper respiratory tract (URT) microbiome data [PMID: 21188149] to generate simulated data? Note that both competitive methods of ANCOM-BC2 (i.e., LinDA and LOCOM) used this dataset to generate synthetic data.

Response: We thank the reviewer for this comment, and we have now considered the upper respiratory tract (URT) microbiome data for simulation purposes. The results can be found in Figures 1 - 3 in the main text. Additionally, we have relegated the simulation results using the QMP data to Supplementary Figures 3 - 5.

2. The authors examined the FDR and power under the different true proportions of DA taxa, however, how were those taxa selected and the selection methods of DA taxa might impact the performance. The authors should explicitly address this issue.

Response:

We thank the reviewer for this comment. In view of the reviewer's comments, we have explained the simulation set-up more clearly in this revision (please refer to the expanded Supplementary Information as well as the Simulation Settings section in the main text). Specifically, for each simulation scenario, we randomly selected DA (differentially abundant) taxa. For instance, in the scenario with 10% DA taxa, we randomly chose 38 taxa out of the 382 taxa in the URT data. To generate the simulation data, we used the Poisson lognormal (PLN) model described in the LDM paper by Hu and Satten (2020) (<https://doi.org/10.1093/bioinformatics/btaa260>) based on the estimated mean and variance from the URT dataset. Each simulation scenario, ranging from 5% to 90% DA taxa, was run 100 times using different seeds in R, resulting in distinct sets of DA taxa.

3. What are the results for more sparse signals, e.g., the percentage of differential taxa is 5%?

Response: Thanks for the suggestion. Accordingly, we have now expanded the simulation studies to include 5% differentially abundant data. The corresponding results were summarized in Figures 1 - 3 in the main text.

4. Which kind of normalization and transformation were used?

Response: We are grateful for your insightful comment. In response, we would like to elaborate on the preprocessing stages for each differential abundance (DA) method discussed in our study. For ANCOM-BC2 and ANCOM-BC, no external normalization or transformation was applied to the input data. These methodologies internally estimate and correct biases, such as sample- and taxon-specific biases in ANCOM-BC2, and sample-specific bias in ANCOM-BC, prior to conducting statistical inferences. Therefore, our proposed methods operate on internally "normalized" or "bias-corrected" counts. In the case of LinDA, it applies the centered Log-Ratio (CLR) transformation to the input data and incorporates an internal normalization procedure as part of its bias-correction process, akin to ANCOM-BC2 and ANCOM-BC. Hence no further external transformation or normalization steps were performed for LinDA either. LOCOM, on the other hand, accepts relative abundances (proportions) as input, which can be viewed as data already subjected to a "total-sum scaling" normalization. LOCOM infers changes in absolute abundance through a transformation similar to the Additive Log-Ratio (ALR) transformation. As such, LOCOM does not require any additional external transformation or normalization, and hence again we did not perform any additional normalization. CORNCOB, designed specifically for analyzing relative abundances, includes an internal "total-sum scaling" normalization procedure. Accordingly, once again, no additional normalization or transformation was performed on the data for CORNCOB. To provide further clarity on these preprocessing procedures used for each DA method, we have introduced a new paragraph entitled "Normalization and Transformation Used for Different DA Methods" in the Supplementary Information.

5. The authors should provide more details regarding how they generated the simulated data, e.g., the distribution of the data and we cannot understand the reason for the significantly high FDR of other methods, including authors previously developed one without those details. For instance, LinDA assumes a log-normal distribution of the absolute abundance. Users might generate the simulated data from Poisson lognormal distribution where the mean and covariates were estimated from relative abundance.

Response: This comment partly relates to an earlier comment by this reviewer (Comment 2 above) as well as a comment made by Reviewer 2. Firstly, as noted above, we have explained the simulation set-up in greater detail in the Supplementary Information as well as in the main text. We constructed microbial counts using the Poisson lognormal (PLN) model, drawing on the methodology delineated in the LDM paper of Hu and Satten (2020) (<https://doi.org/10.1093/bioinformatics/btaa260>). This PLN model assumes that the abundance for the j -th taxon in the i -th sample is derived from a Poisson distribution with mean $N_i \theta_j$, where N_i is the library size for sample i , and $\boldsymbol{\theta} = (\theta_1, \dots, \theta_d)^T$ obeys a multivariate log-normal distribution with mean vector $\boldsymbol{\mu}$ and variance-covariance matrix $\boldsymbol{\Sigma}$.

This model correlates the absolute abundance vector with a Gaussian latent vector. Due to the existence of a latent layer, the PLN model exhibits a greater variance than the Poisson model, reflecting over-dispersion. Furthermore, the covariance (or correlation) between absolute abundances mirrors the

covariance (or correlation) between the corresponding latent variables. The underlying multivariate Gaussian distribution allows greater flexibility in modeling the variance-covariance structure for microbial absolute abundances.

In our simulation study set-up, we refrain from manually specifying the mean vector and the variance-covariance matrix. Instead, we used estimated values of these parameters from the real dataset, namely the upper respiratory tract (URT) microbiome data. Thus, rather than choosing arbitrary and unmotivated values for various parameters in the simulation study, our choice of parameters is motivated by real data.

It is important to note that our ANCOM-BC2 methodology, was not formulated on the basis of the PLN model. Thus, data used in our simulation studies do NOT favor our method over the competing methods. The above points, along with choice of parameters used in the simulation study are highlighted in the Results section and Supplementary Information.

6. The FDR of ANCOM-BC2 is close to 0 in various parameter setup. Is this still true if the authors generate the simulated dataset differently?

Response: We appreciate the reviewer's comment and have taken it into consideration. In response to their suggestion, we have expanded our simulation studies to include the upper respiratory tract (URT) microbiome data. Furthermore, we have now included 5% differentially abundant data in our simulations. As for the simulation results using the QMP data, we have relegated them to Supplementary Figures 3 - 5.

Response to comments by Reviewer 2:

We are very grateful to the reviewer for their valuable time and effort and constructive comments that have led to a substantial improvement in the content and presentation of our manuscript. In the following we provide item by item responses to the comments. Reviewer's comments are in italics and our response follow in regular font.

Lin and Peddada present an extension of the original ANCOM-BC, called ANCOM-BC2, that is capable of testing for patterns and contrasts across multiple experimental groups and accounting for repeated measures. They also introduce some additional bias corrections, such as an estimation of taxon-level efficiencies and filters for false positive results due to the addition of pseudo-counts. They compare ANCOM-BC2 to a small set of competing methods on simulated data and apply it to some example analysis of sequencing data from soil and human gut microbiome samples.

*The microbiome field as a whole is still looking for methods that can detect bacterial abundance changes in amplicon and metagenomic sequencing data with good sensitivity *and* precision, and I would agree that the added designs (grouping patterns and repeated measures) are common and not always handled very well by other methods. However, in its current form, I don't feel the manuscript is making a strong argument that this was achieved. There is a lack of transparency in the used methodology to compare ANCOM-BC2 to other methods, and the presented data does not support many of the claims made regarding performance. In particular, ANCOM-BC2 looks to be severely less powered than all other tested methods for small sample sizes, which limits its applicability in its current form. If those challenges can be addressed, I do think that the method would constitute a great addition to the field.*

Response: We thank the reviewer for the summary and challenges in the microbiome literature. As noted by the reviewer, there is a paucity of methods that can handle multiple groups, pattern analyses, and repeated measurements in a principled manner. In that sense, this paper is first to address such complex designs in a principled manner using constrained inference methods.

We have made significant refinements to the variance formula of the ANCOM-BC2 test statistics and the sensitivity analysis for pseudo-count addition to zeros to improve power while maintaining a good control of FDR. This resulted in two versions of our procedure, one is called ANCOM-BC2 with the sensitivity score filter (ANCOM-BC (SS Filter)) and other is ANCOM-BC2 without the sensitivity score filter (ANCOM-BC2 (No Filter)). The results of our simulation study demonstrate that both versions of ANCOM-BC2 provide a better FDR control over all competing methods, while maintaining high power. In particular, ANCOM-BC2 (SS Filter) consistently controlled the FDR or mdFDR below the nominal level in all simulation settings considered in this paper, while maintaining high power. In contrast, ANCOM-BC2 (No Filter) emerged as

the DA method with the highest power, displaying a smaller FDR or mdFDR when compared with competing methods, other than ANCOM-BC2 (SS Filter). More detailed discussion regarding the power and FDR (and mdFDR) of our methodologies are provided below in response to your comment regarding power/FDR trade-off. Taken it all together, indeed our proposed methods outperform the existing substantially.

Major Comments

The analysis of method performance based on simulated data is a key outcome of the paper. So it was surprising to find basically no description of how the simulation was carried out. In particular, I would have liked to see the relevant key parameters of such a simulation such as the added effect size, the level of noise added to the data, the number of independent simulation runs, and how other model parameters were estimated from the QMP data.

Response:

This comment relates to a comment made by Reviewer 1. Firstly, as noted above, we have explained the simulation set-up in greater detail in the main text as well as the Supplementary Methods. We constructed microbial absolute abundances using the Poisson lognormal (PLN) model, drawing on the methodology delineated in the LDM paper of Hu and Satten (2020) (<https://doi.org/10.1093/bioinformatics/btaa260>). This PLN model presumes that the abundance for the j -th taxon in the i -th sample is derived from a Poisson distribution with mean $N_i\theta_j$, where N_i is the library size for sample i , and $\boldsymbol{\theta} = (\theta_1, \dots, \theta_d)^T$ obeys a multivariate log-normal distribution with mean vector $\boldsymbol{\mu}$ and variance-covariance matrix $\boldsymbol{\Sigma}$.

This model correlates the abundance vector with a Gaussian latent vector. Due to the existence of a latent layer, the PLN model exhibits a greater variance than the Poisson model, reflecting over-dispersion. Furthermore, the covariance (or correlation) between absolute abundances mirrors the covariance (or correlation) between the corresponding latent variables. The underlying multivariate Gaussian distribution allows greater flexibility in modeling the variance-covariance structure for microbial absolute abundances.

In our simulation study set-up, we refrain from manually specifying the mean vector and the variance-covariance matrix. Instead, we used estimated values of these parameters from the real dataset, namely the upper respiratory tract (URT) microbiome data. Thus, rather than choosing arbitrary and unmotivated values for various parameters in the simulation study, our choice of parameters is motivated by real data. We have relocated the simulation results using the QMP data to Supplementary Figures 3 - 5.

It is important to note that our ANCOM-BC2 methodology, was not formulated on the basis of the PLN model. Thus, data used in our simulation studies do NOT favor our method over the competing methods. The above points, along with choice of parameters used in the simulation study are highlighted in the Results section and Supplementary Information.

The authors state in the caption for Figure 1 that “that ANCOM-BC2 outperformed all competing methods in terms of uniformly small FDR (mdFDR) and comparable power”. Though the first part of that statement rings true, the second is not supported by the presented data. In fact, ANCOM-BC2 is much less powered than other methods in all tested scenarios and in small sample sizes that power difference exceeds 50% meaning that in those cases ANCOM-BC2 will fail to detect the majority of true positives. It seems like ANCOM-BC2 just takes up a different spot on the sensitivity-precision tradeoff, where it achieves high precision but sacrifices sensitivity, whereas the other tested methods do the opposite. I appreciate the authors' effort to improve FDR control which is definitely important, but so is power, especially in microbiome data where effect sizes are small and large studies require substantial funding. Maybe ranking methods by F1-score would give a more balanced comparison here.

Response: We agree with the reviewer’s comment about the trade-off between FDR and power. FDR control is one of the major challenges in the field and hence that was our focus. As noted in our response to a previous comment above, we have made significant refinements to the variance formula of the ANCOM-BC2 test statistics and the sensitivity analysis for pseudo-count addition to zeros in order to improve power. Rather than repeating, we request the reviewer to see our response to an earlier comment by the reviewer where we describe the power gains.

The reviewer makes an extremely important comment regarding trade-offs between FDR and power. This is a widespread issue in many contexts, it is hard to optimize both criteria. Motivated by this comment, we introduced a novel concept called “FDR adjusted power (FAP)”, which is defined as the $\ln\left(\frac{\text{power}}{\text{FDR}}\right)$. Thus, a method with low FDR and high power would have a high value of FAP. However, methods with very low power but equally low FDR will also yield a high FAP. Therefore, choosing a method merely based on a

high value of FAP may not be correct. Hence, akin to volcano plots commonly used in genomics, for each method one may plot FAP against the power and choose methodology that has the highest FAP score among the methods exceeding the desired power. Using the simulated data generated in Figure 1 of the main text, we compared the FAP of all competitors in Supplementary Figure 2 in Supplementary Information. As seen from the scatter plots as well as the cumulative distribution functions, for any given power (e.g., 0.8), the two ANCOM-BC2 methods outperform all competitors by a very large margin. The gains in FDR adjusted power made by the two ANCOM-BC2 methods are substantial over all the competitors, they are not modest incremental. Thus, taken together with Figures 1 - 3 and Supplementary Figures 2 - 5, we conclude that our proposed methods provide good control of FDR while achieving suitable power. The discussions on FAP can be found in the Results section as well as in the Supplementary Methods.

The FDR results in Figure 1 were a bit surprising to me. Based on the claims made in <https://doi.org/10.1038/s41467-020-17041-7> (in particular Fig. 4a) shouldn't ANCOM-BC control the FDR pretty well in the settings presented in panels (a) and (b)? Was some additional bias introduced in the simulation that can explain the inflated FDR?

Response: We thank the reviewer for this important comment. Subsequent to the publication of ANCOM-BC, we and other researchers have identified the impact of pseudo-counts on ANCOM-BC, particularly in sparse taxa scenarios such as at the OTU level. In light of this observation, we conducted comprehensive simulation studies specifically focusing on edge cases where ANCOM-BC exhibited limitations. As noted in our responses to the previous comments by this reviewer, we have now addressed the reviewer's concern by refining our methodology.

Building upon the instances where ANCOM-BC demonstrated inflated false discovery rates (FDR), we have introduced modifications to the ANCOM-BC2 methodology to enhance FDR control. These modifications are as follows:

1) Regularization of variance: Borrowing from the framework of the Significance Analysis of Microarrays (SAM) methodology (Tusher, Tibshirani, and Chu 2001), ANCOM-BC2 incorporates a small positive constant into the denominator of its taxon-specific test statistic. This mitigates any undue influence from extremely small standard errors, which are often associated with rare taxa.

2) Sensitivity analysis for pseudo-count addition to zeros: Much like other differential abundance analysis techniques, ANCOM-BC2 necessitates a log transformation of the observed counts. However, the presence of zero counts creates a challenge, and the addition of a pseudo-count prior to the log transformation is a commonly employed strategy. Existing literature indicates that the selection of the pseudo-count can considerably influence the outcome and potentially augment the false positive rate (Costea et al. 2014; Paulson, Bravo, and Pop 2014). To mitigate this concern, we conduct a rigorous sensitivity analysis to evaluate the impact of varying pseudo-counts on zero counts for each taxon. This procedure incorporates the addition of an array of pseudo-counts (ranging from 0.01 to 0.5 in increments of 0.01) to the zero counts of each taxon. Corresponding to each pseudo count, this step is followed by a linear regression model using the bias-corrected log count table, derived from the ANCOM-BC2 bias-correction procedure. The sensitivity score for each taxon is then ascertained as the proportion of instances that the p-value exceeds the pre-specified significance level. If the p-values consistently result in significance or non-significance across the various pseudo-counts and align with the results obtained without adding pseudo-counts to zero counts (the ANCOM-BC2 default setting), then the taxon is deemed to be insensitive to the pseudo-count addition.

With these enhancements, ANCOM-BC2 is well-equipped to control FDR, even with highly sparse microbiome data such as those encountered at the OTU level. The simulation outcomes corroborate that the integration of the sensitivity score filter enables ANCOM-BC2 (SS Filter) to maintain FDR or mdFDR consistently beneath the nominal level while retaining significant power. Even without the sensitivity score filter, ANCOM-BC2 (No Filter) has proven to be the most powerful differential abundance method, exhibiting a lower FDR or mdFDR compared to its rival methods.

In that vein, some of the introduced strategies are not specific to ANCOM-BC2. For instance, the presented method of filtering results sensitive to pseudo-count addition in Remark 2 is mostly a post-hoc filter that could be used with any of the competing methods. From Supplementary Figure 2, it looks like this could actually be responsible for the better FDR control. So I wonder how Figure 1 would look like if that post-hoc filter would be added to all tested methods.

Response: We thank the reviewer for this important comment. It is worth mentioning that the pseudo-count sensitivity analysis does not apply to LOCOM and CORNCOB, as they operate on relative abundances and do not involve log transformation. Our sensitivity analysis was specifically designed for our own bias-correction procedure and may not be directly applicable to other methods. Consequently, although LinDA may potentially benefit from strategies such as our pseudo-count sensitivity analysis, it is not clear how one would do it as it may require developing a new strategy and code for LinDA which is beyond the scope of this paper.

The authors state that “By using real data as a template, we ensured that the data-generating process did not favor our methods, enabling a fair comparison across all methods.”, but it’s impossible to assess that since the manuscript does not specify the actual statistical model used to generate the data. In the mentioned ANCOM-BC2 R package one of the vignettes seems to use a Poisson Log-Normal Model (PNLM) to model the simulated data, but I would argue that this does intrinsically favor methods with similar distribution assumption which includes ANCOM-BC2. Would the results look similar if the simulated data was generated from a Negative Binomial, Beta-Binomial, or a zero-inflated discrete count model?

Response: We thank the reviewer for the comment about the data generative process. We request the reviewer to refer to our responses to their earlier comments.

The set of alternative methods tested is pretty small and limited to methods making similar assumptions as ANCOM-BC(2). How would this look when comparing to other commonly used methods such as ALDEX2 (<https://doi.org/10.1186/2049-2618-2-15>), DESEQ2 (<https://doi.org/10.1186/s13059-014-0550-8>), CORNCOB (<https://doi.org/10.1214/19-aas1283>), or any of the other methods tested in the original LinDA manuscript (<https://doi.org/10.1186/s13059-022-02655-5>)?

Response: We appreciate the reviewer's comment regarding the comparison of our methodology with other existing methods. The key innovation of ANCOM-BC2 is its capacity to undertake multi-group tests, such as pairwise comparisons, testing against a control group, and pattern analysis, while concurrently controlling the FDR or the mixed directional FDR (mdFDR). From our understanding of the current literature, ANCOM-BC2 is distinct in its ability to test a diverse set of hypotheses in multiple group studies.

Nonetheless, for benchmarking purposes, we have compared ANCOM-BC2 with two recently published methods, LinDA (published in *Genome Biology*, 2022) and LOCOM (published in *PNAS*, 2022), for testing hypotheses for which these methods were specifically designed. Our prior work (Lin, Peddada, *Nat. Comm.*, 2020) demonstrated the superior performance of ANCOM-BC in terms of FDR control and enhanced statistical power over other well-known methods such as metagenomeSeq, Differential Ranking (DR), DeSEQ2, and edgeR. Considering that both LinDA and LOCOM have demonstrated their superiority over ALDEx2, we chose not to include these methods in our comparisons for this manuscript.

However, as suggested by the reviewer, we have now included CORNCOB, because it was not benchmarked in any of the previous papers noted above.

Minor Comments

Additional to the methods mentioned above, it might be worthwhile to add a baseline comparison where ANCOM-BC2 is compared to some regular mixed effects models on the CLR-transformed data, because that setup is often used for repeated measures analyses of microbiome data.

Response: We thank the reviewer for this suggestion and have now added linear mixed models on the CLR-transformed data (LMM-CLR) in the corresponding simulation studies.

The code provided in the reproducible capsule is lacking some comments and general explanation of what is happening. It's really hard to follow it the way it is presented right now (pure wall of code with no explanations in the Rmarkdown files).

Response: We have polished the code and improved its readability.

Neither of the illustration examples explains how the raw data was processed and reads were mapped to taxa (which seems to be the input the authors used). This would be fine in the supplement.

Response: Thank you for bringing up this issue. We would like to clarify that the data used in the illustration examples were downloaded directly from the corresponding publications. These datasets have already undergone preprocessing by the original authors, and we did not perform any additional preprocessing steps on these data.

Response to comments by Reviewer 3:

We are very grateful to the reviewer for their valuable time and effort and constructive comments that have led to a substantial improvement in the content and presentation of our manuscript. In the following we provide item by item responses to the comments. Reviewer's comments are in italics and our response follow in regular font.

In this paper, Lin and Peddada extended their earlier work on ANCOM, ANCOM-BC to ANCOM-BC2. The new approach allows for two types of bias: the sample specific bias which is considered to be the same across taxa in the same sample and the taxon-specific bias that is considered to be the same across samples but different for different taxa. The paper also focused on multi-group analysis in which they investigate directional hypothesis, pairwise group comparison, trend test or pattern test. Have a method that is able to adjust for bias is a very important and difficult task for microbiome data analysis, given that there is mounting evidence indicating the existence and huge effect of taxon-specific bias. The proposed method is very complex with some heuristic decisions. Please see the following as my major comments:

Response: We appreciate the summary provided by the reviewer. The problems described in this paper are rather complex but commonly encountered in microbiome studies and hence a methodology such as ANCOM-BC2 was needed. The methodology uses ideas from multiple testing and multiple comparison procedures as well as constrained statistical inference, which involve many technical details as described in the paper. The only heuristic component of the methodology is the threshold used in the pseudo-count sensitivity analysis, which we have now reformulated based on the comments we received from all reviewers. The updated sensitivity analysis, as described in the Methods section of the main text, eliminates the need for users to choose a cutoff value.

1. The definition of "taxon-specific" bias is unclear to me. As I understand it, the taxon-specific bias is generated because some taxa are easier to measure than others, i.e., the sampling fraction is not uneven for different taxa even within a sample. However, in the method section, the author also stated that "Since the sampling fraction is constant for all taxa within a sample, we pool information across taxa within each sample when estimating δ ", which is contractive to my understanding of bias. Could the authors explain this point in more detail?

Response: Basically, we assert that there are two sources of biases. One is sample-specific and the other is taxon-specific. After eliminating the taxon-specific bias, by centering the data, we are left with bias specific

to the sample. For example, different samples may have different library sizes that can lead to differences between samples but within sample all taxa are similarly affected.

2. Was the underlying hypothesis a compositional hypothesis or a hypothesis on relative abundance? In specific, suppose that we have three taxa with real abundance of 100, 200, 300 in one sample, and abundance of 500, 200, 300 in another sample. If we are interested in the compositional hypothesis, only taxon one is differentially abundant. However, if we are interested in the differences in relative abundances, all three taxa are DA. Through the log-linear model setup, I feel that ANCOM-BC2 conducts hypothesis testing at the compositional level. However, this was not clear to me.

Response: Thank you for your insightful comment. We acknowledge that our hypothesis is formulated specifically for differential absolute abundance and not relative abundance. The inclusion of sample-specific bias allows us to account for the underlying compositionality in the data. We have now clarified in the third paragraph of the main text that ANCOM-BC2 is designed to focus on the analysis of differential absolute abundance. In our simulation studies, we evaluated the performance of all differential absolute abundance analysis methods, except CORNCOB. It should be noted that CORNCOB is specifically designed to ascertain differential relative abundances. However, we consciously incorporated CORNCOB into our simulation benchmarking. The motivation behind this inclusion was to extend the performance assessment to examine how differential relative abundance analysis methods could fare under the conditions characteristic of differential absolute abundance analysis.

3. Given the algorithm is quite complex, it may be helpful if a flowchart is created to inform potential users the analysis steps using the ANCOM-BC2 algorithm, providing summary information about which test option is used (such as asymptotic results or the permutation test based on some specific statistics).

Response: We thank the reviewer for this suggestion. We have now included a flowchart in Supplementary Fig. 6.

4. I am not exactly sure how β_j can be calculated at equation (4) and use the iterative approach. To me, both ϑ_j and β_j are non-identifiable. Does the proposed iterative approach achieve only one out of infinitely many parameters from the entire space? The rationale behind obtaining the bias-correction term δ was not clear to me.

Response: Thank you so much for this comment. In a standard linear regression model setting with a single outcome variable (i.e., single taxon) θ_i is not identifiable and hence not estimable. However, in the present setting it is identifiable because corresponding to each θ_i , we have several taxa $j = 1, 2, \dots, d$, on

each subject. Hence the subject-specific sampling fraction θ_i is estimable by borrowing information across all taxa, which is done by our iterative Algorithm 1. Thus, we have an identifiable and an estimable parameter θ_i . Once the biasing constant θ_i corresponding to the i^{th} subject is estimated in a given iteration, the taxon-specific abundance parameter β_j , $j = 1, 2, \dots, d$, is completely identifiable and estimable using the iterative least squares algorithm (Algorithm 1).

5. I worry about the FDR control of the proposed method in more realistic setups. Many decisions in the algorithm are quite heuristic, such as the choice of sensitivity score cutoff (for the pseudo-count imputation) and the choices of multiple comparison adjustment. Heuristic decisions usually generate methods that control type I error and FDR in some simulation set up, but not in others. For example, does the method control FDR when the sample size is much larger? How about in situations when the taxon-specific bias factors are very big? In the simulation, the authors assessed situations when the bias factor is sampled from $C \sim U[0.1, 1]$. However, experiment from mock community shows that the bias factor can be very uneven. How is the model performance under more extreme bias setups?

Response: As mentioned earlier, we have addressed the reviewer's comment by reformulating the sensitivity analysis. The updated approach, outlined in the Methods section of the main text, eliminates the requirement for users to manually choose a cutoff value. We assert that the taxon-specific bias, while varying between taxa, remains consistent within a given taxon across samples. Therefore, by centering the observed abundances across samples, we can effectively mitigate this source of bias. Consequently, we believe that the taxon-specific bias should have minimal influence on the results of the differential abundance analysis. Instead, we believe that the sample-specific bias is a more significant factor in this context. Accordingly, we conducted extensive analyses that highlight scenarios where the sample-specific bias contributes to the presence of rare taxa and strong batch effects, particularly in cases where ANCOM-BC (the earlier version of ANCOM-BC2) failed.

6. The cutoff of the sensitivity score for pseudo-count imputation is quite heuristic. It would be better if the authors can explain the rationale of choosing of sensitivity score of 3 as cutoff.

Response: See above.

7. The comparison on FDR is unfair for the competing methods because they used the BH approach for FDR controls yet the proposed method used Bonferroni correction, which is designed to control the family wise error rate instead of FDR.

Response: As we noted in the previous version of the paper, indeed, all the methods employed the Holm-Bonferroni correction for controlling the FDR, not the BH procedure. It is important to note that controlling

the family-wise error rate (FWER) also ensures control of the FDR. We have explicitly mentioned this in the Results section of the main text, in the Supplementary Methods, and our code.

8. What is the computational time of the proposed algorithm, and compared to the competing methods?

Response: We appreciate the reviewer's suggestion. In response to this feedback, we have included a new section titled "Computational Efficiency and Performance Benchmarking of Various DA Methods" in the Supplementary Methods.

9. The paper can also benefit from the discussion on the sparsity level of the microbiome taxa. Are rare taxa more likely to fail the sensitivity score cutoff compared to common taxa? If this is the case, does it indicate that we should filter the data more extensively? Some of the competing methods, such as LOCOM, generally requires filtering the taxa and only keep taxa that are present in a sufficient number of samples. I am wondering whether the FDR inflation can be due to the insufficient number of data? After all, in the simulation setup, the number of samples can go as low as 10.

Response: We would like to clarify the data preprocessing steps performed for the differential abundance analysis methods considered in the simulation studies. As often done in practice, for ANCOM-BC2, ANCOM-BC, LinDA, and LOCOM, we filtered taxa with less than 10% prevalence across samples and excluded samples with a library size (total counts across taxa) of less than or equal to 1000 counts. For CORNCOB, which is a differential relative abundance analysis method, we excluded samples with a library size of less than or equal to 1000 counts. No taxon filter was applied for CORNCOB as its current implementation does not incorporate this feature. I have now stated all these preprocessing steps explicitly in Simulation Details section in Supplementary Information section. To examine the edge cases where ANCOM-BC (the predecessor of ANCOM-BC2) failed, we intentionally introduced sample-specific bias to ensure the representation of rare taxa (see the Results section), which exhibit over 50% zeros across samples. In our simulation settings, the smallest sample size per group was set to 10, as suggested by the reviewer. Please refer to Figures 1-3 in the main text and Supplementary Figures 3-5 in the Supplementary Information for the simulation results.

10. In assumption 4 of page 12, a sparse correlation was assumed for the estimation of δ . However, in microbiome data, because all bacterial live in the same community, it is possible/or even likely that the correlations between taxa are dense instead of sparse. How critical is the sparse correlation assumption is for the algorithm?

Response: We appreciate the reviewer's insight regarding the correlation structure within microbial communities and the sparse correlation assumption made in our study. In light of our recent publication on

the SECOM method (<https://doi.org/10.1038/s41467-022-32243-x>), we agree that sparse correlation is prevalent in the microbiome datasets we have analyzed. For instance, in our SECOM study, we investigated correlations among genera in two ecosystems (the forehead and palm), and our findings demonstrated that correlations among genera within each site were largely sparse (refer to Fig. 5 in <https://doi.org/10.1038/s41467-022-32243-x>).

Based on these observations, we believe that the sparse correlation assumption is practically reasonable based on our experience with real data and this is implicitly assumed in most DA methods, including the competing methods employed in our simulation studies. Therefore, we believe that this assumption is generally reasonable in the context of microbiome DA analysis.

Decision Letter, first revision:

Our ref: NMETH-A52201A

29th Aug 2023

Dear Dr. Peddada,

Thank you for your letter detailing how you would respond to the reviewer concerns regarding your Article, "Multi-group Analysis of Compositions of Microbiomes with Covariate Adjustments and Repeated Measures" (NMETH-A52201A). After careful discussion with my colleagues, we'll be happy in principle to publish it in Nature Methods, pending minor revisions to satisfy the referees' concerns and to comply with our editorial and formatting guidelines.

TRANSPARENT PEER REVIEW

Please note: we allow redactions to authors' rebuttal and reviewer comments in the interest of confidentiality. If you are concerned about the release of confidential data, please let us know specifically

what information you would like to have removed. Please note that we cannot incorporate redactions for any other reasons. Reviewer names will be published in the peer review files if the reviewer signed the comments to authors, or if reviewers explicitly agree to release their name. For more information, please refer to our [FAQ page](https://www.nature.com/documents/nr-transparent-peer-review.pdf).

ORCID

Sincerely,
Lei

Lei Tang, Ph.D.
Senior Editor
Nature Methods

Reviewer #1 (Remarks to the Author):

In the revised manuscript, the authors have addressed some of our previous comments. However, we still have major concerns regarding the comparison of the proposed method with competing methods, which was also mentioned by other Reviewers. Therefore, we feel the current work does not warrant a publication on Nature Methods. Our specific comments are as follows:

1. As pointed out by Reviewer #3, the extremely low FDR is due to the use of Bonferroni correction, instead of BH control. Bonferroni correction will make the FDR extremely low, even 0. Thus, it is unfair to compare the FDRs produced by two approaches.
2. In the revised Figure 1, the superior power of ANCOM-BC2 is only obvious for very small sample sizes, e.g., 10 and 20. But we did not see any big difference between the power of ANCOM-BC2 with that of other methods if the sample size is larger, e.g., 30, 50, or 100. Also, the computational complexity of ANCOM-BC2 is much higher.
3. The real microbiome data is typically zero-inflated. However, the simulated data generated by the authors almost does not include zeros at all (see the uploaded Figure). All taxa are present in almost all samples (in total 500).

Reviewer #2 (Remarks to the Author):

I thank the authors for their careful consideration of my previous comments and the additional work that was added in addressing them. I agree with the authors that their revised method now does support the claim that ANCOM-BC2 has generally equal or better power than competing methods, while controlling the (m)FDR much better. I feel that the manuscript is now in a pretty good spot and would probably be fine without further adjustments. Nevertheless, I do have some suggestions on the text in case the authors want to integrate them.

Suggestions:

Since the authors argue that their model does not assume and underlying PLN model it would be helpful if the text mentions somewhere what assumptions are made in terms of distributions by ANCOM-BC2 in a way that is understandable by a general audience.

I think the other reviewer brought up a good point asking about computational efficiency/complexity. While I agree with the authors that ANCOM-BC2 is plenty fast for a single dependent variable, some of us are running models against many dependent variables (like untargeted metabolite abundances, for instance). So mentioning that the improved FDR-control and power comes at a slight computational cost in the discussion might help readers to pick a method for a specific dataset in the future.

Reviewer #3 (Remarks to the Author):

The authors have addressed all my previous questions. I have no future comments.

Author Rebuttal, first revision:

Item by item responses to Reviewer #1's comments

We thank the reviewer for their thorough and detailed comments on our revision. We appreciate their time and effort in reviewing our manuscript. In the following address each comment. Reviewer's comments are in italics and our responses follow their comments.

1. As pointed out by Reviewer #3, the extremely low FDR is due to the use of Bonferroni correction, instead of BH control. Bonferroni correction will make the FDR extremely low, even 0. Thus, it is unfair to compare the FDRs produced by two approaches.

Response:

- a) Firstly, we hope the reviewer is not under the impression that we are using different multiple testing correction procedures for the different methods investigated in our paper. That is not the case. As we indicated in the section "Simulations: Settings" in the main text of the paper, as well as in the "Simulation Details" section in the Supplementary Methods, **ALL** DA analysis methods were

evaluated using the Holm-Bonferroni method rather than the Benjamini-Hochberg (BH) procedure. We also note this in the code we provided.

- b) **Why Holm-Bonferroni over the BH procedure?** Unlike many other high-dimensional data, the microbiome is an ecology with complex interactions among microbes. Some of these interactions may even be non-linear (please refer to Lin et al., Nature Comm., 2022 and various references noted therein). Thus, the pairwise correlations are rather complex. The BH procedure mentioned by the reviewer, controls the false discovery rate (FDR) if the underlying microbial abundances satisfy positive regression dependence (Theorem 1.2, Benjamini and Yekutieli, Ann. Statist. 29(4), 1165-1188, August 2001)). This special assumption is not easy to verify for microbiome data (and may not even be true!) because some correlations may be positive, some may be negative, and some may be non-linear. Thus, in the context of microbiome data, we cannot be certain that we will control the FDR using the BH procedure. On the other hand, the Holm-Bonferroni procedure is agnostic to the underlying correlation structure. Thus, we make fewer assumptions about the unknown relationships.
- c) Even after using the conservative Holms-Bonferroni procedure, methods such as LOCOM and LinDA have extremely high FDRs, sometimes as high as 0.7 or even more (see Fig. 1b, panel 1 in the main text). Thus, if 100 taxa are declared to be differentially abundant by these procedures, on average, as many as 70 (or sometimes even more) are false discoveries. Had we used BH procedure, these rates can potentially be even higher.
- d) As an alternative to Holms-Bonferroni procedure, we could have considered Benjamini-Yekutieli (BY) method (Theorem 1.3, Benjamini and Yekutieli, Ann. Statist. 29(4), 1165-1188, August 2001)). Unlike the BH procedure, the BY procedure is expected to control the FDR at the nominal level for any arbitrary correlation structure if the raw p-values are uniformly distributed under the null hypothesis. The BY procedure is expected to be less conservative (i.e., more powerful) than Holms-Bonferroni procedure. We were hesitant to use the BY procedure because intrinsically, due to various sources of biases in microbiome data (e.g., compositionality), under the null hypothesis the raw p-values for DA methods may not be uniformly distributed. Consequently, the BY corrected p-values may not result in FDR control. As clearly seen from the results of our simulation studies reported in the paper, even after using the conservative Holms-Bonferroni procedure the methods such as LOCOM, LinDA, CORNCOB and ANCOM-BC failed to control the FDR. It may get only worse if we used the BY procedure let alone the BH procedure.

To illustrate the performance of all the methods using the BY procedure, we repeated the simulation corresponding to the settings used in Fig. 1b of the main text, and the numerical values of FDR and power are summarized in Table A.2 below. As a comparison, Table A.1 contains the numerical values corresponding to Fig. 1b, which used the Holms-Bonferroni procedure. As expected, the entries in Table A.2 are systematically larger than the corresponding entries in Table A.1. Specifically, the FDRs are much larger if we used BY procedure, although there is gain in power. It is interesting to note that, even using the BY procedure, our proposed ANCOM-BC2 (SS Filter) always controlled the FDR within the nominal rate of 0.05, while being highly competitive in terms of power with all other methods. On the other hand, ANCOM-BC2 (No Filter) had a power at least as large as all other methods while maintaining low FDR compared to all others, with the exception of ANCOM-BC2 (SS Filter). If one were to consider FDR Adjusted Power (FAP), the novel concept

proposed in this paper to represent the power/FDR trade-off, then ANCOM-BC2 (SS Filter) had the highest value followed by ANCOM-BC2 (No Filter). In summary, whether we use Holms-Bonferroni or the BY procedure, our two proposed methods ANCOM-BC2 have the lowest FDR while competing very well in terms of power. On the other hand, the competing methods have a highly inflated FDR which is very undesirable. The reviewer’s suggestion to use the BH procedure will only make it worse for the competing methods and as a matter of statistical principles, it may NOT be valid to use the BH procedure for controlling FDR in the present context and hence, in general should be avoided.

Table A.1

Method	N = 10			N = 20			N = 30			N = 50			N = 100		
	Power	FDR	FAP	Power	FDR	FAP	Power	FDR	FAP	Power	FDR	FAP	Power	FDR	FAP
ANCOM-BC2 (No Filter)	0.93 (0.07)	0.01 (0.03)	10.68 (0.03)	1 (0.01)	0.02 (0.04)	9.19 (0.04)	1 (0)	0.03 (0.07)	8.18 (0.07)	1 (0)	0.06 (0.1)	6.43 (0.1)	1 (0)	0.13 (0.17)	4.34 (0.17)
ANCOM-BC2 (SS Filter)	0.76 (0.09)	0 (0.01)	10.95 (0.01)	0.87 (0.06)	0 (0.01)	10.96 (0.01)	0.89 (0.05)	0 (0.03)	10.71 (0.03)	0.93 (0.04)	0.01 (0.03)	10.5 (0.03)	0.96 (0.04)	0 (0.03)	10.8 (0.03)
ANCOM-BC	0.86 (0.1)	0.18 (0.28)	6.18 (0.28)	0.93 (0.07)	0.26 (0.31)	4.75 (0.31)	0.95 (0.06)	0.35 (0.33)	3.08 (0.33)	0.96 (0.06)	0.46 (0.31)	1.41 (0.31)	0.98 (0.04)	0.57 (0.31)	0.83 (0.31)
CORNCOB	0.71 (0.11)	0.21 (0.16)	2.27 (0.16)	0.85 (0.08)	0.25 (0.15)	1.71 (0.15)	0.9 (0.08)	0.27 (0.15)	1.56 (0.15)	0.94 (0.08)	0.33 (0.15)	1.27 (0.15)	0.96 (0.07)	0.4 (0.18)	1.04 (0.18)
LinDA	0.78 (0.12)	0.08 (0.16)	7.56 (0.16)	0.89 (0.11)	0.19 (0.26)	5.2 (0.26)	0.92 (0.1)	0.27 (0.3)	3.88 (0.3)	0.94 (0.08)	0.39 (0.33)	2.31 (0.33)	0.96 (0.06)	0.51 (0.33)	1.05 (0.33)
LOCOM	0.61 (0.39)	0.13 (0.25)	4.9 (0.25)	0.95 (0.15)	0.24 (0.3)	3.87 (0.3)	0.98 (0.1)	0.3 (0.31)	3.03 (0.31)	1 (0.02)	0.34 (0.32)	2.48 (0.32)	1 (0)	0.36 (0.33)	2.25 (0.33)

Table A.2

Method	N = 10			N = 20			N = 30			N = 50			N = 100		
	Power	FDR	FAP	Power	FDR	FAP	Power	FDR	FAP	Power	FDR	FAP	Power	FDR	FAP
ANCOM-BC2 (No Filter)	0.97 (0.04)	0.01 (0.05)	10.02 (0.05)	1 (0.01)	0.04 (0.09)	7.76 (0.09)	1 (0)	0.07 (0.12)	6.06 (0.12)	1 (0)	0.1 (0.15)	5 (0.15)	1 (0)	0.19 (0.21)	3.16 (0.21)
ANCOM-BC2 (SS Filter)	0.77 (0.09)	0 (0.02)	10.84 (0.02)	0.86 (0.06)	0 (0.03)	10.83 (0.03)	0.89 (0.05)	0.01 (0.05)	10.51 (0.05)	0.93 (0.05)	0.01 (0.05)	10.37 (0.05)	0.96 (0.04)	0.01 (0.05)	10.6 (0.05)
ANCOM-BC	0.91 (0.08)	0.23 (0.32)	5.32 (0.32)	0.96 (0.04)	0.37 (0.35)	3.58 (0.35)	0.98 (0.03)	0.44 (0.35)	2.44 (0.35)	0.98 (0.03)	0.53 (0.33)	1.12 (0.33)	0.99 (0.01)	0.6 (0.33)	0.85 (0.33)
CORNCOB	0.77 (0.1)	0.25 (0.18)	1.88 (0.18)	0.88 (0.08)	0.29 (0.17)	1.49 (0.17)	0.92 (0.08)	0.32 (0.17)	1.43 (0.17)	0.95 (0.07)	0.37 (0.17)	1.16 (0.17)	0.96 (0.07)	0.44 (0.2)	0.96 (0.2)
LinDA	0.85 (0.12)	0.13 (0.21)	5.87 (0.21)	0.93 (0.11)	0.29 (0.31)	3.44 (0.31)	0.94 (0.1)	0.37 (0.33)	2.45 (0.33)	0.95 (0.1)	0.47 (0.33)	1.22 (0.33)	0.96 (0.08)	0.56 (0.32)	0.83 (0.32)
LOCOM	0.61 (0.38)	0.12 (0.24)	4.86 (0.24)	0.94 (0.16)	0.24 (0.28)	3.59 (0.28)	0.98 (0.09)	0.28 (0.29)	2.98 (0.29)	1 (0.01)	0.33 (0.3)	2.59 (0.3)	1 (0.01)	0.35 (0.31)	2.5 (0.31)

2. In the revised Figure 1, the superior power of ANCOM-BC2 is only obvious for very small sample sizes, e.g., 10 and 20. But we did not see any big difference between the power of ANCOM-BC2 with that of other methods if the sample size is larger, e.g., 30, 50, or 100. Also, the computational complexity of ANCOM-BC2 is much higher.

Response: We believe this is an inaccurate comment by the reviewer. Perhaps from the figures we provided in the main text of the paper, it was difficult for the reviewer to discern the differences in power and FDR among the various methods. To help you see the differences better, as noted earlier, in Table A.1 we

summarize the exact numerical values of power, FDR, and FAP of various methods from Fig. 1b. We will be happy to create a supplementary file containing the actual values corresponding to Figures 1 – 3 that might help the reader.

Regarding the reviewer's question, here is a general comment about all reasonable statistical tests. As the sample size grows, every good statistical test should increase in power to 1 (i.e., 100% power) and thus, for large sample sizes, the distinctions between methods in terms of power may not be appreciable. However, it is important that the FDR of the methods **should not be too large**. As can be seen from the above Table A.1, all methods have powers (highlighted in green) within a few percentage points of 100% when $n = 50$ or 100 . However, it is striking that only the two ANCOM-BC2 procedures have relatively much smaller FDRs (highlighted in yellow). In particular, ANCOM-BC2 (SS filter) controls the FDR well within the nominal level of 0.05. In fact, ANCOM-BC2 (SS filter) has best of both worlds – very small FDR while having a power as good as LOCOM, LinDA and others. In simpler terms, when the sample size is 100, out of 100 taxa declared as differentially abundant by ANCOM-BC2(SS filter) on average 0 of them are falsely declared as differentially abundant, whereas LinDA, LOCOM and CORNCOB would falsely declare 51, 36, and 40 as differentially abundant (Table A.1 in bold red), respectively. **For all sample sizes**, ANCOM-BC2 (No filter) maintained an extremely high power, often at least as large as all the competing methods, while maintaining a substantially smaller FDR than LOCOM, LinDA, CORNCOB, and ANCOM-BC. For example, when $n = 30$ (the case identified by Reviewer #1), ANCOM-BC2 (No filter) had a power of 1 (i.e., 100%) with FDR of only 0.03. Thus, it discovered all the true positives and if it identified 100 DA taxa, then only 3 were not truly differentially abundant (Table A.1 bold brown). On the other hand, for this same case, LOCOM had a power of 0.98 (98%) but had an unacceptably high FDR of 0.30. Thus, if it identified 100 DA taxa, then as many as 30 were not in reality differentially abundant. Similarly, poor numbers are seen for other competing methods such as LinDA and CORNCOB. Thus, ANCOM-BC2 (No filter) enjoys best of both worlds. If FDR is the priority, then for all sample sizes, ANCOM-BC2 (SS filter) always controlled FDR within the nominal level while maintaining high power. Even in the instances when it had zero estimated FDR, which was concerning to the reviewer, it had very good power. As commented by one of the reviewers of the previous draft, the balance between FDR and power is important in practice, hence we introduced FDR Adjusted Powers (FAP) in our revision. In all our simulations we found ANCOM-BC2 (SS filter) and ANCOM-BC2 (No filter) had the highest FAP values (Supplementary Figure 2). Please also refer to the above Table A.1. Here too, ANCOM-BC2 (SS filter) and ANCOM-BC2 (No filter) had the largest FAP as compared to all competing methods across all different sample sizes (highlighted in blue).

With regards to the comment of computational complexity, once again we are surprised by the comment. To illustrate the computation times of various methods, we calculated the CPU times of all the methods considered in this paper for analyzing the "atlas1006" dataset (Lahti et al. (Nat. Comm. 5:4344, 2014, <https://doi.org/10.1038/ncomms5344>). Since not all methods are designed to test hypothesis regarding multiple groups, we focused on two groups, namely, "lean" and "obese" subjects. Furthermore, to circumvent complications associated with repeated measures - given that several of the methods under consideration are not optimized for such analyses - we restricted our dataset to baseline values. As a result, there was a total of 130 genera and 630 samples for comparing the two groups. The CPU times are

summarized in Table B below. LOCOM was the slowest but even that took barely over a minute (62 seconds). Although LinDA took a fraction of a second (0.05 seconds), it comes with a huge FDR, as demonstrated in the simulations. On the other hand, ANCOM-BC2 (SS Filter) which appears to be the best in terms of FDR control, while maintaining high power, takes half the time as LOCOM and CORNCOB. ANCOM-BC2 (No Filter) is even faster, taking just about 8.58 CPU seconds. Thus, we disagree with the reviewers that our proposed methods are computationally more intensive than others. Putting into a proper perspective, if a biological or an epidemiological study can take several weeks/months or even years, what difference does it make if a computational method takes another 7 seconds or 30 seconds or even one minute to run, as long as it is reliable? We are happy to provide Table B in the Supplementary Information with the corresponding code as well.

Table B

Method	CPU Time (Seconds)
ANCOM-BC2 (No Filter)	8.58
ANCOM-BC2 (SS Filter)	29.47
ANCOM-BC	6.02
CORNCOB	59.34
LinDA	0.05
LOCOM	62.3

3. *The real microbiome data is typically zero-inflated. However, the simulated data generated by the authors almost does not include zeros at all (see the uploaded Figure). All taxa are present in almost all samples (in total 500).*

Response: We believe there is a misunderstanding by Reviewer #1 regarding inflated zeros. He/she plotted the frequency distribution of a wrong parameter, they should have plotted the distribution of the observed counts (“obs_data” in the code) and not the absolute unobservable counts (“abn_data” in the code). The zero-inflation discussed in the literature is with regards to zero counts in the observed data.

We would like to remind that our simulation studies were inspired by the following two real data sets:

(1) “atlas1006” dataset, sourced from Lahti et al. (Nat. Comm. 5:4344, 2014, <https://doi.org/10.1038/ncomms5344>), contains 130 genus-like taxonomic groups across 1006 Western adults without any documented health anomalies. Overall, this dataset contains 20.9% zeros. One can check the proportion of zeros by running the following in R:

```
data(atlas1006, package = "microbiome")
feature_table = abundances(atlas1006)
sum(feature_table == 0)/(nrow(feature_table) * ncol(feature_table))
```

(2) “dietswap” dataset, originating from O’Keefe et al. (Nat. Comm. 6:6342, 2015, <https://doi.org/10.1038/ncomms7342>), contains 130 genus-like taxonomic groups spread across 222 adults in a fortnight diet exchange study between Western (USA) and traditional (rural Africa) diets. Overall, this dataset contains 20.6% zeros. One can check the proportion of zeros by running the following in R:

```
data(dietswap, package = "microbiome")
feature_table = abundances(dietswap)
sum(feature_table == 0)/(nrow(feature_table) * ncol(feature_table))
```

We evaluated the proportions of zeros in the observed counts for the simulation study based on the URT data template with a binary exposure (settings in Fig. 1b in the main text). The box plots in Fig. A below represent the distribution of proportion of zero across all simulations for different sample sizes. We see that the average proportion of zeros is 20%, consistent with the real data that was used for simulations. The proportion of zeros ranged from 5% to 40%. We've attached the code associated with generating the Fig. A.

Fig. A

The reviewer is correct that in high-resolution microbiome analyses, especially at the OTU or species level, datasets may contain as many as 80% or more zeros. However, in standard research practice, the convention is to filter out highly sparse taxa—typically excluding taxa appearing in fewer than 90% of samples—prior to formal analyses of the data. This procedural step is evident in existing analytical tools such

as LinDA, which offers a "prev.filter" parameter, LOCOM with its "prev.cut" option (set at a default of 0.2), and ANCOM-BC, provides a "prv_cut" option with a default threshold of 0.1. Similarly, in our ANCOM-BC2 software, we have provisioned the "prv_cut" option, maintaining the default threshold at 0.1. Upon filtering out these extreme rarities, the resultant pre-processed data generally display fewer proportions of zeros, which is corroborated by the two datasets mentioned above. However, an important distinction between ANCOM-BC2 and other procedures such as LOCOM and LinDA is that, before the above filtering step, our ANCOM-BC2 pipeline consists of using ANCOM-II procedure to discover structural zeros which are analyzed separately. These structural zeros refer to situations wherein a specific taxon is conspicuously absent, either entirely or almost so, in one particular group and not in the other. For example, some taxa specific to a desert environment may be completely absent in the rain forest, and vice-versa. Those taxa are informative and characterized by their environment. These are called structural zeros and unlike some of the above standard methods, they should not be filtered away but analyzed separately. ANCOM-BC2 does precisely that. In a binary group scenario, taxa manifesting structural zeros typically exhibit over 50% zero abundance. This aspect has been explicitly delineated in our R package documentation, accessible at the online vignette: <https://www.bioconductor.org/packages/release/bioc/vignettes/ANCOMBC/inst/doc/ANCOMBC2.html>. Perhaps Reviewer #1 missed this important feature of our method because we did not highlight it. Should revisions be permitted, we shall highlight this feature in the main text. In summation, ANCOM-BC2 employs a two-fold strategy to address the challenge of zeros. (1) As the first step, the algorithm identifies all taxa that are potentially structural zeros and hence differentially abundant between groups. (2) After separating the structural zeros, analogous to other DA methods, ANCOM-BC2 implements a filtration mechanism based on their prevalence.

While we contend that data preprocessing is a standard procedure when performing analysis of microbiomes, in response to Reviewer #1's concern, we conducted additional simulations mirroring the settings depicted in Fig. 1b but with the inclusion of additional taxa exhibiting high sparsity. The results of this additional simulation work are summarized below. The distribution of proportion of zeros in each sample is summarized in Fig. B1. As we can see from the box plots, the average proportion of zeros is 65%, and ranges between 55% and 75%. The corresponding power/FDR outcomes are provided in Fig. B2. The numerical results or power, FDR, and FAP corresponding to Fig. B2 are summarized in Table C.

Fig. B1

Fig. B2

Table C

Method	N = 10			N = 20			N = 30			N = 50			N = 100		
	Power	FDR	FAP	Power	FDR	FAP	Power	FDR	FAP	Power	FDR	FAP	Power	FDR	FAP
ANCOM-BC2 (No Filter)	0.96 (0.06)	0.01 (0.04)	10.14 (0.04)	1 (0.01)	0.04 (0.09)	7.44 (0.09)	1 (0)	0.06 (0.11)	6.38 (0.11)	1 (0)	0.1 (0.14)	4.83 (0.14)	1 (0)	0.18 (0.2)	3.12 (0.2)
ANCOM-BC2 (SS Filter)	0.76 (0.09)	0 (0.01)	10.91 (0.01)	0.87 (0.06)	0 (0.01)	10.79 (0.01)	0.9 (0.05)	0.01 (0.07)	10.51 (0.07)	0.92 (0.05)	0.01 (0.06)	10.42 (0.06)	0.96 (0.03)	0.01 (0.05)	10.66 (0.05)
ANCOM-BC	0.91 (0.08)	0.27 (0.33)	4.93 (0.33)	0.97 (0.04)	0.34 (0.35)	3.76 (0.35)	0.98 (0.03)	0.43 (0.35)	2.38 (0.35)	0.99 (0.02)	0.52 (0.33)	1.23 (0.33)	0.99 (0.02)	0.6 (0.33)	0.85 (0.33)
CORNCOB	0.76 (0.1)	0.24 (0.17)	2.06 (0.17)	0.89 (0.08)	0.29 (0.17)	1.51 (0.17)	0.92 (0.08)	0.31 (0.16)	1.37 (0.16)	0.94 (0.08)	0.38 (0.18)	1.13 (0.18)	0.96 (0.07)	0.44 (0.2)	0.96 (0.2)
LinDA	0.85 (0.12)	0.14 (0.22)	5.52 (0.22)	0.93 (0.11)	0.24 (0.28)	3.98 (0.28)	0.94 (0.11)	0.34 (0.32)	2.43 (0.32)	0.95 (0.1)	0.46 (0.34)	1.44 (0.34)	0.96 (0.08)	0.55 (0.32)	0.84 (0.32)
LOCOM	0.62 (0.37)	0.14 (0.26)	4.39 (0.26)	0.97 (0.08)	0.28 (0.29)	2.99 (0.29)	0.98 (0.08)	0.28 (0.28)	2.99 (0.28)	1 (0.05)	0.34 (0.3)	2.48 (0.3)	1 (0)	0.36 (0.31)	2.24 (0.31)

The FDRs and powers of various methods are determined using BY procedure. These findings mirror our earlier findings described in Fig. 1b of the main manuscript. Here again, both versions of ANCOM-BC2 consistently had a better control of FDR compared to the alternative methods, while still maintaining high power. Specifically, ANCOM-BC2 (No Filter) outperformed all other DA methodologies in terms of power and ANCOM-BC2 (SS) filter had uniformly smallest FDR. Importantly, all DA approaches demonstrated robustness against increased sparsity, attributable to their intrinsic filters designed to exclude excessively sparse taxa. If required, we are ready to substitute the main text's simulation settings with the high-sparsity scenarios reflected in Fig. B2.

Item by item responses to Reviewer #2's comments

We thank the reviewer for their comments on our revision. We appreciate their time and effort in reviewing our manuscript. In the following address each comment. Reviewer's comments are in italics and our responses follow their comments.

I thank the authors for their careful consideration of my previous comments and the additional work that was added in addressing them. I agree with the authors that their revised method now does support the claim that ANCOM-BC2 has generally equal or better power than competing methods, while controlling the (m)FDR much better. I feel that the manuscript is now in a pretty good spot and would probably be fine without further adjustments. Nevertheless, I do have some suggestions on the text in case the authors want to integrate them.

Suggestions:

Since the authors argue that their model does not assume and underlying PLN model it would be helpful if the text mentions somewhere what assumptions are made in terms of distributions by ANCOM-BC2 in a way that is understandable by a general audience.

I think the other reviewer brought up a good point asking about computational efficiency/complexity. While I agree with the authors that ANCOM-BC2 is plenty fast for a single dependent variable, some of us are running models against many dependent variables (like untargeted metabolite abundances, for instance). So mentioning that the improved FDR-control and power comes at a slight computational cost in the discussion might help readers to pick a method for a specific dataset in the future.

Response: We thank the reviewer for their kind remarks. The Methods section details all the assumptions made by ANCOM-BC2 and in the Supplementary text we have included a paragraph regarding the computational complexity of all methods investigated in this paper.

Item by item responses to Reviewer #3's comments

We thank the reviewer for their comment "*The authors have addressed all my previous questions. I have no future comments.*" We appreciate their time and effort in reviewing our manuscript.

Response to the final comments by Reviewer #1

Reviewer's comments are in italics which are followed by our responses in plain text.

(1) We assume those FDR values were obtained by many iterations and the final values shown in those tables are the mean and deviation over different iterations. Are FDRs of all iterations being lower than 0.05 or the average FDR over iterations is lower than 0.05?

Our Response: Mathematically, for an experiment consisting of MM hypotheses (e.g., MM microbes), suppose a statistical procedure (e.g., LOCOM) identifies RR taxa to be differentially abundant and out of those RR , suppose VV are in truth not differentially abundant. If such an experiment is repeated many times (i.e.,

large number of iterations or simulations), then FDR is defined as the average of $\frac{VV}{RR}$ taken over all such

iterations. All publications report this as the FDR, as we do too. Ideally you want this average to be

controlled within a nominal level. I do not think any statistician who would control $\frac{VV}{RR}$ for each iteration. That will be a new measure and likely to be extremely conservative.

(2) We used the code that the authors mentioned in the paper:

<https://github.com/FrederickHuangLin/ANCOM-BC2-Code-Archive/tree/main/code>.

We tested one of the simulations: "urt_sim_fixed" with binary exposure. To save time, we only used 5 iterations and 3 fractions of DA taxa

Our Response: First of all, the error message "passed_ss_bin_cov2" should have been a clear signal to the reviewer that may have used an incorrect version of the package. **Reproducibility of results is an extremely high priority for us, and we take it very seriously.** To underscore this, we've consistently included a "Session Information" section at the end of our code. This section transparently states that the ANCOMBC package version required is 2.2.2, found on our GitHub repository's bugfix branch (<https://github.com/FrederickHuangLin/ANCOMBC/tree/bugfix>). However, for those who are less inclined to use development versions, the stable release, version 2.2.1, is available on Bioconductor

and demands Bioconductor version 3.17. I'd expect that ensuring the right environment wouldn't pose a significant challenge; however, should this be the case, we've preemptively provided a Code Ocean capsule that meets all the requirements.

Furthermore, we also executed the “urt_sim_fixed” using binary exposure with 5 iterations and 3 fractions of DA taxa as the reviewer did. The outcomes are documented in Figure A1 below. Please note that FDR and power estimates based solely on 5 iterations warrant cautious interpretation due to the potential for large standard errors in the estimates, and such results might not facilitate equitable comparisons across all methodologies. Nevertheless, even with this small number of iterations, both ANCOM-BC2 methods prominently feature as the top-performing methods in terms of FDR and power, just as we claimed in the revision and our earlier letter. For the benefit of thoroughness and transparency, we have curated a dedicated GitHub repository for this specific code:

<https://github.com/FrederickHuangLin/NMETH-A52201A-Reviewer-Comments/tree/main>. While our commitment to reproducibility remains steadfast, I'd urge a more detailed examination of sections such as the “Session Information” to avoid misinterpretations in the future.

Figure A1

Regarding LOCOM, we have to clarify that in our manuscript, we treated LOCOM with the same due diligence as all other methods. We adhered rigorously to the guidelines provided on LOCOM official GitHub page (<https://github.com/yijuanhu/LOCOM-Archive>). Regrettably, LOCOM still produced 'NAs' during certain simulation runs. One might surmise that these issues stem from LOCOM's software, particularly as it has not undergone comprehensive testing nor been formally released on platforms like CRAN or Bioconductor. We were left with no choice but to work with its development version. Consequently, when determining FDR and power for LOCOM, we had to sidestep the outputs flagged as "NA" and rely solely on the non-NA results.

To be clear, the challenges we faced were not a consequence of our method of implementation but appear intrinsic to LOCOM's current code. While it might be pertinent to mention this in the manuscript, I believe it's more appropriate and scholarly to abstain from highlighting specific limitations of LOCOM, particularly when its performance has no bearing on our principal findings associated with ANCOM-

BC2.

(3) The corresponding figure that the authors saved in the result folder is shown in Figure 2. We found that this figure was generated by directly importing the results from the previously archived files, as evidenced by their code: ...

Our Response: We do not understand this comment. If the reference is to the output depicted in Figure B2 (Table C) from our last response letter — erroneously labeled as Figure 2 in the reviewer’s report — our figures and statements stand accurate as delineated in that response. On the other hand, the code that the reviewer referred to seems to be related to Figure 1 in our main manuscript. It is dedicated to generating the FDR/power figure from simulation data with binary exposure. We need to clarify that our decision to import archived files was a practical necessity. Running simulations during the knitting of the Rmarkdown file isn't feasible, especially considering that some methods, like LOCOM, can be exceedingly time-intensive. Hence, for efficiency, these simulations were executed on clusters independently. It's worth noting that the code for these cluster runs is transparently available in our GitHub repository under the “slurm_jobs” folder

(https://github.com/FrederickHuangLin/ANCOM-BC2-Code-Archive/tree/main/slurm_jobs).

Final Decision Letter:

17th Oct 2023

Dear Dr Peddada,

I am pleased to inform you that your Article, "Multi-group Analysis of Compositions of Microbiomes with Covariate Adjustments and Repeated Measures", has now been accepted for publication in Nature Methods. Your paper is tentatively scheduled for publication in our Jan 2024 print issue, and will be published online prior to that. The received and accepted dates will be 4th Apr 2023 and 17th Oct 2023. This note is intended to let you know what to expect from us over the next month or so, and to let you know where to address any further questions.

Over the next few weeks, your paper will be copyedited to ensure that it conforms to Nature Methods style. Once your paper is typeset, you will receive an email with a link to choose the appropriate publishing options for your paper and our Author Services team will be in touch regarding any additional information that may be required.

You will receive a link to your electronic proof via email with a request to make any corrections within 48 hours. If, when you receive your proof, you cannot meet this deadline, please inform us at

rjsproduction@springernature.com immediately.

Please note that *Nature Methods* is a Transformative Journal (TJ). Authors may publish their research with us through the traditional subscription access route or make their paper immediately open access through payment of an article-processing charge (APC). Authors will not be required to make a final decision about access to their article until it has been accepted. [Find out more about Transformative Journals](https://www.springernature.com/gp/open-research/transformative-journals)

Your paper will now be copyedited to ensure that it conforms to Nature Methods style. Once proofs are generated, they will be sent to you electronically and you will be asked to send a corrected version within 24 hours. It is extremely important that you let us know now whether you will be difficult to contact over the next month. If this is the case, we ask that you send us the contact information (email, phone and fax) of someone who will be able to check the proofs and deal with any last-minute problems.

If, when you receive your proof, you cannot meet the deadline, please inform us at rjsproduction@springernature.com immediately.

Once your manuscript is typeset and you have completed the appropriate grant of rights, you will receive a link to your electronic proof via email with a request to make any corrections within 48 hours. If, when you receive your proof, you cannot meet this deadline, please inform us at rjsproduction@springernature.com immediately.

Once your paper has been scheduled for online publication, the Nature press office will be in touch to confirm the details.

Once your paper has been scheduled for online publication, the Nature press office will be in touch to confirm the details.

Content is published online weekly on Mondays and Thursdays, and the embargo is set at 16:00 London time (GMT)/11:00 am US Eastern time (EST) on the day of publication. If you need to know the exact publication date or when the news embargo will be lifted, please contact our press office after you have submitted your proof corrections. Now is the time to inform your Public Relations or Press Office about your paper, as they might be interested in promoting its publication. This will allow them time to prepare an accurate and satisfactory press release. Include your manuscript tracking number NMETH-A52201B and the name of the journal, which they will need when they contact our office.

About one week before your paper is published online, we shall be distributing a press release to news organizations worldwide, which may include details of your work. We are happy for your institution or funding agency to prepare its own press release, but it must mention the embargo date and Nature Methods. Our Press Office will contact you closer to the time of publication, but if you or your Press Office have any inquiries in the meantime, please contact press@nature.com.

Nature Portfolio journals [encourage authors to share their step-by-step experimental protocols](https://www.nature.com/nature-research/editorial-policies/reporting-standards#protocols) on a protocol sharing platform of their choice. Nature Portfolio 's Protocol Exchange is a free-to-use and open resource for protocols; protocols deposited in Protocol Exchange are citable and can be linked from the published article. More details can found at www.nature.com/protocolexchange/about.

Best regards,
Lei

Lei Tang, Ph.D.
Senior Editor
Nature Methods